# On the Optimal Memorization Capacity of Transformers

**Tokio Kajitsuka & Issei Sato**
Department of Computer Science
The University of Tokyo
`{kajitsuka-tokio,sato}@g.ecc.u-tokyo.ac.jp`

## Abstract

Recent research in the field of machine learning has increasingly focused on the memorization capacity of Transformers, but how efficient they are is not yet well understood. We demonstrate that Transformers can memorize labels with $\tilde{O}(\sqrt{N})$ parameters in a next-token prediction setting for $N$ input sequences of length $n$, which is proved to be optimal up to logarithmic factors. This indicates that Transformers can efficiently perform memorization with little influence from the input length $n$ owing to the benefit of parameter sharing. We also analyze the memorization capacity in the sequence-to-sequence setting, and find that $\tilde{O}(\sqrt{nN})$ parameters are not only sufficient, but also necessary at least for Transformers with hardmax. These results suggest that while self-attention mechanisms can efficiently identify input sequences, the feed-forward network becomes a bottleneck when associating a label to each token.

## 1 Introduction

In recent years, the Transformer architecture (Vaswani et al., 2017) has played a pivotal role in the field of machine learning, becoming indispensable for a variety of models in the community. In addition to the original breakthroughs in natural language processing, such as the GPT series (Brown et al., 2020; Radford et al., 2018; 2019), it has been observed that in numerous applications, higher accuracy can be achieved by replacing existing models with Transformers. In particular, models such as the Vision Transformer (Dosovitskiy et al., 2021) in image processing and the Diffusion Transformer (Peebles & Xie, 2023) in generative tasks have demonstrated exceptional performances in a wide variety of tasks. These examples demonstrate how effective and versatile Transformers are for a diverse range of purposes.

Although the high performance of Transformers has led to their widespread use in practice, there are ongoing attempts to theoretically analyze what exactly contributes to their superior performance. In particular, one important aspect of Transformers is their representational capabilities. Previous studies have explored from a variety of angles why Transformers have high expressive capacity and can memorize vast amounts of data (Edelman et al., 2022; Gurevych et al., 2022; Takakura & Suzuki, 2023). For example, it has been shown that Transformers are universal approximators (having the ability to approximate arbitrary functions) (Yun et al., 2019) or that a particular Transformer configuration can memorize a given set of data (Kim et al., 2023; Kajitsuka & Sato, 2023; Mahdavi et al., 2023; Madden et al., 2024).

Nevertheless, while various studies have suggested that Transformers are indeed capable of memorizing data, our understanding of how efficiently they can do so remains limited. Specifically, it is not yet fully clear how certain characteristics of Transformers, such as parameter sharing, influence the reduction of model parameters and overall efficiency with respect to their **memorization capacity**, the minimum size of networks required for memorizing any sequence of a given number of data.

There are several key advantages to investigating whether a Transformer can efficiently memorize data, such as the possibility of gaining a better understanding of Transformer's strengths and providing useful insights for model design and selection. In addition, knowledge of memorization

efficiency can provide important information for evaluating generalization error (Belkin et al., 2019; Nakkiran et al., 2021). Alternatively, if it turns out that Transformers do not offer a significant efficiency advantage over feed-forward networks, it may suggest that currently widely used Transformers may in fact be substitutable for feed-forward networks.

This paper investigates the efficiency of Transformers in achieving data memorization by analyzing the necessary and sufficient model complexity for this task. To be more precise, we establish both upper and lower bounds on the number of parameters needed for memorization in the next-token prediction setting and demonstrate that they are of the same order up to logarithmic factors, thereby showing that Transformers can achieve data memorization with nearly optimal efficiency.

Furthermore, the upper bound on memorization capacity in the next-token prediction setting can be naturally extended to the sequence-to-sequence setting. This upper bound is also proved to be optimal up to logarithmic factors in the sequence-to-sequence setting, at least for Transformers with the hardmax function.

## 2 RELATED WORK

### MEMORIZATION CAPACITY

Research on memorization capacity began at least as late as the 1960s (Cover, 1965; Nilsson, 1965; Minsky & Papert, 1969). Specifically, Nilsson (1965) showed that one-hidden-layer neural networks with $N - 1$ nodes is able to compute any label assignments for $N$ data points. Later, Baum (1988) exhibited that $\lceil N/d \rceil$ neurons are sufficient for one-hidden-layer neural networks with threshold units to memorize any set of $N$ input-label pairs with the input dimension $d$, and Huang & Babri (1998); Zhang et al. (2021) extended the results to more general activation functions.

The analysis of memorization capacity is closely linked to the concept of the Vapnik-Chervonenkis (VC) dimension. While the memorization capacity of a model refers to the minimum size of the model required for memorizing *any* tuple of $N$ input-label pairs for some $N$, the VC dimension considers whether the model is capable of shattering, that is, memorizing any possible label assignments for *some* set of $N$ input points, which in turn provides a lower bound on the memorization capacity. For example, Goldberg & Jerrum (1995) estimated that the VC dimension of a feed-forward network with ReLU activation functions and $W$ parameters is at most $O(W^2)$ by reducing the network to a boolean formula. From this upper bound, it can be inferred that a feed-forward network with ReLU activation functions requires at least $\Omega(\sqrt{N})$ parameters to memorize arbitrary $N$ data points. Bartlett et al. (2019) further refined this analysis by examining the behavior of the network as a function of its parameters and analyzing it layer by layer, and demonstrated that the VC dimension of a ReLU network with width $W$ and depth $L$ is $O(WL \log W)$.

Remarkably, Park et al. (2021) proposed a construction method under the assumption that the data points are separated by at least $\delta$, showing that a feed-forward network using sigmoid or ReLU activation functions with a sub-linear parameter order $O(N^{2/3} + \log \delta)$ can memorize $N$ data points. Later, Vardi et al. (2022) demonstrated that, under similar assumptions, a ReLU network with $O(\sqrt{N \log N})$ parameters suffices for memorizing arbitrary $N$ data points. This result is *optimal* up to logarithmic factors, as it matches the lower bound $\Omega(\sqrt{N})$ implied by the VC dimension discussed above. Note that the assumption that data points are well separated is crucial to achieve sub-linear memorization capacity; in fact, it has been shown that at least $(N - 1)/2$ parameters are required to memorize arbitrary $N$ distinct data points without such separation (Sontag, 1997). Additionally, Siegel (2024) proved that $\Omega(N)$ parameters are necessary for memorizing $N$ data points when the separation $\delta$ between data points is exponentially small with respect to $N$.

Memorization capacity is not only theoretically intriguing but also practically significant. As the model size increases, classical learning theory predicts that the training error decreases while the generalization error follows a U-shaped curve. However, recent observations of the double descent phenomenon (Belkin et al., 2019; Nakkiran et al., 2021) revealed that after achieving zero training loss, the generalization error begins to decrease again. Analyzing memorization capacity helps identify the critical model size at which this shift occurs, providing valuable insights into the dynamics of model performance.

**Table 1:** Comparisons between our results and related work regarding the memorization capacity of Transformers. The variable $\omega$ in the bounds presented by Madden et al. (2024) represents the vocabulary size, or the number of distinct word vectors that appear in input sequences.

| Paper | Setting | Input | #layers | Upper bound | Lower bound |
|---|---|---|---|---|---|
| Kim et al. (2023) | seq-to-seq | token-wise $(r, \delta)$-separated | $\tilde{O}(n + \sqrt{nN})$ | $\tilde{O}(n + \sqrt{nN})$ | - |
| Mahdavi et al. (2023) | next-token | linearly independent | 1 | $O(d^2 N/n)$ | - |
| Kajitsuka & Sato (2023) | seq-to-seq | token-wise $(r, \delta)$-separated | 1 | $O(dnN + d^2)$ | - |
| Madden et al. (2024) | next-token | with positional encoding | 1 | $O(\omega N)$ | $\Omega(\omega N)$ |
| **Ours** | next-token | token-wise $(r, \delta)$-separated | $\tilde{O}(\sqrt{N})$ | $\tilde{O}(\sqrt{N})$ | $\Omega(\sqrt{N})$ |
| | seq-to-seq | token-wise $(r, \delta)$-separated | $\tilde{O}(\sqrt{nN})$ | $\tilde{O}(\sqrt{nN})$ | $\Omega\left(\sqrt{\frac{nN}{\log(nN)}}\right)$ |

EXPRESSIVITY OF TRANSFORMERS

One of the foundational studies on the representation power of Transformers is the work by Yun et al. (2019), who demonstrated that Transformers are universal approximators. Their proof already incorporates the idea of constructing a contextual mapping from data points to contexts and linking these context ids to labels. Kim et al. (2023), whose work is most closely related to our work, improved their contextual mapping approach and demonstrated that this mapping, constructed using $2n$ layers of self-attention for $N$ input sequences of length $n$, allows for memorization with $\tilde{O}(n + \sqrt{nN})$ parameters under the same assumption that data points are well separated as in Park et al. (2021); Vardi et al. (2022). Later, Kajitsuka & Sato (2023) showed that a single-layer, single-head Transformer already possesses memorization capacity under the same assumption, while self-attention with hardmax does not. In contrast to the studies mentioned above, Mahdavi et al. (2023) demonstrated that under the assumption that data points are linearly independent, a multi-head attention with $H$ heads and embedding dimension $d > n$ can memorize $\Omega(Hn)$ data points in a next-token prediction like setting. Madden et al. (2024) proved upper and lower bounds on the memorization capacity of one-layer Transformers with parameters of infinite precision in the next-token prediction setting. Chen & Zou (2024) investigated the behavior of Transformers with varying depths, and specifically demonstrated that a single-layer Transformer can achieve memorization if input sequences are sufficiently zero-padded. However, they noted that their objective was not to explore efficient constructions. The comparisons between our results and related work are summarized in Table 1. Note that all the papers listed here that investigate single-layer Transformers assume either infinite parameter precision or do not consider the bit-length required to represent parameters.

In addition to memorization capacity, there are studies highlighting other perspectives on Transformers, including their function approximation capacity (Gurevych et al., 2022; Takakura & Suzuki, 2023; Jiang & Li, 2024), and their ability to efficiently represent sparse functions (Edelman et al., 2022; Bhattamishra et al., 2023; Sanford et al., 2023; Trauger & Tewari, 2024; Wang et al., 2024b).

## 3 PRELIMINARIES

### 3.1 NOTATION

We denote vectors and matrices by bold lowercase and uppercase letters, respectively. Given a vector $\boldsymbol{v}$, we denote its $i$-th element as $v_i$. Given a matrix $\boldsymbol{A}$, we denote its $i$-th row as $\boldsymbol{A}_{i,:}$, its $j$-th column as $\boldsymbol{A}_{:,j}$ and the element at position $(i, j)$ as $A_{i,j}$. For a natural number $m \in \mathbb{N}^+$, we use $[m]$ to denote the set $\{1, \ldots, m\}$. In the context of the self-attention mechanism, we use $\sigma_S$ to represent the column-wise softmax function. Specifically, for a matrix $\boldsymbol{A} \in \mathbb{R}^{a \times b}$, $\sigma_S[\boldsymbol{A}] \in \mathbb{R}^{a \times b}$ is calculated by $\sigma_S[\boldsymbol{A}]_{i,j} := \exp(A_{i,j}) / \sum_{k=1}^{a} \exp(A_{k,j})$. Likewise, we use $\sigma_H$ to denote the column-wise hardmax function. Note that if there are multiple values in a column, its outputs are normalized

so that they sum up to 1. Mathematically, for a matrix $\boldsymbol{A} \in \mathbb{R}^{a \times b}$, $\sigma_H[\boldsymbol{A}] \in \mathbb{R}^{a \times b}$ is calculated as follows.

$$\sigma_H[\boldsymbol{A}]_{i,j} := \begin{cases} 1/|I_j| & \text{if } A_{i,j} = \max_k A_{k,j}, \\ 0 & \text{otherwise,} \end{cases} \tag{1}$$

where $I_j := \arg\max_k A_{k,j} := \{k' \in [a] \mid A_{k',j} = \max_k A_{k,j}\}$ for any $j \in [b]$. We use $\sigma_R$ to denote the ReLU activation function, that is, $\sigma_R[x] := \max(0, x)$. Unlike $\sigma_S$ and $\sigma_H$, $\sigma_R$ is always applied element-wise, regardless of whether the input is a vector or a matrix. For any natural number $x \in \mathbb{N}$, $\text{BIN}_{i:j}(x) \in \mathbb{N}$ represents the sequence of bits from the $i$-th bit to the $j$-th bit (counting from the left) of $x$, interpreted as a natural number. For a vector $\boldsymbol{v} \in \mathbb{R}^a$, the $L^2$ norm of $\boldsymbol{v}$ is denoted by $\|\boldsymbol{v}\|_2 := \sum_{i=1}^a v_i^2$. We use standard asymptotic notation. Specifically, $f(n) = O(g(n))$ indicates that the function $f$ grows *at most* as fast as $g$ for sufficiently large $n$, and $f(n) = \tilde{O}(g(n))$ represents that $f$ grows at most as fast as $g$, up to logarithmic factors. Likewise, $f(n) = \Omega(g(n))$ means that the function $f$ grows *at least* as fast as $g$ for sufficiently large $n$. $f(n) \lesssim g(n)$ means that there exists a positive constant $c$ such that $f(n) \leq cg(n)$ holds.

In this paper, we basically use $n$ to denote the length of an input sequence, $N$ to denote the number of input sequences, $C$ to denote the number of classes, and $d$ to denote the dimensionality of each token. Additionally, index $i$ is typically used to refer to the position of input sequences, while index $k$ is used to refer to the position of the token within an input sequence.

## 3.2 TRANSFORMER BLOCK

In this subsection, we introduce the architecture of Transformers (Vaswani et al., 2017). We basically follow the notations by Kim et al. (2023). Transformers are defined by stacking multiple Transformer blocks, each of which consists of a self-attention layer and a feed-forward layer.

**Self-attention layer**: Given an input sequence $\boldsymbol{Z} \in \mathbb{R}^{m \times n}$, the output of a self-attention layer $\mathcal{F}_l^{(\text{SA})} : \mathbb{R}^{m \times n} \to \mathbb{R}^{m \times n}$ at block $l \in [L]$ is calculated by

$$\mathcal{F}_l^{(\text{SA})}(\boldsymbol{Z}) := \boldsymbol{Z} + \sum_{h=1}^H \boldsymbol{W}_{hl}^{(O)} \boldsymbol{W}_{hl}^{(V)} \boldsymbol{Z} \sigma_S \left[ \left( \boldsymbol{W}_{hl}^{(K)} \boldsymbol{Z} \right)^\top \left( \boldsymbol{W}_{hl}^{(Q)} \boldsymbol{Z} \right) \right] \in \mathbb{R}^{m \times n}, \tag{2}$$

where $\boldsymbol{W}_{hl}^{(V)}, \boldsymbol{W}_{hl}^{(K)}, \boldsymbol{W}_{hl}^{(Q)} \in \mathbb{R}^{s \times m}$ and $\boldsymbol{W}_{hl}^{(O)} \in \mathbb{R}^{m \times s}$ are value, key, query and projection matrices at head $h \in [H]$ with head size $s$, respectively.

**Feed-forward layer**: The output $\boldsymbol{H} \in \mathbb{R}^{m \times n}$ of the self-attention layer at block $l$ is then passed to the feed-forward layer, which performs the following token-wise operation:

$$\mathcal{F}_l^{(\text{FF})}(\boldsymbol{H})_{:,k} := \boldsymbol{H}_{:,k} + \boldsymbol{W}_l^{(2)} \sigma_R \left[ \boldsymbol{W}_l^{(1)} \boldsymbol{H}_{:,k} + \boldsymbol{b}_l^{(1)} \right] + \boldsymbol{b}_l^{(2)} \in \mathbb{R}^m \quad (k \in [n]), \tag{3}$$

where $\boldsymbol{W}_l^{(1)} \in \mathbb{R}^{q \times m}$ and $\boldsymbol{W}_l^{(2)} \in \mathbb{R}^{m \times q}$ are weight matrices with hidden dimension $q$, and $\boldsymbol{b}_l^{(1)} \in \mathbb{R}^q$ and $\boldsymbol{b}_l^{(2)} \in \mathbb{R}^m$ are bias terms.

Using the self-attention layer and the feed-forward layer, the Transformer block $\mathcal{F}_l : \mathbb{R}^{m \times n} \to \mathbb{R}^{m \times n}$ at block $l \in [L]$ is defined as a composition of these two layers, that is, $\mathcal{F}_l := \mathcal{F}_l^{(\text{FF})} \circ \mathcal{F}_l^{(\text{SA})}$, and the whole architecture of the Transformer $\mathcal{N} : \mathbb{R}^{d \times n} \to \mathbb{R}^{1 \times n}$ is expressed by

$$\mathcal{N} := \mathcal{E}_{\text{out}} \circ \mathcal{F}_L \circ \cdots \circ \mathcal{F}_1 \circ \mathcal{E}_{\text{in}}, \tag{4}$$

where $\mathcal{E}_{\text{in}} : \mathbb{R}^{d \times n} \to \mathbb{R}^{m \times n}$ and $\mathcal{E}_{\text{out}} : \mathbb{R}^{m \times n} \to \mathbb{R}^{1 \times n}$ are token-wise linear mappings.

In a Transformer, the width is determined by the combination of self-attention layers and feed-forward layers. According to the definition proposed by Kim et al. (2023), the **width** of the Transformer model is defined as $\max(m, sH, q)$. We define the **depth** of a Transformer by the number of blocks $L$.

**Remark 3.1.** *The use of in/out token-wise linear mappings comes from the fact that Transformer blocks by definition have the same input and output dimensions. The token-wise linear mappings can be removed at the cost of a linear dependence of the number of parameters required for memorization on the embedding dimension $d$.*

## 3.3 BIT COMPLEXITY

In this paper, we consider not only the number of parameters but also the number of bits required to represent the model. Specifically, we adopt the definition of **bit complexity** proposed by Vardi et al. (2022). According to this definition, the bit complexity of a parameter is defined as the number of bits needed to represent that parameter. The bit complexity of a model is then defined as the maximum bit complexity among its individual parameters. It is important to note that by multiplying the bit complexity of the model by the number of parameters, we can estimate the total number of bits required to represent the entire model.

## 4 MEMORIZATION CAPACITY OF TRANSFORMERS

In this section, we state the main theorems of this paper regarding the optimal memorization capacity of Transformers. Section 4.1 defines the memorization capacity of Transformers and discuss the main challenge behind this concept. In Sections 4.2 and 4.3, we provide upper and lower bounds on the number of parameters required for Transformers to achieve memorization in the next-token prediction setting and the sequence-to-sequence prediction setting, respectively.

### 4.1 PROBLEM SETTING

The aim of this study is to analyze the memorization capacity of Transformers. Informally, memorization capacity refers to the minimum size of a model that can memorize a specific number of arbitrary data points. To be more precise, let $\mathcal{X}$ and $\mathcal{Y}$ be input space and output space, respectively. Then, given $N$ input-label pairs $(\boldsymbol{X}^{(1)}, y^{(1)}), \ldots, (\boldsymbol{X}^{(N)}, y^{(N)}) \in \mathcal{X} \times \mathcal{Y}$, we are interested in the model complexity of a model $f : \mathcal{X} \to \mathcal{Y}$ such that $f(\boldsymbol{X}^{(i)}) = y^{(i)}$ holds for any $i \in [N]$. In the case of Transformers, the input space $\mathcal{X}$ consists of input sequences made up of $n$ tokens, each of which is a $d$-dimensional vector. Hence, we define the input space $\mathcal{X}$ as $\mathcal{X} := \mathbb{R}^{d \times n}$.

Without any assumptions on the input data, it has been shown by Sontag (1997), that a linear order of parameters is required to memorize arbitrary $N$ data points. To achieve a sub-linear memorization capacity, in this paper, we assume that the data points are well separated, a common assumption in prior work (Park et al., 2021; Vardi et al., 2022; Kim et al., 2023; Kajitsuka & Sato, 2023; Siegel, 2024). In the case of Transformers, this concept is formalized as token-wise $(r, \delta)$-separatedness (Kim et al., 2023; Kajitsuka & Sato, 2023).

**Assumption 4.1** (Token-wise separatedness). *Let $\boldsymbol{X}^{(1)}, \ldots, \boldsymbol{X}^{(N)} \in \mathbb{R}^{d \times n}$ be $N$ input sequences, each of which consists of $n$ word vectors with its dimension $d$. Then, we say that $\boldsymbol{X}^{(1)}, \ldots, \boldsymbol{X}^{(N)}$ are **token-wise** $(\mathbf{r}, \boldsymbol{\delta})$-**separated** for some $r, \delta > 0$ if the following two conditions are satisfied:*

1. *for every $i \in [N]$ and $k \in [n]$, $\|\boldsymbol{X}^{(i)}_{:,k}\|_2 \leq r$ holds.*

2. *for every $i, j \in [N]$ and $k, l \in [n]$, either $\boldsymbol{X}^{(i)}_{:,k} = \boldsymbol{X}^{(j)}_{:,l}$ or $\|\boldsymbol{X}^{(i)}_{:,k} - \boldsymbol{X}^{(j)}_{:,l}\|_2 \geq \delta$ holds.*

The notion of token-wise $(r, \delta)$-separatedness ensures that the word vectors appearing in the input sequences have an $L^2$ norm of at most $r$, and are separated by at least $\delta$ in $L^2$ norm from each other.

The main difficulty of memorization with Transformers, compared with feed-forward networks, lies in the fact that tokens with identical values do not necessarily correspond to the same label. Instead, it is crucial to capture the context in which each token appears within the entire input sequence. In Transformers, while feed-forward layers operate on individual tokens, self-attention layers are the only place that enables interactions between tokens within the input sequence. Therefore, the central question we consider in this paper is:

*how efficiently can self-attention layers capture the context of tokens?*

To explore this issue, we analyze both upper and lower bounds on the number of parameters required for memorization with Transformers in two settings: next-token prediction and sequence-to-sequence prediction.

## 4.2 NEXT-TOKEN PREDICTION SETTING

### 4.2.1 UPPER BOUND

First, given $N$ input sequences of length $n$, consider the problem setting in which a Transformer memorizes labels corresponding to the $n$-th token of all input sequences. We call this task **next-token prediction** setting. In this problem setting, how many parameters does a Transformer architecture require? Surprisingly, $\tilde{O}(\sqrt{N})$ is sufficient, that is, *the input length $n$ has almost no effect on the number of parameters required for memorization*, as the following theorem states.

In the next theorem, $\mathcal{F}_1^{(\mathrm{FF})}$ and $\mathcal{F}_2^{(\mathrm{FF})}$ represent feed-forward networks of arbitrary depth, unlike eq. (3), which is limited to two layers. Note that deep feed-forward networks can also be implemented with standard Transformers, by setting the projection matrix of the self-attention layer in each block to zero. Furthermore, the assumption of consistency on labels in Theorem 4.1 is a necessary requirement to perform memorization with a Transformer, due to its permutation equivariance.

**Theorem 4.1** (Next-token prediction). *Let* $(\boldsymbol{X}^{(1)}, y^{(1)}), \ldots, (\boldsymbol{X}^{(N)}, y^{(N)}) \in \mathbb{R}^{d \times n} \times [C]$ *be a sequence of input-label pairs such that*

*1.* $(\boldsymbol{X}^{(1)}, y^{(1)}), \ldots, (\boldsymbol{X}^{(N)}, y^{(N)})$ *are consistently labeled, in the sense that for any* $i, j \in [N]$, *we have* $y^{(i)} = y^{(j)}$ *if*

$$\boldsymbol{X}_{:,n}^{(i)} = \boldsymbol{X}_{:,n}^{(j)} \quad and \quad \boldsymbol{X}^{(i)} = \boldsymbol{X}^{(j)} \text{ up to permutations.} \tag{5}$$

*2.* $\boldsymbol{X}^{(1)}, \ldots, \boldsymbol{X}^{(N)}$ *are token-wise* $(r, \delta)$*-separated for some* $r \geq 1$ *and* $0 < \delta \leq 1$.

*Then, there exists a Transformer* $\mathcal{N} : \mathbb{R}^{d \times n} \to \mathbb{R}^n$ *with width* 14 *and depth* $\tilde{O}(\sqrt{N})$ *that memorizes the dataset, that is,*

$$\mathcal{N}\left(\boldsymbol{X}^{(i)}\right)_n = \mathcal{E}_{\mathrm{out}} \circ \mathcal{F}_2^{(\mathrm{FF})} \circ \mathcal{F}^{(\mathrm{SA})} \circ \mathcal{F}_1^{(\mathrm{FF})} \circ \mathcal{E}_{\mathrm{in}}\left(\boldsymbol{X}^{(i)}\right)_n = y^{(i)} \tag{6}$$

*holds for every* $i \in [N]$, *as long as* $n, C, r\delta^{-1} = N^{O(1)}$ *as* $N \to \infty$.

The formal statement of Theorem 4.1 and its proof can be found in Appendix B.1.1.

**Remark 4.1** (Deep sets). *In fact, Theorem 4.1 can be extended to deep sets (Zaheer et al., 2017), which is a popular architecture to model a mapping from sets to labels. For details on this result, see Appendix D.*

**Remark 4.2** (Embedding layer). *A similar result holds for a Transformer with an embedding layer. However, in this case, the presence of an embedding layer introduces a dependency on the size of the vocabulary, which may result in a non-optimal order of parameters in the worst-case scenario. Details regarding this discussion can be found in Appendix E.*

**Remark 4.3** (Dependence on $d$). *The Transformer architecture defined by eq. (4) includes token-wise linear mappings* $\mathcal{E}_{\mathrm{in}} : \mathbb{R}^d \to \mathbb{R}^m$ *and* $\mathcal{E}_{\mathrm{out}} : \mathbb{R}^m \to \mathbb{R}^d$, *leading to* $\tilde{O}(d + \sqrt{N})$ *parameters for a Transformer with depth* $\tilde{O}(\sqrt{N})$ *and width* 14. *As noted by Vardi et al. (2022) and Kim et al. (2023), this dependence on the dimension $d$ is unavoidable to preserve the information of the input tokens.*

Theorem 4.1 demonstrates that as long as the dimension $d$ is of the order $d = \tilde{O}(\sqrt{N})$, the Transformer with a single self-attention layer can memorize $N$ input sequences and their labels for next-token prediction with $\tilde{O}(\sqrt{N})$ parameters, showing negligible dependence on the input length $n$. In contrast, to accomplish the same task with a feed-forward network, it is necessary to use $d \times n$ parameters to retain the information of the input sequence in $\mathbb{R}^{d \times n}$. This illustrates a significant efficiency advantage of Transformers over feed-forward networks, thanks to parameter sharing.

### 4.2.2 PROOF OUTLINE OF THEOREM 4.1

Here we provide an outline of the proof of Theorem 4.1. See Appendix B.1.1 for its full proof.

The proof strategy is to construct a contextual mapping as in Yun et al. (2019), Kim et al. (2023) and Kajitsuka & Sato (2023), and then construct a mapping from the context id to the label. Here, a

contextual mapping is a function used to distinguish tokens in each input sequence with the following properties:

**Definition 4.1** (Contextual mapping). Let $\boldsymbol{X}^{(1)}, \ldots, \boldsymbol{X}^{(N)} \in \mathbb{R}^{d \times n}$ be input sequences. Then, a map $\mathcal{CM} : \mathbb{R}^{d \times n} \to \mathbb{R}^n$ is called an $(\mathbf{r}, \boldsymbol{\delta})$-**contextual mapping** if the following two conditions hold:

1. For any $i \in [N]$ and $k \in [n]$, $\left| \mathcal{CM}(\boldsymbol{X}^{(i)})_k \right| \leq r$ holds.

2. For any $i, j \in [N]$ and $k, l \in [n]$ such that $\boldsymbol{X}_{:,k}^{(i)} \neq \boldsymbol{X}_{:,l}^{(j)}$ or $\boldsymbol{X}^{(i)} \neq \boldsymbol{X}^{(j)}$ up to permutations, $\left| \mathcal{CM}(\boldsymbol{X}^{(i)})_k - \mathcal{CM}(\boldsymbol{X}^{(j)})_l \right| \geq \delta$ holds.

In particular, $\mathcal{CM}(\boldsymbol{X}^{(i)})_k$ is called a **context id** of the $k$-th token in $\boldsymbol{X}^{(i)}$.

Intuitively, the two conditions above ensure that the contextual mapping is injective from "distinct" data points to scalars. If such a mapping can be constructed, then a mapping from context ids to labels can be realized using a feed-forward network with $\tilde{O}(\sqrt{N})$ parameters, as shown by Vardi et al. (2022). In particular, if we can associate each distinct input sequence with a unique value, referred to as a **sequence id**, then the context id of, for example, the $k$-th token in $\boldsymbol{X}^{(i)}$ can be constructed from the sequence id of $\boldsymbol{X}^{(i)}$ and the token vector $\boldsymbol{X}_{:,k}^{(i)}$. Therefore, the primary focus of our proof is on how to construct a mapping from each input sequence to its sequence id using a feed-forward network and a single self-attention layer.

From a high-level perspective, our goal is to construct a feed-forward network $\phi : \mathbb{R}^d \to \mathbb{R}$ with $\tilde{O}(\sqrt{N})$ parameters such that the sums

$$\sum_{k=1}^{n} \phi(\boldsymbol{X}_{:,k}^{(1)}), \ldots, \sum_{k=1}^{n} \phi(\boldsymbol{X}_{:,k}^{(N)}) \tag{7}$$

are well-separated [1]. The sum $\sum_{k=1}^{n} \phi(\boldsymbol{X}_{:,k}^{(i)})$ $(i \in [N])$ is then used as the sequence id of $\boldsymbol{X}^{(i)}$.

Crucial observations for constructing $\phi$ with $\tilde{O}(\sqrt{N})$ parameters are as follows.

1. To distinguish $N$ input sequences, it is sufficient to focus on at most $N$ distinct word vectors. More precisely, given $N$ input sequences, there are at most $N$ distinct word vectors such that the input sequences can be identified by counting occurrences of these $N$ words $A = \{\boldsymbol{v}_1, \ldots, \boldsymbol{v}_N\} \subset \mathbb{R}^d$ (Lemma B.1).

2. Although a feed-forward network requires $\Omega(\sqrt{N})$ parameters to memorize $N$ data points and their labels (Goldberg & Jerrum, 1995), a network that outputs *zero* for additional data points not among $N$ data points can be constructed without significantly affecting the order of the parameter count (Lemma C.1). Together with the first observation, all we need is to construct a feed-forward network $\phi : \mathbb{R}^d \to \mathbb{R}$ such that

$$\sum_{k=1, \boldsymbol{X}_{:,k}^{(1)} \in A}^{n} \phi(\boldsymbol{X}_{:,k}^{(1)}), \ldots, \sum_{k=1, \boldsymbol{X}_{:,k}^{(N)} \in A}^{n} \phi(\boldsymbol{X}_{:,k}^{(N)}) \tag{8}$$

are well-separated.

3. The final key observation is that rather than directly constructing $\phi$, we first consider the high-dimensional representation. Concretely, given arbitrary bijection $g : A \to [N]$, we can map each input sequence $\boldsymbol{X}^{(i)}$ to a high-dimensional vector $\tilde{\boldsymbol{X}}^{(i)} \in \mathbb{R}^N$ as follows:

$$\tilde{\boldsymbol{X}}^{(i)} := \sum_{k=1, \boldsymbol{X}_{:,k}^{(i)} \in A}^{n} \boldsymbol{e}_{g(\boldsymbol{X}_{:,k}^{(i)})}, \tag{9}$$

where $\boldsymbol{e}_{g(\boldsymbol{X}_{:,k}^{(i)})} \in \{0, 1\}^N$ is a one-hot vector with 1 only in the $g(\boldsymbol{X}_{:,k}^{(i)})$-th position. While $\tilde{\boldsymbol{X}}^{(1)}, \ldots, \tilde{\boldsymbol{X}}^{(N)}$ are distinct from the first observation and suitable candidates for sequence

---

[1] For simplicity, here we assume that the input sequences $\boldsymbol{X}^{(1)}, \ldots, \boldsymbol{X}^{(N)}$ are distinct up to permutations.

ids, it requires $\Omega(N)$ parameters to express these $N$-dimensional vectors with feed-forward networks. This problem can be circumvented by compressing the high-dimensional vectors into scalars using an adequate vector $\boldsymbol{v}$, and we define $\phi$ by $\phi(\boldsymbol{x}) := \boldsymbol{v}^\top \boldsymbol{e}_{g(\boldsymbol{x})}$.

To ensure that a feed-forward network with $\tilde{O}(\sqrt{N})$ parameters can indeed implement the function $\phi$, we need to carefully analyze how separated the compressed versions of the high-dimensional representations $\tilde{\boldsymbol{X}}^{(1)}, \ldots, \tilde{\boldsymbol{X}}^{(N)}$ are. Detailed proof of this implementation is provided in Lemma B.3.

### 4.2.3 LOWER BOUND

In this subsection, we evaluate the minimal model complexity required for memorization with Transformers in the next-token prediction setting to determine how close to optimal Theorem 4.1 is.

First, notice that the model obtained in Theorem 4.1 is *optimal, in terms of bit counts*.

**Remark 4.4** (Optimality in terms of bit counts). *As previously discussed in Remark 4.3, the Transformer model obtained in Theorem 4.1 has $\tilde{O}(\sqrt{N})$ parameters as long as $d = \tilde{O}(\sqrt{N})$. On the other hand, the bit complexity of the model is $\tilde{O}(\log d + \sqrt{N})$ (see the formal statement in Appendix B.1.1). Therefore, if $d = \tilde{O}(\sqrt{N})$, the total number of bits required to represent the model is $\tilde{O}(N)$. Given that there are $2^N$ possible label assignments for $N$ distinct data points with binary labels, $\tilde{O}(N)$ bits are optimal up to logarithmic factors for this setting. The more general case where bit complexity is restricted to $\tilde{O}(N^\epsilon)$ for some $\epsilon \in [0, 1/2]$ is discussed in Appendix F.*

Having established the optimality in terms of bit counts, we now turn to evaluating how efficient the number of parameters of the Transformer model considered in Theorem 4.1 is. The next theorem provides a lower bound on the number of parameters required for memorization in the next-token prediction setting.

**Theorem 4.2** (Lower bound). *Suppose a Transformer $\mathcal{N} : \mathbb{R}^{d \times n} \to \mathbb{R}^n$ defined by eq. (4) can shatter a set of $N$ distinct input sequences $\boldsymbol{X}^{(1)}, \ldots, \boldsymbol{X}^{(N)} \in \mathbb{R}^{d \times n}$ with $X_{:,k}^{(i)} = X_{:,1}^{(i)}$ for any $i \in [N]$ and $k \in [n]$, in the sense that for any label assignments $y^{(1)}, \ldots, y^{(N)} \in \{0, 1\}$, there are parameters with which $\mathcal{N}(\boldsymbol{X}^{(i)})_n = y^{(i)}$ holds for any $i \in [N]$. Then, the Transformer $\mathcal{N}$ has at least $\Omega(\sqrt{N})$ parameters.*

The proof of this theorem can be found in Appendix B.1.2. This result indicates that the Transformer model described in Theorem 4.1 is also *optimal in terms of the number of parameters*. Specifically, since memorization in the next-token prediction setting requires the ability to distinguish $N$ input sequences, this result provides the following crucial insight.

**A Transformer with a single layer of self-attention already possesses necessary and sufficient expressive capacity to identify input sequences.**

In fact, as indicated in the proof outline in Section 4.2.2, we only employ the self-attention layer as an averaging operation in the model obtained by Theorem 4.1. The observation that simple averaging provides sufficient representational power has been confirmed experimentally by Yu et al. (2022) with their PoolFormer architecture. In this paper, we provide theoretical support by demonstrating that a Transformer with just a simple averaging operation already has optimal memorization capacity. We also conducted experiments on two real-world datasets and a randomly generated dataset, confirming that even a single layer of self-attention, as averaging, possesses sufficient representational capacity for memorization. For further details, please refer to Appendix H.

### 4.3 SEQUENCE-TO-SEQUENCE PREDICTION SETTING

Next, we consider the problem setting in which each token in an input sequence is assigned some label and a Transformer memorizes them all. We call this task a **sequence-to-sequence prediction** setting, or seq-to-seq prediction for short.

It is readily apparent that the seq-to-seq prediction can be regarded as rearranging the input sequence so that each token is placed at the end of the sequence, and then performing next-token prediction on $nN$ input sequences obtained in this way. From this observation, we have the following corollary from Theorem 4.1.

**Corollary 4.1** (Seq-to-seq prediction). *Let $(\boldsymbol{X}^{(1)}, \boldsymbol{y}^{(1)}), \ldots, (\boldsymbol{X}^{(N)}, \boldsymbol{y}^{(N)}) \in \mathbb{R}^{d \times n} \times [C]^n$ be a sequence of input-label pairs such that*

1. *$(\boldsymbol{X}^{(1)}, \boldsymbol{y}^{(1)}), \ldots, (\boldsymbol{X}^{(N)}, \boldsymbol{y}^{(N)})$ are consistently labeled, in the sense that for any $i, j \in [N]$ and $k, l \in [n]$, we have $y_k^{(i)} = y_l^{(j)}$ if*

$$\boldsymbol{X}_{:,k}^{(i)} = \boldsymbol{X}_{:,l}^{(j)} \quad and \quad \boldsymbol{X}^{(i)} = \boldsymbol{X}^{(j)} \text{ up to permutations.} \tag{10}$$

2. *$\boldsymbol{X}^{(1)}, \ldots, \boldsymbol{X}^{(N)}$ are token-wise $(r, \delta)$-separated for some $r \geq 1$ and $0 < \delta \leq 1$.*

*Then, there exists a Transformer $\mathcal{N} : \mathbb{R}^{d \times n} \to \mathbb{R}^n$ with width $14$ and depth $\tilde{O}(\sqrt{nN})$ that memorizes the dataset, that is,*

$$\mathcal{N} \left( \boldsymbol{X}^{(i)} \right)_k = \mathcal{E}_{\text{out}} \circ \mathcal{F}_2^{(\text{FF})} \circ \mathcal{F}^{(\text{SA})} \circ \mathcal{F}_1^{(\text{FF})} \circ \mathcal{E}_{\text{in}} \left( \boldsymbol{X}^{(i)} \right)_k = y_k^{(i)} \tag{11}$$

*holds for every $i \in [N]$ and $k \in [n]$, as long as $C, r\delta^{-1} = (nN)^{O(1)}$ as $nN \to \infty$.*

**Remark 4.5** (Sparse Transformers). *While Corollary 4.1 demonstrates that a Transformer with a single-layer self-attention can achieve memorization in the seq-to-seq prediction setting, it inevitably requires $O(n^2)$ computational complexity due to the self-attention mechanism. In line with recent efforts to improve the scalability of Transformers by making attention maps sparse (Zaheer et al., 2020; Yun et al., 2020), using two self-attention layers and appending an additional token to the input sequence allows us to achieve the same behavior with an $O(n)$ connections without affecting the order of parameter counts. This idea of aggregating global information into the additional token has gained interest in recent studies (Darcet et al., 2023; Wang et al., 2024a).*

This corollary shows that at least $\tilde{O}(\sqrt{nN})$ parameters with bit complexity $\tilde{O}(\sqrt{nN})$ are enough to memorize $N$ input sequences of input length $n$. The next question is: is this order optimal for the seq-to-seq prediction setting? As in the case of next-token prediction setting (Remark 4.4), we can leverage a similar argument to show that this is optimal, at least in terms of bit counts.

**Remark 4.6** (Optimality in terms of bit counts). *If $d = \tilde{O}(\sqrt{nN})$, the construction by Corollary 4.1 uses $\tilde{O}(\sqrt{nN})$ parameters with bit complexity $\tilde{O}(\sqrt{nN})$ to memorize $N$ input sequences of input length $n$, which amounts to $\tilde{O}(nN)$ bits. If all word vectors in input sequences are different, there are $2^{nN}$ binary label patterns. Therefore, to memorize such patterns, the number of states of the model must be at least $2^{nN}$, which means that $\log 2^{nN} = nN$ bits are required.*

Unlike the next-token prediction setting, it is challenging to analyze the optimal lower bound on the number of parameters necessary to memorize $N$ input sequences with input length $n$ for the seq-to-seq prediction setting, mainly due to the presence of the softmax function. However, we partially answer this question by considering a Transformer that uses not the softmax function, but instead the *hardmax* function, often viewed as an approximation of the softmax.

More rigorously, we introduce the following self-attention layer with the hardmax function, which we call the **hard attention** layer. For each block $l \in [L]$ and its input $\boldsymbol{Z} \in \mathbb{R}^{m \times n}$, the hard attention layer at block $l$ calculates

$$\mathcal{F}_l^{(\text{HA})} (\boldsymbol{Z}) := \boldsymbol{Z} + \sum_{h=1}^H \boldsymbol{W}_{hl}^{(O)} \boldsymbol{W}_{hl}^{(V)} \boldsymbol{Z} \sigma_H \left[ \left( \boldsymbol{W}_{hl}^{(K)} \boldsymbol{Z} \right)^\top \left( \boldsymbol{W}_{hl}^{(Q)} \boldsymbol{Z} \right) \right] \in \mathbb{R}^{m \times n}, \tag{12}$$

where $\sigma_H : \mathbb{R}^{n \times n} \to [0, 1]^{n \times n}$ is the column-wise hardmax function (see eq. (1) for its definition), and $\boldsymbol{W}_{hl}^{(V)}$, $\boldsymbol{W}_{hl}^{(K)}$, $\boldsymbol{W}_{hl}^{(Q)} \in \mathbb{R}^{s \times m}$ and $\boldsymbol{W}_{hl}^{(O)} \in \mathbb{R}^{m \times s}$ are value, key, query and projection matrices at head $h \in [H]$ with head size $s$, respectively. It is worth noting that a simple averaging operation can also be implemented using a hard attention layer by setting key and query matrices to zero.

With this definition, we demonstrate that the number of parameters by Corollary 4.1 is actually optimal up to logarithmic factors, at least for Transformers with the hardmax function. To state the theorem, let $W$ be the number of parameters and $\boldsymbol{\theta} \in \mathbb{R}^W$ be a vector of all parameters of the Transformer. We also denote by $\mathcal{N}_{\boldsymbol{\theta}}$ the Transformer to emphasize the presence of the parameter vector $\boldsymbol{\theta}$.

**Theorem 4.3** (Lower bound). *Let $\mathcal{N}_{\boldsymbol{\theta}} : \mathbb{R}^{d \times n} \to \mathbb{R}^n$ be a Transformer defined by eq. (4) with self-attention layers replaced with hard attention layers (eq. (12)). In addition, suppose $\mathcal{N}_{\boldsymbol{\theta}}$ can shatter a set of $N$ input sequences $\boldsymbol{X}^{(1)}, \ldots, \boldsymbol{X}^{(N)} \in \mathbb{R}^{d \times n}$ with $X_{:,k}^{(i)} \neq X_{:,l}^{(j)}$ for any $i, j \in [N]$ and $k, l \in [n]$ ($i \neq j$ or $k \neq l$), in the sense that for any label assignments $\boldsymbol{y}^{(1)}, \ldots, \boldsymbol{y}^{(N)} \in \{0,1\}^n$, there is a parameter vector $\boldsymbol{\theta} \in \mathbb{R}^W$ such that*

$$\mathcal{N}_{\boldsymbol{\theta}}(\boldsymbol{X}^{(i)}) = \boldsymbol{y}^{(i)} \tag{13}$$

*for any $i \in [N]$. Then, the Transformer has at least $W = \Omega\left(\sqrt{\frac{nN}{\log(nN)}}\right)$ parameters.*

The proof of Theorem 4.3 builds on the approach used by Bartlett et al. (2019) to evaluate a lower bound on the VC dimension of feed-forward networks. Specifically, considering a Transformer as a function in variable its parameter vector, we partition the parameter space of the Transformer in such a way that, within each cell of this partition, the function can be expressed as a polynomial in terms of its parameters, and then evaluate the number of cells and the properties of the polynomials within those cells.

The key novelty of the proof lies in the analysis of how parameter sharing and the hardmax function affect the memorization capacity of Transformers. Parameter sharing in Transformers allows the model to effectively behave like a network with its width scaled by the number of tokens, without actually increasing the number of parameters. However, the proof shows that merely increasing the width by a factor of $n$ does not lead to a fundamental improvement in the memorization capacity of the Transformer. The full proof of Theorem 4.3 can be found in Appendix B.2.

Theorem 4.3 demonstrates that the number of parameters in the model from Corollary 4.1 is within logarithmic factors of the optimal lower bound. In addition, it provides another crucial insight. As shown in the next-token prediction setting, Transformers can identify $N$ input sequences with $\tilde{O}(\sqrt{N})$ parameters and single self-attention layer, which implies that they are capable of capturing the context of each token. In contrast, the memorization capacity in the seq-to-seq setting is provably lower-bounded by $\tilde{\Omega}(\sqrt{nN})$, which includes an additional $\sqrt{n}$ factor compared to the $\tilde{O}(\sqrt{N})$ bound in the next-token prediction setting. Therefore, in the seq-to-seq prediction setting, *the primary bottleneck is not the contextual mapping of tokens, but rather the feed-forward layers' capacity to map this token-level contextual information to labels.*

We conclude this section by leaving an open problem. Based on Theorem 4.3, for a Transformer to memorize $N$ sequences of length $n$ with $o(\sqrt{nN})$ parameters, it is necessary to exploit the unique characteristics of the softmax function, rather than using it as an approximation of hardmax.

**Open Problem.** *Does a Transformer using the softmax function require $\Omega(\sqrt{nN})$ parameters to memorize $N$ input-label pairs $(\boldsymbol{X}^{(1)}, \boldsymbol{y}^{(1)}), \ldots, (\boldsymbol{X}^{(N)}, \boldsymbol{y}^{(N)}) \in \mathbb{R}^{d \times n} \times [C]^n$? Alternatively, is it possible to construct a Transformer with $o(\sqrt{nN})$ parameters that can shatter arbitrary $N$ token-wise $(r, \delta)$-separated input sequences in the seq-to-seq setting?*

## 5 CONCLUSIONS

In this paper, we showed that in the next-token prediction setting, a Transformer with $\tilde{O}(\sqrt{N})$ parameters can memorize $N$ input sequences of length $n$ and their labels, which we showed to be optimal up to logarithmic factors. This result indicates that Transformers can perform next-token prediction with almost no impact from the length of the input sequence. Notably, its proof indicates that even a single self-attention layer used as an averaging operation possesses sufficient expressive power to distinguish between input sequences efficiently. Furthermore, we demonstrated that in the seq-to-seq prediction setting, $\tilde{O}(\sqrt{nN})$ parameters are also sufficient, and we proved that this is optimal up to logarithmic factors, at least for Transformers using hardmax. These findings highlight that the main bottleneck in seq-to-seq prediction tasks lies in the feed-forward layers' capacity to map each token to the corresponding label.

Given that a single layer of self-attention as an averaging operation suffices for distinguishing input sequences from a memorization perspective, our results suggest that the advantages of using self-attention might rather lie in the perspectives of optimization and generalization.

ACKNOWLEDGMENTS

This work was supported by JSPS KAKENHI Grant Number JP24H00709. We would like to thank all the collaborators and anonymous reviewers for constructive discussions.

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

## Notation Table

**Numbers and Arrays**

$a$      A scalar

$\boldsymbol{a}$      A vector

$\boldsymbol{A}$      A matrix

$n$      The length of an input sequence

$N$      The number of input sequences

$C$      The number of output classes

$d$      Embedding dimension

$\boldsymbol{X}^{(i)}$      $i$-th input sequence, consisting of $n$ tokens of embedding dimension $d$

**Sets**

$\{\{\dots\}\}$ Multiset (see Definition A.1)

$\mathbb{R}$       Set of real numbers

$\mathbb{N}^{\mathcal{X}}$      Set of all multisets over the domain $\mathcal{X}$

$[m]$      Set of all integers from 1 to $m$

**Indexing**

$a_i$      Element $i$ of vector $\boldsymbol{a}$, with indexing starting at 1

$A_{i,j}$      Element $i, j$ of matrix $\boldsymbol{A}$

$\boldsymbol{A}_{:,i}$      Column $i$ of matrix $\boldsymbol{A}$

$\boldsymbol{A}_{i,:}$      Row $i$ of matrix $\boldsymbol{A}$

**Functions**

$\|\boldsymbol{x}\|_2$      $L^2$ norm of $\boldsymbol{x}$

$\mathrm{supp}(m)$ Support of $m$ (see Definition A.2)

$\mathrm{BIN}_{i:j}(x)$ The sequence of bits from the $i$-th bit to the $j$-th bit (counting from the left) of $x$

$\sigma_S$      Softmax function

$\sigma_H$      Hardmax function

$\sigma_R$      ReLU activation function

$\mathcal{F}^{(HA)}$ Hardmax-based self-attention mechanism with a skip-connection

$\mathcal{F}^{(SA)}$ Softmax-based self-attention mechanism with a skip-connection

$\mathcal{F}^{(FF)}$ Feed-forward neural network with a skip-connection

$\mathcal{N}_{\boldsymbol{\theta}}$      Transformer with a parameter vector $\boldsymbol{\theta}$

**Asymptotics**

$f(n) = O(g(n))$   $f$ grows at most as fast as $g$ for sufficiently large $n$

$f(n) = \tilde{O}(g(n))$   $f$ grows at most as fast as $g$ for sufficiently large $n$, up to logarithmic factors

$f(n) = \Omega(g(n))$   $f$ grows at least as fast as $g$ for sufficiently large $n$

$f \lesssim g$   There exists a positive constant $c$ such that $f \leq cg$ holds

## A    DEFINITION OF MULTISETS

A multiset is a generalization of a set whose elements are allowed to be duplicated.

**Definition A.1** (Multiset). A **multiset** over the domain $\mathcal{X}$ is identified by a function $m : \mathcal{X} \to \mathbb{N}$, which indicates the multiplicity $m(x)$ of each element $x \in \mathcal{X}$ in the multiset. The set of all multisets over the domain $\mathcal{X}$ is denoted by $\mathbb{N}^{\mathcal{X}}$.

**Definition A.2.** The **support** of a multiset $m \in \mathbb{N}^{\mathcal{X}}$ is defined by $\mathrm{supp}(m) = \{x \in \mathcal{X} \mid m(x) > 0\}$.

In addition, the **cardinality** of a multiset $m \in \mathbb{N}^{\mathcal{X}}$ is defined by

$$|m| := \begin{cases} \sum_{x \in \mathrm{supp}(m)} m(x) & \text{if } |\mathrm{supp}(m)| < \infty, \\ \infty & \text{otherwise,} \end{cases} \tag{14}$$

and the multiset $m$ is called **finite** if $|m| < \infty$.

In this paper, we only consider finite multisets, and in an abuse of notation we sometimes denote a fintie multiset $m \in \mathbb{N}^{\mathcal{X}}$ by $\{\{x_1, \dots, x_{|m|}\}\} \in \mathbb{N}^{\mathcal{X}}$, where $x_1, \dots, x_{|m|} \in \mathcal{X}$ are possibly duplicated elements.

The following assumption guarantees that each value of the multiset is separated by a certain amount, and the token-wise separatedness in the analysis of Transformer's memorization can be translated into this assumption.

**Assumption A.1** (Element-wise separatedness). *Let $\mathcal{X} := \mathbb{R}^d$ and $m^{(1)}, \ldots, m^{(N)} \in \mathbb{N}^{\mathcal{X}}$ be a sequence of finite multisets with $m^{(i)} = \{\{x_1^{(i)}, \ldots, x_{|m^{(i)}|}^{(i)}\}\}$ for each $i \in [N]$. Then, we say that $m^{(1)}, \ldots, m^{(N)}$ are **element-wise $(\mathbf{r}, \boldsymbol{\delta})$-separated** for some $r, \delta > 0$ if the following two conditions are satisfied:*

1. *for every $i \in [N]$ and $k \in [|m^{(i)}|]$, $\|x_k^{(i)}\|_2 \le r$ holds.*

2. *for every $i, j \in [N]$ and $k \in [|m^{(i)}|], l \in [|m^{(j)}|]$, either $x_k^{(i)} = x_l^{(j)}$ or $\|x_k^{(i)} - x_l^{(j)}\|_2 \ge \delta$ holds.*

# B PROOF OF MAIN RESULTS

In the following, we will extensively use the concept of multisets. For the definition of multisets and the notation used in this paper, refer to Appendix A. We also use the operator $\lesssim$ frequently. See Section 3.1 for its definition.

To prove Theorem 4.1, we present several lemmas. Lemma B.1 shows that to distinguish between $N$ distinct multisets, it suffices to focus on the occurrence counts of at most $N$ values. Lemma B.2 establishes the existence of a function that computes sequence ids used to distinguish between $N$ different multisets. Finally, Lemma B.3 states that the function obtained from Lemma B.2 with Lemma B.1 can be implemented using a feed-forward network with $\tilde{O}(\sqrt{N})$ parameters.

**Definition B.1.** Let $A \subset \mathcal{X}$ and $m \in \mathbb{N}^{\mathcal{X}}$ be a multiset. Then, we define the restriction of $m$ to $A$ by

$$m|_A(x) := \begin{cases} m(x) & \text{if } x \in A, \\ 0 & \text{otherwise.} \end{cases} \tag{15}$$

**Lemma B.1.** *Let $m^{(1)}, \ldots, m^{(N)} \in \mathbb{N}^{\mathcal{X}}$ be a sequence of distinct multisets. Then, there exists a subset $A \subset \mathcal{X}$ with its cardinality at most $N$ such that $m^{(1)}|_A, \ldots, m^{(N)}|_A$ are distinct.*

*Proof.* We prove the lemma by induction. The base case of $N = 1$ is obvious.

Suppose that the lemma is correct for the case $N = k$, and we prove the case for $N = k + 1$.

Let $m^{(1)}, \ldots, m^{(k+1)} \in \mathbb{N}^{\mathcal{X}}$ be a sequence of distinct multisets. Then, by applying the assumption to the first $k$ multisets $m^{(1)}, \ldots, m^{(k)} \in \mathbb{N}^{\mathcal{X}}$, we have a subset $A \subset \mathcal{X}$ with its cardinality at most $k$ such that $m^{(1)}|_A, \ldots, m^{(k)}|_A$ are distinct. If $m^{(1)}|_A, \ldots, m^{(k+1)}|_A$ are distinct, there is nothing to prove. So we assume that $m^{(k+1)}|_A$ coincides with $m^{(i)}|_A$ for some $i \in [k]$. Notice that for any $j \in [k]$ with $j \ne i$, $m^{(j)}|_A$ and $m^{(k+1)}|_A$ are distinct by the assumption.

Since $m^{(i)}$ and $m^{(k+1)}$ are distinct, there is an element $x \in \mathcal{X} \setminus A$ such that $m^{(i)}(x) \ne m^{(k+1)}(x)$. Then, the subset $A' \subset \mathcal{X}$ defined by $A' := A \cup \{x\}$ is the desired set for the case $N = k + 1$. $\square$

In the next lemma, we say that scalars $a_1, \ldots, a_m$ are $(r, \delta)$-separated if $|a_i| \le r$ for all $i \in [m]$ and $|a_i - a_j| \ge \delta$ for all $i, j \in [m]$ with $a_i \ne a_j$.

**Lemma B.2.** *Let $m^{(1)}, \ldots, m^{(N)} \in \mathbb{N}^{\mathcal{X}}$ be a sequence of finite and distinct multisets with $|m^{(i)}| \le M$ for every $i \in [N]$. Furthermore, let $S \subset \mathcal{X}$ be the union of all supports; that is, $S := \bigcup_{i=1}^{N} \text{supp}(m^{(i)})$.*

*Then, there exists a function $f : S \to [\lceil 4N^2 |S| \sqrt{\pi} \rceil]$ such that*

$$\sum_{x \in \text{supp}(m^{(1)})} m^{(1)}(x) f(x), \ldots, \sum_{x \in \text{supp}(m^{(N)})} m^{(N)}(x) f(x) \in \mathbb{R} \tag{16}$$

*are $(4MN^2 |S| \sqrt{\pi}, \sqrt{|S|})$-separated.*

*Proof.* Let $g : S \to [|S|] = \{1, \ldots, |S|\}$ be an arbitrary bijective function. For each multiset $m^{(i)}$ with $i = 1, \ldots, N$, we define its high-dimensional representation by

$$\tilde{\boldsymbol{m}}^{(i)} \coloneqq \sum_{x \in \mathrm{supp}(m^{(i)})} m^{(i)}(x) \boldsymbol{e}_{g(x)} \in \mathbb{N}^{|S|}, \tag{17}$$

where $\boldsymbol{e}_{g(x)} \in \{0, 1\}^{|S|}$ is a one-hot vector with 1 in the $g(x)$-th position. Since $m^{(1)}, \ldots, m^{(N)}$ are distinct with $|m^{(i)}| \leq M$ for every $i \in [N]$, we have

$$\left\| \tilde{\boldsymbol{m}}^{(i)} - \tilde{\boldsymbol{m}}^{(j)} \right\|_2^2 = \sum_{x \in S} \left( m^{(i)}(x) - m^{(j)}(x) \right)^2 \geq 1, \tag{18}$$

for any $i, j \in [N]$ with $i \neq j$. , and the norm of each $\tilde{\boldsymbol{m}}^{(i)}$ is upper-bounded by

$$\left\| \tilde{\boldsymbol{m}}^{(i)} \right\|_2 \leq \sum_{x \in \mathrm{supp}(m^{(i)})} m^{(i)}(x) \left\| \boldsymbol{e}_{g(x)} \right\|_2 = |m^{(i)}| \leq M. \tag{19}$$

By applying Lemma G.1 to $\tilde{\boldsymbol{m}}^{(1)}, \ldots, \tilde{\boldsymbol{m}}^{(N)}$, there is a unit vector $\boldsymbol{v} \in \mathbb{R}^{|S|}$ such that

$$\frac{1}{N^2} \sqrt{\frac{8}{\pi |S|}} \left\| \tilde{\boldsymbol{m}}^{(i)} - \tilde{\boldsymbol{m}}^{(j)} \right\|_2 \leq \left| \boldsymbol{v}^\top \left( \tilde{\boldsymbol{m}}^{(i)} - \tilde{\boldsymbol{m}}^{(j)} \right) \right| \leq \left\| \tilde{\boldsymbol{m}}^{(i)} - \tilde{\boldsymbol{m}}^{(j)} \right\|_2 \tag{20}$$

holds for any $i, j \in [N]$. Let $h$ be the function $h : S \to \mathbb{Z}, x \mapsto \lceil N^2 |S| \sqrt{\pi} v_{g(x)} \rceil$. Hereafter, we see that this function has the desired properties.

Let $\overline{\boldsymbol{v}} \coloneqq (\lceil N^2 |S| \sqrt{\pi} v_1 \rceil, \ldots, \lceil N^2 |S| \sqrt{\pi} v_N \rceil)^\top \in \mathbb{Z}^{|S|}$, i.e., the vector approximating $N^2 |S| \sqrt{\pi} \boldsymbol{v}$ with integers. The approximation error is estimated as follows:

$$\left\| N^2 |S| \sqrt{\pi} \boldsymbol{v} - \overline{\boldsymbol{v}} \right\|_2^2 \leq \sum_{i=1}^{|S|} \left( N^2 |S| \sqrt{\pi} v_i - \lceil N^2 |S| \sqrt{\pi} v_i \rceil \right)^2 \leq |S|, \tag{21}$$

which means that $\left\| N^2 |S| \sqrt{\pi} \boldsymbol{v} - \overline{\boldsymbol{v}} \right\|_2 \leq \sqrt{|S|}$. Notice that

$$\sum_{x \in \mathrm{supp}(m^{(i)})} m^{(i)}(x) h(x) = \sum_{x \in \mathrm{supp}(m^{(i)})} m^{(i)}(x) \cdot \lceil N^2 |S| \sqrt{\pi} v_{g(x)} \rceil$$

$$= \sum_{x \in \mathrm{supp}(m^{(i)})} m^{(i)}(x) \cdot \overline{\boldsymbol{v}}^\top \boldsymbol{e}_{g(x)}$$

$$= \overline{\boldsymbol{v}}^\top \tilde{\boldsymbol{m}}^{(i)} \tag{22}$$

holds for every $i \in [N]$. Then, the absolute value of the left-hand side is upper-bounded by

$$\left| \sum_{x \in \mathrm{supp}(m^{(i)})} m^{(i)}(x) h(x) \right| = \left| \overline{\boldsymbol{v}}^\top \tilde{\boldsymbol{m}}^{(i)} \right|$$

$$\leq \| \overline{\boldsymbol{v}} \|_2 \| \tilde{\boldsymbol{m}}^{(i)} \|_2$$

$$\leq 2 N^2 |S| \sqrt{\pi} \cdot M \tag{23}$$

since the norm of $\overline{\boldsymbol{v}}$ is upper-bounded by

$$\| \overline{\boldsymbol{v}} \|_2 \leq \left\| N^2 |S| \sqrt{\pi} \boldsymbol{v} \right\|_2 + \left\| N^2 |S| \sqrt{\pi} \boldsymbol{v} - \overline{\boldsymbol{v}} \right\|_2$$

$$\leq N^2 |S| \sqrt{\pi} + \sqrt{|S|}$$

$$\leq 2 N^2 |S| \sqrt{\pi}. \tag{24}$$

On the other hand, for any $i, j \in [N]$ with $i \neq j$, we have

$$
\begin{aligned}
& \left| \sum_{x \in \mathrm{supp}(m^{(i)})} m^{(i)}(x) h(x) - \sum_{x \in \mathrm{supp}(m^{(j)})} m^{(j)}(x) h(x) \right| \\
&= \left| \overline{\boldsymbol{v}}^\top \left( \tilde{\boldsymbol{m}}^{(i)} - \tilde{\boldsymbol{m}}^{(j)} \right) \right| \\
&\geq \left| N^2 |S| \sqrt{\pi} \boldsymbol{v}^\top \left( \tilde{\boldsymbol{m}}^{(i)} - \tilde{\boldsymbol{m}}^{(j)} \right) \right| - \left| \left( N^2 |S| \sqrt{\pi} \boldsymbol{v} - \overline{\boldsymbol{v}} \right)^\top \left( \tilde{\boldsymbol{m}}^{(i)} - \tilde{\boldsymbol{m}}^{(j)} \right) \right| \\
&\geq \left| N^2 |S| \sqrt{\pi} \boldsymbol{v}^\top \left( \tilde{\boldsymbol{m}}^{(i)} - \tilde{\boldsymbol{m}}^{(j)} \right) \right| - \left\| N^2 |S| \sqrt{\pi} \boldsymbol{v} - \overline{\boldsymbol{v}} \right\|_2 \cdot \left\| \tilde{\boldsymbol{m}}^{(i)} - \tilde{\boldsymbol{m}}^{(j)} \right\|_2 \\
&> 2\sqrt{|S|} \left\| \tilde{\boldsymbol{m}}^{(i)} - \tilde{\boldsymbol{m}}^{(j)} \right\|_2 - \sqrt{|S|} \left\| \tilde{\boldsymbol{m}}^{(i)} - \tilde{\boldsymbol{m}}^{(j)} \right\|_2 \\
&\geq \sqrt{|S|},
\end{aligned}
\tag{25}
$$

since eq. (20) implies

$$
\begin{aligned}
\left| N^2 |S| \sqrt{\pi} \boldsymbol{v}^\top \left( \tilde{\boldsymbol{m}}^{(i)} - \tilde{\boldsymbol{m}}^{(j)} \right) \right| &= N^2 |S| \sqrt{\pi} \left| \boldsymbol{v}^\top \left( \tilde{\boldsymbol{m}}^{(i)} - \tilde{\boldsymbol{m}}^{(j)} \right) \right| \\
&\geq N^2 |S| \sqrt{\pi} \cdot \frac{1}{N^2} \sqrt{\frac{8}{\pi |S|}} \left\| \tilde{\boldsymbol{m}}^{(i)} - \tilde{\boldsymbol{m}}^{(j)} \right\|_2 \\
&> 2\sqrt{|S|} \left\| \tilde{\boldsymbol{m}}^{(i)} - \tilde{\boldsymbol{m}}^{(j)} \right\|_2 .
\end{aligned}
\tag{26}
$$

Finally, the output of the function $h$ is always bounded by

$$
|h(x)| = |\lceil N^2 |S| \sqrt{\pi} v_{g(x)} \rceil| \leq N^2 |S| \sqrt{\pi} + 1 \quad (\forall x \in S).
\tag{27}
$$

Thus, by setting $f(x) := h(x) + \lfloor 2N^2 |S| \sqrt{\pi} \rfloor$, we have a desired function. $\qquad\square$

**Lemma B.3** (Separation of multisets). *Let $\mathcal{X} := \mathbb{R}^d$ and $m^{(1)}, \dots, m^{(N)} \in \mathbb{N}^{\mathcal{X}}$ be a sequence of multisets with $m^{(i)} = \{\{\boldsymbol{x}_1^{(i)}, \dots, \boldsymbol{x}_{|m^{(i)}|}^{(i)}\}\}$ for each $i \in [N]$. Suppose that $m^{(1)}, \dots, m^{(N)}$ satisfy the following three conditions:*

1. *$m^{(1)}, \dots, m^{(N)}$ are finite multisets whose cardinalities are at most $M$.*

2. *$m^{(1)}, \dots, m^{(N)}$ are distinct.*

3. *$m^{(1)}, \dots, m^{(N)}$ are element-wise $(r, \delta)$-separated for some $r \geq 1$, $0 < \delta \leq 1$.*

*Let $C_\phi := \lceil 4N^3 \sqrt{\pi} \rceil$ and $R_\phi := 20r(NM)^2 \delta^{-1} \sqrt{\pi d}$. Then, there exists a neural network $\tilde{\phi} : \mathbb{R}^d \to \mathbb{R}$ with width 12, depth*

$$
\lesssim \sqrt{N \log N} + \sqrt{\frac{N}{\log N}} \cdot \max\{\log R_\phi, \log C_\phi\},
\tag{28}
$$

*(for the definition of $\lesssim$, see Section 3.1) and bit complexity bounded by*

$$
\lesssim \log d + \sqrt{\frac{N}{\log N}} \cdot \max\{\log R_\phi, \log C_\phi\}
\tag{29}
$$

*such that $\tilde{\phi}(\boldsymbol{x}) \in [[\lceil 4N^3 \sqrt{\pi} \rceil]] \cup \{0\}$ holds for any $\boldsymbol{x} \in \bigcup_{i=1}^N \mathrm{supp}(m^{(i)})$, and*

$$
\sum_{k=1}^{|m^{(1)}|} \tilde{\phi}(\boldsymbol{x}_k^{(1)}), \dots, \sum_{k=1}^{|m^{(N)}|} \tilde{\phi}(\boldsymbol{x}_k^{(N)})
\tag{30}
$$

*are $(4MN^3 \sqrt{\pi}, 1)$-separated.*

*Proof.* By applying Lemma B.1 to the sequence of distinct multisets $m^{(1)}, \ldots, m^{(N)}$, we have a finite subset $A \subset \mathbb{R}^d$ with $|A| \leq N$ such that $m^{(1)}|_A, \ldots, m^{(N)}|_A$ are distinct. Then, according to Lemma B.2, there exists a function $f : A \to [[4N^2|A|\sqrt{\pi}]]$ such that

$$\sum_{\boldsymbol{x} \in \text{supp}(m^{(1)}|_A)} m^{(1)}|_A(\boldsymbol{x}) f(\boldsymbol{x}), \ldots, \sum_{\boldsymbol{x} \in \text{supp}(m^{(N)}|_A)} m^{(N)}|_A(\boldsymbol{x}) f(\boldsymbol{x}) \qquad (31)$$

are $(4MN^2|A|\sqrt{\pi}, \sqrt{|A|})$-separated, and in particular $(4MN^3\sqrt{\pi}, 1)$-separated.

Hereafter, we consider a function $\phi : \mathbb{R}^d \to \mathbb{R}$ such that

$$\phi(\boldsymbol{x}) := \begin{cases} f(\boldsymbol{x}) & \text{if } \boldsymbol{x} \in A, \\ 0 & \text{otherwise,} \end{cases} \qquad (32)$$

and simulate $\phi$ by a neural network. Notice that the possible number of inputs for the function $\phi$ is at most $MN$, and all outputs are natural numbers equal to or less than $\lceil 4N^2|A|\sqrt{\pi} \rceil \leq \lceil 4N^3\sqrt{\pi} \rceil$. We define constants $R_\phi$ and $C_\phi$ by

$$C_\phi := \lceil 4N^3\sqrt{\pi} \rceil, \qquad (33)$$

$$R_\phi := 20r(NM)^2 \delta^{-1} \sqrt{\pi d}. \qquad (34)$$

Then, Lemma C.1 guarantees the existence of the feed-forward network $\tilde{\phi}$ with width 12, depth

$$\lesssim \sqrt{N \log N} + \sqrt{\frac{N}{\log N}} \cdot \max\{\log R_\phi, \log C_\phi\}, \qquad (35)$$

and bit complexity bounded by

$$\lesssim \log d + \sqrt{\frac{N}{\log N}} \cdot \max\{\log R_\phi, \log C_\phi\} \qquad (36)$$

such that for any $i \in [N]$ with $m^{(i)} = \{\{\boldsymbol{x}_1^{(i)}, \ldots, \boldsymbol{x}_{|m^{(i)}|}^{(i)}\}\}$ and any $k \in [|m^{(i)}|]$, we have

$$\tilde{\phi}(\boldsymbol{x}_k^{(i)}) = \begin{cases} f(\boldsymbol{x}_k^{(i)}) & \text{if } \boldsymbol{x}_k^{(i)} \in A, \\ 0 & \text{otherwise.} \end{cases} \qquad (37)$$

Thus, the outputs of $\tilde{\phi}$ coincide with those of $\phi$ for all inputs $\boldsymbol{x}_k^{(i)}$ with $i \in [N]$ and $k \in [|m^{(i)}|]$.

Finally, we verify that the neural network $\tilde{\phi}$ actually satisfies the desired property. For any $i \in [N]$, we have

$$\sum_{k=1}^{|m^{(i)}|} \tilde{\phi}(\boldsymbol{x}_k^{(i)}) = \sum_{\boldsymbol{x} \in \text{supp}(m^{(i)})} m^{(i)}(\boldsymbol{x}) \phi(\boldsymbol{x})$$

$$= \sum_{\boldsymbol{x} \in \text{supp}(m^{(i)}) \cap A} m^{(i)}(\boldsymbol{x}) f(\boldsymbol{x})$$

$$= \sum_{\boldsymbol{x} \in \text{supp}(m^{(i)}|_A)} m^{(i)}|_A(\boldsymbol{x}) f(\boldsymbol{x}). \qquad (38)$$

Thus, eq. (31) implies that $\sum_{k=1}^{|m^{(1)}|} \tilde{\phi}(\boldsymbol{x}_k^{(1)}), \ldots, \sum_{k=1}^{|m^{(N)}|} \tilde{\phi}(\boldsymbol{x}_k^{(N)})$ are $(4MN^3\sqrt{\pi}, 1)$-separated. □

### B.1 NEXT-TOKEN PREDICTION SETTING

#### B.1.1 UPPER BOUND

Here we state the complete statement of Theorem 4.1 with its bit complexity. [2] Before moving on to the theorem, we introduce a **uniform attention** layer; that is, a self-attention layer with the

---

[2] While the upper bounds provided in Theorem B.1 is in the form $O(\sqrt{N \log N} \cdot \log n)$, these upper bounds can actually be reduced to $O(\sqrt{N \log(nN)})$ by modifying the data point partitioning in Stage II of Lemma C.1 to use $\sqrt{N \log(nN)}$ subsets, each containing $\sqrt{\frac{N}{\log(nN)}}$ elements.

softmax function replaced by simple averaging. For an input $\boldsymbol{Z} \in \mathbb{R}^{m \times n}$, the uniform attention layer calculates

$$\mathcal{F}^{(\mathrm{UA})}(\boldsymbol{Z}) := \boldsymbol{Z} + \boldsymbol{W}^{(O)} \boldsymbol{W}^{(V)} \frac{1}{n} \sum_{k=1}^{n} \boldsymbol{Z}_{:,k} \underbrace{(1, \dots, 1)}_{\in \mathbb{R}^{1 \times n}} \in \mathbb{R}^{m \times n}, \tag{39}$$

where $\boldsymbol{W}^{(V)} \in \mathbb{R}^{s \times m}$ and $\boldsymbol{W}^{(O)} \in \mathbb{R}^{m \times s}$ are value and projection matrices with head size $s$, respectively. A uniform attention layer is a subset of a self-attention layer as it can be implemented using a self-attention layer by setting key or query matrices to zero.

In the next theorem, $\mathcal{F}_1^{(\mathrm{FF})}$ and $\mathcal{F}_2^{(\mathrm{FF})}$ represent feed-forward networks of arbitrary depth, unlike eq. (3), which is limited to two layers.

**Theorem B.1** (Next-token prediction). *Let* $(\boldsymbol{X}^{(1)}, y^{(1)}), \dots, (\boldsymbol{X}^{(N)}, y^{(N)}) \in \mathbb{R}^{d \times n} \times [C]$ *be a sequence of input-label pairs such that*

1. $(\boldsymbol{X}^{(1)}, y^{(1)}), \dots, (\boldsymbol{X}^{(N)}, y^{(N)})$ *are consistently labeled, in the sense that for any* $i, j \in [N]$, *we have* $y^{(i)} = y^{(j)}$ *if*

$$\boldsymbol{X}_{:,n}^{(i)} = \boldsymbol{X}_{:,n}^{(j)} \quad \text{and} \quad \boldsymbol{X}^{(i)} = \boldsymbol{X}^{(j)} \text{ up to permutations.} \tag{40}$$

2. $\boldsymbol{X}^{(1)}, \dots, \boldsymbol{X}^{(N)}$ *are token-wise* $(r, \delta)$-*separated for some* $r \geq 1$ *and* $0 < \delta \leq 1$.

*Let* $R := 400\sqrt{3d} n^3 r N^5 \delta^{-1} \pi$. *Then, there exists a Transformer* $\mathcal{N} : \mathbb{R}^{d \times n} \to \mathbb{R}^n$ *with width 14, depth*

$$\lesssim \sqrt{N \log N} + \sqrt{\frac{N}{\log N}} \cdot \max\{\log R, \log C\}, \tag{41}$$

*(for the definition of* $\lesssim$, *see Section 3.1) and bit complexity bounded by*

$$\lesssim \log d + \sqrt{\frac{N}{\log N}} \cdot \max\{\log R, \log C\} \tag{42}$$

*that memorizes the dataset, i.e.,*

$$\mathcal{N}\left(\boldsymbol{X}^{(i)}\right)_n = \mathcal{E}_{\mathrm{out}} \circ \mathcal{F}_2^{(\mathrm{FF})} \circ \mathcal{F}^{(\mathrm{UA})} \circ \mathcal{F}_1^{(\mathrm{FF})} \circ \mathcal{E}_{\mathrm{in}}\left(\boldsymbol{X}^{(i)}\right)_n = y^{(i)} \tag{43}$$

*holds for every* $i \in [N]$.

*Proof.* For simplicity, we assume in this proof that there is no skip-connection in feed-forward layers, as the modification for networks with skip-connections is straightforward. For details on implementing the memorization results for feed-forward networks in Transformers, refer to Kim et al. (2023).

For each input sequence $\boldsymbol{X}^{(i)}$ with $i \in [N]$, we define its multiset expression $m^{(i)} \in \mathbb{N}^{(\mathbb{R}^d)}$ by

$$m^{(i)} : \mathbb{R}^d \to \mathbb{N}, \boldsymbol{x} \mapsto \left| \left\{ k \in [n] \mid \boldsymbol{X}_{:,k}^{(i)} = \boldsymbol{x} \right\} \right|. \tag{44}$$

The cardinality of $m^{(i)}$ for each $i \in [N]$ is at most $n$, and the token-wise $(r, \delta)$-separatedness of $\boldsymbol{X}^{(1)}, \dots, \boldsymbol{X}^{(N)}$ implies that $m^{(1)}, \dots, m^{(N)}$ are element-wise $(r, \delta)$-separated. In addition, the consistency on the labels are rephrased as follows: for any $i, j \in [N]$, we have $y^{(i)} = y^{(j)}$ if $X_{:,n}^{(i)} = X_{:,n}^{(j)}$ and $m^{(i)} = m^{(j)}$ hold.

**Construction of** $\mathcal{F}_1^{(\mathrm{FF})}$: Applying Lemma B.3 to a sequence of all distinct multisets which appear in $\{m^{(1)}, \dots, m^{(N)}\}$, we have a feed-forward network $\tilde{\phi} : \mathbb{R}^d \to \mathbb{R}$ with width 12, depth

$$\lesssim \sqrt{N \log N} + \sqrt{\frac{N}{\log N}} \cdot \max\{\log R_1, \log C_1\} \tag{45}$$

with $C_1 := \lceil 4N^3\sqrt{\pi} \rceil$ and $R_1 := 20r(nN)^2\delta^{-1}\sqrt{\pi d}$, and bit complexity bounded by

$$\lesssim \log d + \sqrt{\frac{N}{\log N}} \cdot \max\{\log R_1, \log C_1\} \tag{46}$$

such that $\tilde{\phi}(\boldsymbol{X}_{:,k}^{(i)}) \in [[4N^3\sqrt{\pi}]]$ holds for any $i \in [N]$ and $k \in [n]$, and

$$\left| \sum_{\boldsymbol{x}\in\text{supp}(m^{(i)})} \tilde{\phi}(\boldsymbol{x}) - \sum_{\boldsymbol{x}\in\text{supp}(m^{(j)})} \tilde{\phi}(\boldsymbol{x}) \right| = \left| \sum_{k=1}^{n} \tilde{\phi}(\boldsymbol{X}_{:,k}^{(i)}) - \sum_{k=1}^{n} \tilde{\phi}(\boldsymbol{X}_{:,k}^{(j)}) \right| \geq 1 \tag{47}$$

holds for any $i, j \in [N]$ such that $m^{(i)} \neq m^{(j)}$.

We extend the feed-forward network $\tilde{\phi}$ to retain the information of the input token. Let $\mathcal{V}$ be a set of all input tokens, that is, $\mathcal{V} = \{\boldsymbol{X}_{:,k}^{(i)} \mid i \in [N],\ k \in [n]\}$. Since the input sequences are token-wise $(r, \delta)$-separated, by applying Lemma C.2 to $\mathcal{V}$, we have a feed-forward network $F : \mathbb{R}^d \to \mathbb{R}$ with width 1, depth 2 and bit complexity $\log(3dr(nN)^2\sqrt{\pi}\delta^{-1})$ such that

$$0 \leq F(\boldsymbol{X}_{:,k}^{(i)}) \leq 10r(nN)^2\delta^{-1}\sqrt{\pi d} \tag{48}$$

for every $i \in [N]$ and $k \in [n]$, and

$$\left| F(\boldsymbol{X}_{:,k}^{(i)}) - F(\boldsymbol{X}_{:,l}^{(j)}) \right| \geq 2 \tag{49}$$

for every $i, j \in [N]$ and $k, l \in [n]$ with $\boldsymbol{X}_{:,k}^{(i)} \neq \boldsymbol{X}_{:,l}^{(j)}$. Notice that the depth of the feed-forward network $\tilde{\phi}$ is at least 2. Thus, it is possible to parallelly attach the above 2-layer network $F$ to the first 2-layer of $\tilde{\phi}$, and extend the hidden dimension of the remaining layers of $\tilde{\phi}$ by one to propagate the value of $F$ to the last layer. Furthermore, we augment the output dimension by one more and pad by 0, which is used to store the average value of $\tilde{\phi}$. Let $f_1^{(\text{FF})} : \mathbb{R}^d \to \mathbb{R}^3$ be the network obtained by the above procedure, that is, for any $\boldsymbol{x} \in \mathbb{R}^d$, the output of $f_1^{(\text{FF})}$ is

$$f_1^{(\text{FF})}(\boldsymbol{x}) = (\tilde{\phi}(\boldsymbol{x}), F(\boldsymbol{x}), 0)^\top. \tag{50}$$

Then, the width of $f_1^{(\text{FF})}$ is that of $\tilde{\phi}$ plus two, which is 14. The depth and the bit complexity of $f_1^{(\text{FF})}$, on the other hand, remain the same, because the depth and the bit complexity of $F$ is smaller than those of $\tilde{\phi}$. We also define a token-wise operation $\mathcal{F}_1^{(\text{FF})} : \mathbb{R}^{d\times n} \to \mathbb{R}^{3\times n}$ by

$$\mathcal{F}_1^{(\text{FF})}(\boldsymbol{X})_{:,k} := f_1^{(\text{FF})}(\boldsymbol{X}_{:,k}) \quad (k = 1, \ldots, n). \tag{51}$$

**Construction of the self-attention layer**: Let $\boldsymbol{W}^{(V)} \in \mathbb{R}^{3\times 3}$ and $\boldsymbol{W}^{(O)} \in \mathbb{R}^{3\times 3}$ be any value matrix and projection matrix such that their multiplication is

$$\boldsymbol{W}^{(O)}\boldsymbol{W}^{(V)} = \begin{pmatrix} 0 & 0 & 0 \\ 0 & 0 & 0 \\ 1 & 0 & 0 \end{pmatrix}. \tag{52}$$

The output, which we denote by $\boldsymbol{s}_k^{(i)} \in \mathbb{R}^3$, of the self-attention layer with the value matrix $\boldsymbol{W}^{(V)}$ and projection matrix $\boldsymbol{W}^{(O)}$ for the input $\boldsymbol{X}^{(i)}$ at index $k \in [n]$ is calculated as

$$\begin{aligned}
\boldsymbol{s}_k^{(i)} &:= \mathcal{F}^{(\text{UA})} \circ \mathcal{F}_1^{(\text{FF})}\left(\boldsymbol{X}^{(i)}\right)_{:,k} \\
&= \frac{1}{n}\sum_{l=1}^{n} \boldsymbol{W}^{(O)}\boldsymbol{W}^{(V)} f_1^{(\text{FF})}\left(\boldsymbol{X}_{:,l}^{(i)}\right) + f_1^{(\text{FF})}\left(\boldsymbol{X}_{:,k}^{(i)}\right) \\
&= \frac{1}{n}\sum_{l=1}^{n} \begin{pmatrix} 0 & 0 & 0 \\ 0 & 0 & 0 \\ 1 & 0 & 0 \end{pmatrix} \begin{pmatrix} \tilde{\phi}(\boldsymbol{X}_{:,l}^{(i)}) \\ F(\boldsymbol{X}_{:,l}^{(i)}) \\ 0 \end{pmatrix} + \begin{pmatrix} \tilde{\phi}(\boldsymbol{X}_{:,k}^{(i)}) \\ F(\boldsymbol{X}_{:,k}^{(i)}) \\ 0 \end{pmatrix} \\
&= \begin{pmatrix} \tilde{\phi}(\boldsymbol{X}_{:,k}^{(i)}) \\ F(\boldsymbol{X}_{:,k}^{(i)}) \\ \frac{1}{n}\sum_{l=1}^{n}\tilde{\phi}(\boldsymbol{X}_{:,l}^{(i)}) \end{pmatrix}. \tag{53}
\end{aligned}$$

We verify that the right-hand side is a context id, in the sense of Definition 4.1. Fix any $i, j \in [N]$. If $\boldsymbol{X}_{:,n}^{(i)} \neq \boldsymbol{X}_{:,n}^{(j)}$, then according to eq. (49), we have $\left|F(\boldsymbol{X}_{:,n}^{(i)}) - F(\boldsymbol{X}_{:,n}^{(j)})\right| \geq 2$. On the other hand, if $\boldsymbol{X}^{(i)}$ are not permutation of $\boldsymbol{X}^{(j)}$, i.e., $m^{(i)} \neq m^{(j)}$, then eq. (47) implies that

$$\left| \frac{1}{n} \sum_{k=1}^{n} \tilde{\phi}(\boldsymbol{X}_{:,k}^{(i)}) - \frac{1}{n} \sum_{k=1}^{n} \tilde{\phi}(\boldsymbol{X}_{:,k}^{(j)}) \right| \geq \frac{1}{n}. \tag{54}$$

Therefore, the difference of any two $n$-th outputs of the self-attention layer is lower-bounded by

$$\left\| \boldsymbol{s}_n^{(i)} - \boldsymbol{s}_n^{(j)} \right\|_2 = \left\| \begin{pmatrix} \tilde{\phi}(\boldsymbol{X}_{:,n}^{(i)}) \\ F(\boldsymbol{X}_{:,n}^{(i)}) \\ \frac{1}{n} \sum_{k=1}^{n} \tilde{\phi}(\boldsymbol{X}_{:,k}^{(i)}) \end{pmatrix} - \begin{pmatrix} \tilde{\phi}(\boldsymbol{X}_{:,n}^{(j)}) \\ F(\boldsymbol{X}_{:,n}^{(j)}) \\ \frac{1}{n} \sum_{k=1}^{n} \tilde{\phi}(\boldsymbol{X}_{:,k}^{(j)}) \end{pmatrix} \right\|_2$$

$$\geq \min \left\{ \left| F(\boldsymbol{X}_{:,n}^{(i)}) - F(\boldsymbol{X}_{:,n}^{(j)}) \right|, \left| \frac{1}{n} \sum_{k=1}^{n} \tilde{\phi}(\boldsymbol{X}_{:,k}^{(i)}) - \frac{1}{n} \sum_{k=1}^{n} \tilde{\phi}(\boldsymbol{X}_{:,k}^{(j)}) \right| \right\}$$

$$\geq \frac{1}{n} \tag{55}$$

for any $i, j \in [N]$ such that either $\boldsymbol{X}_{:,n}^{(i)} \neq \boldsymbol{X}_{:,n}^{(j)}$ or $m^{(i)} \neq m^{(j)}$ holds. As for the magnitude of each output of the self-attention layer, it is upper-bounded by

$$\left\| \boldsymbol{s}_n^{(i)} \right\|_2 = \left\| \begin{pmatrix} \tilde{\phi}(\boldsymbol{X}_{:,n}^{(i)}) \\ F(\boldsymbol{X}_{:,n}^{(i)}) \\ \frac{1}{n} \sum_{k=1}^{n} \tilde{\phi}(\boldsymbol{X}_{:,k}^{(i)}) \end{pmatrix} \right\|_2$$

$$\leq \left| \tilde{\phi}(\boldsymbol{X}_{:,n}^{(i)}) \right| + \left| F(\boldsymbol{X}_{:,n}^{(i)}) \right| + \left| \frac{1}{n} \sum_{k=1}^{n} \tilde{\phi}(\boldsymbol{X}_{:,k}^{(i)}) \right|$$

$$\leq \lceil 4N^3 \sqrt{\pi} \rceil + 10r(nN)^2 \delta^{-1} \sqrt{\pi d} + \lceil 4N^3 \sqrt{\pi} \rceil$$

$$\leq 20rn^2 N^3 \delta^{-1} \sqrt{\pi d}, \tag{56}$$

where we used the assumption $r \geq 1$ and $\delta \leq 1$ in the last line.

**Construction of $\mathcal{F}_2^{(\mathrm{FF})}$:** What remains to do is construct a network $f_2^{(\mathrm{FF})} : \mathbb{R}^3 \to \mathbb{R}$ which associates outputs of the self-attention layer with their corresponding labels. Specifically, since we know from eqs. (55) and (56) that the sequence of unique elements in $\boldsymbol{s}_n^{(1)}, \ldots, \boldsymbol{s}_n^{(N)}$ are $(20rn^2 N^3 \delta^{-1} \sqrt{\pi d}, 1/n)$-separated, by applying Lemma C.1 to $N$ inputs $\boldsymbol{s}_n^{(1)}, \ldots, \boldsymbol{s}_n^{(N)}$ and their labels $y^{(1)}, \ldots, y^{(N)}$, we have a feed-forward network $f_2^{(\mathrm{FF})} : \mathbb{R}^3 \to \mathbb{R}$ with width 12, depth

$$\lesssim \sqrt{N \log N} + \sqrt{\frac{N}{\log N}} \cdot \max\{\log R_2, \log C\} \tag{57}$$

with $R_2 := 20 \cdot 20rn^2 N^3 \delta^{-1} \sqrt{\pi d} \cdot N^2 \cdot n \cdot \sqrt{3\pi} = 400\sqrt{3d} n^3 rN^5 \delta^{-1} \pi$, and bit complexity bounded by

$$\lesssim \sqrt{\frac{N}{\log N}} \cdot \max\{\log R_2, \log C\} \tag{58}$$

such that $f_2^{(\mathrm{FF})}(\boldsymbol{s}_n^{(i)}) = y^{(i)}$ for every $i \in [N]$. In particular, this means that by defining a token-wise operation $\mathcal{F}_2^{(\mathrm{FF})} : \mathbb{R}^{3 \times n} \to \mathbb{R}^n$ as

$$\mathcal{F}_2^{(\mathrm{FF})}(\boldsymbol{X})_k := f_2^{(\mathrm{FF})}(\boldsymbol{X}_{:,k}) \quad (k = 1, \ldots, n), \tag{59}$$

we have

$$\mathcal{F}_2^{(\mathrm{FF})} \circ \mathcal{F}^{(\mathrm{UA})} \circ \mathcal{F}_1^{(\mathrm{FF})} \left( \boldsymbol{X}^{(i)} \right)_{:,n} = y^{(i)} \tag{60}$$

for every $i \in [N]$.

**Model complexity**: The width of the Transformer $\mathcal{F}_2^{(\text{FF})} \circ \mathcal{F}^{(\text{UA})} \circ \mathcal{F}_1^{(\text{FF})}$ is the maximum of widths of $\mathcal{F}_1^{(\text{FF})}$, $\mathcal{F}^{(\text{UA})}$ and $\mathcal{F}_2^{(\text{FF})}$, which is $\max(14, 3, 12) = 14$. The depth is upper-bounded by the addition of depths of $\mathcal{F}_1^{(\text{FF})}$ and $\mathcal{F}_2^{(\text{FF})}$ plus one, which implies that the depth is

$$\lesssim \sqrt{N \log N} + \sqrt{\frac{N}{\log N}} \cdot \max\{\log R_1, \log C_1\}$$

$$+ \sqrt{N \log N} + \sqrt{\frac{N}{\log N}} \cdot \max\{\log R_2, \log C\}$$

$$\lesssim \sqrt{N \log N} + \sqrt{\frac{N}{\log N}} \cdot \max\{\log R, \log C\} \tag{61}$$

with $R := R_2 = 400\sqrt{3}dn^3rN^5\delta^{-1}\pi \geq \max\{\log R_1, \log C_1, \log R_2\}$. Likewise, the bit complexity is

$$\lesssim \log d + \sqrt{\frac{N}{\log N}} \cdot \max\{\log R_1, \log C_1, \log R_2, \log C\}$$

$$\lesssim \log d + \sqrt{\frac{N}{\log N}} \cdot \max\{\log R, \log C\}. \tag{62}$$

$\square$

### B.1.2 LOWER BOUND

For convenience, we restate the statement of Theorem 4.2 below.

**Theorem B.2.** *Suppose a Transformer $\mathcal{N} : \mathbb{R}^{d \times n} \to \mathbb{R}^n$ defined by eq. (4) can shatter a set of $N$ distinct input sequences $\boldsymbol{X}^{(1)}, \ldots, \boldsymbol{X}^{(N)} \in \mathbb{R}^{d \times n}$ with $X_{:,k}^{(i)} = X_{:,1}^{(i)}$ for any $i \in [N]$ and $k \in [n]$, in the sense that for any label assignments $y^{(1)}, \ldots, y^{(N)} \in \{0, 1\}$, there are parameters with which $\mathcal{N}(\boldsymbol{X}^{(i)})_n = y^{(i)}$ holds for any $i \in [N]$. Then, the Transformer $\mathcal{N}$ has at least $\Omega(\sqrt{N})$ parameters.*

*Proof.* We denote by $L$ the depth of the Transformer $F$, and a feature matrix at block $l = 1, \ldots, L$ by

$$h_l(\boldsymbol{X}) := \mathcal{F}_l^{(\text{FF})} \circ \mathcal{F}_l^{(\text{SA})} \circ \cdots \circ \mathcal{F}_1^{(\text{FF})} \circ \mathcal{F}_1^{(\text{SA})} \circ \mathcal{E}_{\text{in}}(\boldsymbol{X}) \in \mathbb{R}^{m \times n}, \tag{63}$$

with $h_0(\boldsymbol{X}) = \mathcal{E}_{\text{in}}(\boldsymbol{X})$ and $\mathcal{N}(\boldsymbol{X}) = \mathcal{E}_{\text{out}} \circ h_L(\boldsymbol{X})$ for any input $\boldsymbol{X} \in \mathbb{R}^{d \times n}$. Then, the permutation equivariance of Transformers implies that the feature matrix $h_l$ at block $l = 1, \ldots, L$ satisfies

$$h_l(\boldsymbol{X}^{(i)})_{:,1} = \cdots = h_l(\boldsymbol{X}^{(i)})_{:,n} \tag{64}$$

for each $i \in [N]$. Thus, the self-attention layer at block $l = 1, \ldots, L$ can be calculated by

$$\mathcal{F}_l^{(\text{SA})} \left(h_{l-1}(\boldsymbol{X}^{(i)})\right)_{:,k}$$

$$= h_{l-1}(\boldsymbol{X}^{(i)})_{:,k} + \sum_{h=1}^{H} \boldsymbol{W}_{h,l}^{(O)} \boldsymbol{W}_{h,l}^{(V)} h_{l-1}(\boldsymbol{x}^{(i)}) \sigma_S \left[ \left(\boldsymbol{W}_{h,l}^{(K)} h_{l-1}(\boldsymbol{X}^{(i)})\right)^\top \left(\boldsymbol{W}_{h,l}^{(Q)} h_{l-1}(\boldsymbol{X}^{(i)})\right) \right]$$

$$= h_{l-1}(\boldsymbol{X}^{(i)})_{:,k} + \sum_{h=1}^{H} \boldsymbol{W}_{h,l}^{(O)} \boldsymbol{W}_{h,l}^{(V)} h_{l-1}(\boldsymbol{X}^{(i)})_{:,k}$$

$$= \left(\boldsymbol{I} + \sum_{h=1}^{H} \boldsymbol{W}_{h,l}^{(O)} \boldsymbol{W}_{h,l}^{(V)}\right) h_{l-1}(\boldsymbol{X}^{(i)})_{:,k}, \tag{65}$$

where $\boldsymbol{W}_{h,l}^{(O)}$, $\boldsymbol{W}_{h,l}^{(V)}$, $\boldsymbol{W}_{h,l}^{(K)}$ and $\boldsymbol{W}_{h,l}^{(Q)}$ with $h \in H$ are weight matrices for the self-attention at block $l$, and $\boldsymbol{I} \in \mathbb{R}^{m \times m}$ is the identity matrix. This observation indicates that calculations of self-attention layers for inputs $\boldsymbol{X}^{(1)}, \ldots, \boldsymbol{X}^{(N)}$ reduces to linear transformations, which in turn implies that the behavior of the Transformer $\mathcal{N}$ at inputs $\boldsymbol{X}^{(1)}, \ldots, \boldsymbol{X}^{(N)}$ can be simulated by a feed-forward network with equal or fewer parameters, and with inputs $X_{:,1}^{(1)}, \ldots, X_{:,1}^{(N)}$. Since it is known that the VC dimension of ReLU-based feed-forward networks with $W$ parameters is at most $O(W^2)$ (Goldberg & Jerrum, 1995), the Transformer $\mathcal{N}$ must have at least $\Omega(\sqrt{N})$ parameters. $\qquad\square$

## B.2 SEQUENCE-TO-SEQUENCE SETTING - LOWER BOUND

Before proceeding to the proof of Theorem 4.3, we cite the following lemma. Here sgn is the sign function:

$$\mathrm{sgn}(x) := \begin{cases} 1 & \text{if } x > 0, \\ 0 & \text{if } x = 0, \\ -1 & \text{if } x < 0. \end{cases} \tag{66}$$

**Lemma B.4** (Goldberg & Jerrum (1995)). *Suppose $W \leq M$ and let $P_1, \ldots, P_M$ be polynomials of degree at most $D$ in $W$ variables. Define*

$$K := \left| \left\{ (\mathrm{sgn}(P_1(\boldsymbol{a})), \ldots, \mathrm{sgn}(P_M(\boldsymbol{a}))) \mid \boldsymbol{a} \in \mathbb{R}^W \right\} \right|, \tag{67}$$

*i.e., $K$ is the number of possible sign vectors attained by the polynomials. Then we have $K \leq (8eMD/W)^W$.*

Hereafter, let $W$ be the nubmer of parameters and $\boldsymbol{\theta} \in \mathbb{R}^W$ be a vector of all parameters of a Transformer. We also denote by $\mathcal{N}_{\boldsymbol{\theta}}$ the Transformer to emphasize the presence of the parameter vector $\boldsymbol{\theta}$. For convenience, we present the statement of Theorem 4.3 below.

**Theorem B.3** (Lower bound). *Let $\mathcal{N}_{\boldsymbol{\theta}} : \mathbb{R}^{d \times n} \to \mathbb{R}^n$ be a Transformer defined by eq. (4) with self-attention layers replaced with hard attention layers (eq. (12)). In addition, suppose $\mathcal{N}_{\boldsymbol{\theta}}$ can shatter a set of $N$ input sequences $\boldsymbol{X}^{(1)}, \ldots, \boldsymbol{X}^{(N)} \in \mathbb{R}^{d \times n}$ with $X_{:,k}^{(i)} \neq X_{:,l}^{(j)}$ for any $i, j \in [N]$ and $k, l \in [n]$ ($i \neq j$ or $k \neq l$), in the sense that for any label assignments $\boldsymbol{y}^{(1)}, \ldots, \boldsymbol{y}^{(N)} \in \{0,1\}^n$, there is a parameter vector $\boldsymbol{\theta} \in \mathbb{R}^W$ such that*

$$\mathcal{N}_{\boldsymbol{\theta}}(\boldsymbol{X}^{(i)}) = \boldsymbol{y}^{(i)} \tag{68}$$

*for any $i \in [N]$. Then, the Transformer has at least $W = \Omega\left(\sqrt{\frac{nN}{\log(nN)}}\right)$ parameters.*

*Proof.* Recall that the Transformer $\mathcal{N}_{\boldsymbol{\theta}} : \mathbb{R}^{d \times n} \to \mathbb{R}^n$ is defined as

$$\mathcal{N}_{\boldsymbol{\theta}} := \mathcal{E}_{\mathrm{out}} \circ \mathcal{F}_L \circ \cdots \circ \mathcal{F}_1 \circ \mathcal{E}_{\mathrm{in}}, \tag{69}$$

where the $l$-th block $\mathcal{F}_l : \mathbb{R}^{m \times n} \to \mathbb{R}^{m \times n}$ is composed of a self-attention layer and a feed-forward layer. For the Transformer $\mathcal{N}_{\boldsymbol{\theta}}$ to memorize all label assignments for given $N$ input sequences with length $n$, the number of possible sign assignments for outputs of the Transformer must be at least equal to or more than $2^{nN}$, that is,

$$2^{nN} \leq K := \left| \left\{ \left( \mathrm{sgn}\left( \mathcal{N}_{\boldsymbol{\theta}}(\boldsymbol{X}^{(i)})_k \right) \right)_{\substack{i \in [N] \\ k \in [n]}} \middle| \boldsymbol{\theta} \in \mathbb{R}^W \right\} \right| \tag{70}$$

must hold. We estimate the upper-bound on the right-hand of the above inequality.

Our strategy is to partition the set of parameters inductively with respect to the layers, so that on each cell the output of the Transformer can be expressed by some polynomial function on the parameters. To be more precise, we construct a sequence of partitions $\mathcal{S}_0, \mathcal{S}_1, \ldots, \mathcal{S}_L \in \mathcal{P}(\mathbb{R}^W)$ such that

1. for each $l = 0, 1, \ldots, L$, $\mathcal{S}_l$ is a partition of the set of parameters, that is,

$$S_i \cap S_j = \emptyset \quad (\forall S_i, S_j \in \mathcal{S}_l \text{ with } S_i \neq S_j) \quad \text{and} \quad \bigcup_{S \in \mathcal{S}_l} S = \mathbb{R}^W, \tag{71}$$

and is also a refinement of $\mathcal{S}_{l-1}$ when $l \geq 1$, in the sense that for every cell $S \in \mathcal{S}_l$, there is a cell $S' \in \mathcal{S}_{l-1}$ with $S \subset S'$.

2. for each $l \in [L]$, the number of cells in $S_l$ satisfies

$$\frac{|\mathcal{S}_l|}{|\mathcal{S}_{l-1}|} \leq \left( \frac{8e \cdot n^3 HN \cdot (8l-4)}{W_{l-1} + 2msH} \right)^{W_{l-1}+2msH}$$
$$\cdot \left( \frac{8e \cdot nqN \cdot 4l}{W_{l-1} + 2msH + (m+1)q} \right)^{W_{l-1}+2msH+(m+1)q}, \qquad (72)$$

where $W_{l-1}$ is the number of parameters up to the $(l-1)$-th block, with $W_0 := dm$, the number of parameters in $\mathcal{E}_{\text{in}}$.

3. for each $l = 0, 1, \ldots, L$, outputs of the $l$-th block for input $\boldsymbol{X}^{(i)}$ on each cell $S \in \mathcal{S}_l$

$$p_{l,u,k,S}^{(i)}(\boldsymbol{\theta}_l) := \mathcal{F}_l \circ \cdots \circ \mathcal{F}_1 \circ \mathcal{E}_{\text{in}}(\boldsymbol{X}^{(i)})_{u,k} \quad (u \in [m], \ k \in [n]) \qquad (73)$$

are polynomial functions in variable $\boldsymbol{\theta}_l$ of degree at most $4l+1$, as long as $\boldsymbol{\theta}$ varies within the cell $S$. Here $\boldsymbol{\theta}_l \in \mathbb{R}^{W_l}$ is a part of $\boldsymbol{\theta}$ corresponding to parameters up to the $l$-th block, with $\boldsymbol{\theta}_0$ defined by a parameter vector of $\mathcal{E}_{\text{in}}$.

First, we set $\mathcal{S}_0 := \{\mathbb{R}^W\}$. Notice that outputs $\mathcal{E}_{\text{in}}(\boldsymbol{X}^{(i)})_{u,k}$ for all $u \in [m], k \in [n]$ and $i \in [N]$ are polynomial functions in variable $\boldsymbol{\theta}_0$ of degree 1, because $\mathcal{E}_{\text{in}} : \mathbb{R}^{d \times n} \to \mathbb{R}^{m \times n}$ is a token-wise linear mapping.

Next, suppose a sequence of partitions $\mathcal{S}_0, \ldots, \mathcal{S}_{l-1}$ for $l \in [L-1]$ is already given, and we construct a partition $\mathcal{S}_l$ from them. Specifically, we subdivide each cell $S \in \mathcal{S}_{l-1}$ to create a new partition $\mathcal{S}_l$. By assumption, on each cell $S \in \mathcal{S}_{l-1}$ the inputs of the $(l-1)$-th block $\mathcal{F}_{l-1}$ for the input $\boldsymbol{X}^{(i)}$

$$p_{l-1,u,k,S}^{(i)}(\boldsymbol{\theta}_{l-1}) := \mathcal{F}_{l-1} \circ \cdots \circ \mathcal{F}_1 \circ \mathcal{E}_{\text{in}}(\boldsymbol{X}^{(i)})_{u,k} \quad (u \in [m], \ k \in [n]), \qquad (74)$$

are polynomial functions in variable a parameter vector $\boldsymbol{\theta}_{l-1}$ of degree no more than $4(l-1)+1$, as long as $\boldsymbol{\theta}$ varies in the cell $S$.

**Self-attention subblock**: Recall that the self-attention layer with the hardmax function in the $l$-the block for the input sequence $\boldsymbol{X}^{(i)}$ is calculated as follows.

$$\mathcal{F}_l^{(\text{HA})}(\boldsymbol{Z}^{(i)}) := \boldsymbol{Z}^{(i)} + \sum_{h=1}^{H} \boldsymbol{W}_{hl}^{(O)} \boldsymbol{W}_{hl}^{(V)} \boldsymbol{Z}^{(i)} \sigma_H \left[ \left( \boldsymbol{W}_{hl}^{(K)} \boldsymbol{Z}^{(i)} \right)^{\top} \left( \boldsymbol{W}_{hl}^{(Q)} \boldsymbol{Z}^{(i)} \right) \right], \qquad (75)$$

where $\boldsymbol{Z}^{(i)} \in \mathbb{R}^{m \times n}$ is the input of the self-attention layer for input $\boldsymbol{X}^{(i)}$. In particular, when $\boldsymbol{\theta}$ varies in a cell $S \in \mathcal{S}_{l-1}$, $Z_{u,k}^{(i)}$ for each $u \in [m], k \in [n]$ can be expressed by the polynomial function $p_{l-1,u,k,S}^{(i)}(\boldsymbol{\theta}_{l-1})$ of degree $4(l-1)+1$.

Hereafter, we subdivide each cell $S \in \mathcal{S}_{l-1}$ to construct a refinement $\mathcal{S}_l^{(\text{SA})}$ of $\mathcal{S}_{l-1}$ so that the hardmax patterns

$$\sigma_H \left[ \left( \boldsymbol{W}_{hl}^{(K)} \boldsymbol{Z}^{(i)} \right)^{\top} \left( \boldsymbol{W}_{hl}^{(Q)} \boldsymbol{Z}^{(i)} \right) \right] \in \mathbb{R}^{n \times n} \quad (\forall h \in H) \qquad (76)$$

remain the same on each cell $S' \in \mathcal{S}_l^{(\text{SA})}$. The $(k,k')$-th element of the attention matrix at head $h \in [H]$ can be written by

$$a_{l,h,k,k',S}^{(i)}(\boldsymbol{\theta}_{l-1}, \boldsymbol{W}_{hl}^{(K)}, \boldsymbol{W}_{hl}^{(Q)}) := \left( \boldsymbol{W}_{hl}^{(K)} \boldsymbol{Z}^{(i)} \right)^{\top}_{:,k} \left( \boldsymbol{W}_{hl}^{(Q)} \boldsymbol{Z}^{(i)} \right)_{:,k'}$$
$$= \sum_{u,u'=1}^{m} \left( \boldsymbol{W}_{hl}^{(K)\top} \boldsymbol{W}_{hl}^{(Q)} \right)_{u,u'} p_{l-1,u,k,S}^{(i)}(\boldsymbol{\theta}_{l-1}) p_{l-1,u',k',S}^{(i)}(\boldsymbol{\theta}_{l-1}),$$
$$\qquad (77)$$

from which we see that each element of the attention matrix is a polynomial function in variables $\boldsymbol{W}_{hl}^{(K)}, \boldsymbol{W}_{hl}^{(Q)}$ and $\boldsymbol{\theta}_{l-1}$, of degree at most $8(l-1)+4$, as long as $\boldsymbol{\theta}$ varies in the cell $S \in \mathcal{S}_{l-1}$.

We define a partition $\mathcal{P}_{l,S}^{(\mathrm{SA})}$ of $S$ based on the hardmax patterns, that is, the minimal partition of $S$ such that on each cell, all outputs of the hardmax function remain the same. To estimate the size of $\mathcal{P}_{l,S}^{(\mathrm{SA})}$, we instead consider sign patterns of polynomials

$$\left\{ a_{l,h,k,k',S}^{(i)} - a_{l,h,k'',k',S}^{(i)} \;\middle|\; i \in [N],\; h \in [H],\; k,k',k'' \in [n] \right\}, \tag{78}$$

because whenever sign patterns of the above set of polynomials do not change on some subset of the parameter space, the hardmax patterns must also remain the same. Applying Lemma B.4 to the above collection of polynomials, the size of the partition $\mathcal{P}_{l,S}^{(\mathrm{SA})}$ is upper-bounded by

$$\left( \frac{8e \cdot n^3 HN \cdot (8l - 4)}{W_{l-1} + 2msH} \right)^{W_{l-1} + 2msH}. \tag{79}$$

We define a refinement $\mathcal{S}_l^{(\mathrm{SA})}$ of $\mathcal{S}_{l-1}$ by subdividing each cell $S \in \mathcal{S}_{l-1}$ in this way, and its size is upper-bounded by

$$\left| \mathcal{S}_l^{(\mathrm{SA})} \right| \le \left| \mathcal{S}_{l-1} \right| \cdot \left( \frac{8e \cdot n^3 HN \cdot (8l - 4)}{W_{l-1} + 2msH} \right)^{W_{l-1} + 2msH}. \tag{80}$$

On each cell $S' \in \mathcal{S}_l^{(\mathrm{SA})}$, the hardmax patterns remain unchanged, which implies that

$$\left( \boldsymbol{Z}^{(i)} \sigma_H \left[ \left( \boldsymbol{W}_{hl}^{(K)} \boldsymbol{Z}^{(i)} \right)^{\top} \left( \boldsymbol{W}_{hl}^{(Q)} \boldsymbol{Z}^{(i)} \right) \right] \right)_{u,k} \qquad (u \in [m],\; k \in [n],\; h \in [H]) \tag{81}$$

are polynomial functions in variable $\boldsymbol{\theta}_{l-1}$ of degree at most $4(l-1) + 1$, as long as $\boldsymbol{\theta}$ moves in the cell $S' \in \mathcal{S}_l^{(\mathrm{SA})}$. This further means that each element of the output $\mathcal{F}_l^{(\mathrm{HA})}(\boldsymbol{Z}^{(i)})$ is a polynomial function in variables $\boldsymbol{W}_{hl}^{(O)}, \boldsymbol{W}_{hl}^{(V)}$ with $h \in [H]$ and $\boldsymbol{\theta}_{l-1}$, of degree at most $4(l-1) + 3$ on each cell $S' \in \mathcal{S}_l^{(\mathrm{SA})}$.

**Feed-forward subblock**: As for feed-forward layers, we follow the analysis given by Bartlett et al. (2019). On each cell $S' \in \mathcal{S}_l^{(\mathrm{SA})}$, the hidden layer at the $k$-th token for input $\boldsymbol{X}^{(i)}$ is

$$\boldsymbol{W}_l^{(1)} \mathcal{F}_l^{(\mathrm{HA})}(\boldsymbol{Z}^{(i)})_{:,k} + \boldsymbol{b}_l^{(1)} \in \mathbb{R}^q, \tag{82}$$

whose $v$-th element is a polynomial function in variables $\boldsymbol{W}_{hl}^{(O)}, \boldsymbol{W}_{hl}^{(V)}$ with $h \in [H]$, $\boldsymbol{W}_l^{(1)}, \boldsymbol{b}_l^{(1)}$ and $\boldsymbol{\theta}_{l-1}$ of degree at most $4(l-1) + 4$. Notice that sign patterns of polynomials

$$\left\{ \boldsymbol{W}_{l,v,:}^{(1)} \mathcal{F}_l^{(\mathrm{HA})}(\boldsymbol{Z}^{(i)})_{:,k} + b_{l,v}^{(1)} \;\middle|\; i \in [N],\; v \in [q],\; k \in [n] \right\} \tag{83}$$

completely determine whether or not the ReLU activation function in the middle layer fires. Therefore, by defining $\mathcal{P}_{l,S'}^{(\mathrm{FF})}$ as the minimal partition of $S'$ such that the activation pattern remains the same on each cell, the size of $\mathcal{P}_{l,S'}^{(\mathrm{FF})}$ is upper-bounded by Lemma B.4 as

$$\left| \mathcal{P}_{l,S'}^{(\mathrm{FF})} \right| \le \left( \frac{8e \cdot nqN \cdot 4l}{W_{l-1} + 2msH + (m+1)q} \right)^{W_{l-1} + 2msH + (m+1)q}. \tag{84}$$

We define a refinement $\mathcal{S}_l^{(\mathrm{FF})}$ of $\mathcal{S}_l^{(\mathrm{SA})}$ by subdividing each cell $S' \in \mathcal{S}_l^{(\mathrm{SA})}$ into $\mathcal{P}_{l,S'}^{(\mathrm{FF})}$. Then, outputs of the feed-forward layer

$$p_{l,u,k,S''}^{(i)}(\boldsymbol{\theta}_l) = \mathcal{F}_l^{(\mathrm{HA})}(\boldsymbol{Z}^{(i)})_{u,k} + \boldsymbol{W}_{l,u,:}^{(2)} \sigma_R \left[ \boldsymbol{W}_l^{(1)} \mathcal{F}_l^{(\mathrm{HA})}(\boldsymbol{Z}^{(i)})_{:,k} + \boldsymbol{b}_l^{(1)} \right] + b_{l,u}^{(2)} \tag{85}$$

for any $u \in [m], k \in [n]$ and $i \in [N]$ are polynomial functions in variable $\boldsymbol{\theta}_l$ of degree at most $4(l-1) + 5 = 4l + 1$, as long as the parameter vector $\boldsymbol{\theta}$ varies in the cell $S'' \in \mathcal{S}_l^{(\mathrm{FF})}$.

Finally, we set $\mathcal{S}_l$ as $\mathcal{S}_l := \mathcal{S}_l^{(\mathrm{FF})}$. Then, from the above observations we know that outputs of the $l$-th block are polynomial functions in variables $W_l$ of degree at most $4l + 1$ as long as $\boldsymbol{\theta}$ moves

within each cell $S'' \in \mathcal{S}_l$, as desired. In addition, we have

$$\frac{|\mathcal{S}_l|}{|\mathcal{S}_{l-1}|} \leq \left(\frac{8e \cdot n^3 HN \cdot (8l-4)}{W_{l-1} + 2msH}\right)^{W_{l-1}+2msH}$$
$$\cdot \left(\frac{8e \cdot nqN \cdot 4l}{W_{l-1} + 2msH + (m+1)q}\right)^{W_{l-1}+2msH+(m+1)q}, \tag{86}$$

which satisfies the second property. In this way, we have a desired sequence of partitions $\mathcal{S}_0, \ldots, \mathcal{S}_L$.

Outputs of the $L$-th block for input $\boldsymbol{X}^{(i)}$ ($i \in [N]$)

$$\mathcal{F}_L \circ \cdots \circ \mathcal{F}_1 \circ \mathcal{E}_{\text{in}}(\boldsymbol{X}^{(i)})_{u,k} \quad (u \in [m],\ k \in [n]) \tag{87}$$

are polynomial functions in variable $\boldsymbol{\theta}_L$ of degree at most $4L+1$ as long as $\boldsymbol{\theta}$ varies in each cell of $\mathcal{S}_L$, which in turn implies that final outputs of the Transformer

$$p_{k,S}^{(i)}(\boldsymbol{\theta}) := \mathcal{N}_{\boldsymbol{\theta}}(\boldsymbol{X}^{(i)})_k = \mathcal{E}_{\text{out}} \circ \mathcal{F}_L \circ \cdots \circ \mathcal{F}_1 \circ \mathcal{E}_{\text{in}}(\boldsymbol{X}^{(i)})_k \quad (k \in [n]) \tag{88}$$

are polynomial functions in variable $\boldsymbol{\theta}$ of degree at most $4L+2$ if the parameter vector $\boldsymbol{\theta}$ moves within each cell $S \in \mathcal{S}_L$.

Applying Lemma B.4 to the set of polynomials $\{p_{k,S}^{(i)}(\boldsymbol{\theta})\}_{i \in [N], k \in [n]}$ on each cell of $\mathcal{S}_L$ allows us to upper-bound $K$ as follows.

$$K = \left| \left\{ \left( \text{sgn}\left( \mathcal{N}_{\boldsymbol{\theta}}(\boldsymbol{X}^{(i)})_k \right) \right)_{\substack{i \in [N] \\ k \in [n]}} \ \middle|\ \boldsymbol{\theta} \in \mathbb{R}^W \right\} \right|$$
$$\leq \sum_{S \in \mathcal{S}_L} \left| \left\{ \left( \text{sgn}\left( \mathcal{N}_{\boldsymbol{\theta}}(\boldsymbol{X}^{(i)})_k \right) \right)_{\substack{i \in [N] \\ k \in [n]}} \ \middle|\ \boldsymbol{\theta} \in S \right\} \right|$$
$$\leq |\mathcal{S}_L| \cdot \left(\frac{8e \cdot nN \cdot (4L+2)}{W}\right)^W. \tag{89}$$

Since $|\mathcal{S}_L| = |\mathcal{S}_0| \cdot \prod_{l=1}^{L} |\mathcal{S}_l|/|\mathcal{S}_{l-1}|$ and $|\mathcal{S}_0| = 1$, the right-hand side is further expanded as

$$K \leq \left(\frac{8e \cdot nN \cdot (4L+2)}{W}\right)^W \cdot \prod_{l=1}^{L} \frac{|\mathcal{S}_l|}{|\mathcal{S}_{l-1}|}$$
$$\leq \left(\frac{8e \cdot nN \cdot (4L+2)}{W}\right)^W$$
$$\cdot \prod_{l=1}^{L} \left(\frac{8e \cdot n^3 HN \cdot (8l-4)}{W_{l-1} + 2msH}\right)^{W_{l-1}+2msH} \left(\frac{8e \cdot nqN \cdot 4l}{W_{l-1} + 2msH + (m+1)q}\right)^{W_{l-1}+2msH+(m+1)q}$$
$$\leq \left(\frac{8e \cdot nN \cdot (4L+2) + \sum_{l=1}^{L} \left[8e \cdot n^3 HN \cdot (8l-4) + 8e \cdot nqN \cdot 4l\right]}{\overline{W}}\right)^{\overline{W}}, \tag{90}$$

where we used the weighted arithmetic-geometric inequality in the last line, with $\overline{W}$ defined by

$$\overline{W} := W + \sum_{l=1}^{L} \left[W_{l-1} + 2msH + W_{l-1} + 2msH + (m+1)q\right]. \tag{91}$$

Notice that $W_l$ for each $l \in [L]$ is the number of parameters up to the $l$-th block, which indicates

$$W_l = md + \sum_{l'=1}^{l} \left[4msH + 2(m+1)q\right]$$
$$= md + l\left[4msH + 2(m+1)q\right]$$
$$\geq 4lH + 2lq \tag{92}$$

with $W = W_L + md$. With this observation, the numerator on the right-hand side of eq. (90) is upper-bounded by

$$8e \cdot nN \cdot (4L + 2) + \sum_{l=1}^{L} \left[ 8e \cdot n^3 HN \cdot (8l - 4) + 8e \cdot nqN \cdot 4l \right]$$

$$\leq 8e \cdot nN \cdot (4L + 2) + 8e \cdot n^3 N \cdot \sum_{l=1}^{L} (8lH + 4lq)$$

$$\leq 8e \cdot nN \cdot (4L + 2) + 8e \cdot n^3 N \cdot \sum_{l=1}^{L} 2W_l$$

$$\leq 48e \cdot n^3 N \cdot \overline{W}, \tag{93}$$

where we used $4L + 2 \leq 4\overline{W}$ and $\sum_{l=1}^{L} 2W_l \leq 2W + \sum_{l=1}^{L} 2W_{l-1} \leq 2\overline{W}$ in the last line. Thus, the right-hand side of eq. (90) is upper-bounded by

$$K \leq \left( \frac{48en^3 N \cdot \overline{W}}{\overline{W}} \right)^{\overline{W}} = \left( 48en^3 N \right)^{\overline{W}}. \tag{94}$$

Recall that in order to memorize all label assignments for $N$ input sequences with length $n$, $K$ is at least equal to or more than $2^{nN}$, which gives us an upper-bound of $nN$:

$$nN \leq \log_2 \left[ \left( 48en^3 N \right)^{\overline{W}} \right]$$

$$= \overline{W} \log_2 \left( 48en^3 N \right)$$

$$\leq 3\overline{W} \log_2 \left( 48enN \right). \tag{95}$$

Here we evaluate a crude upper-bound of $\overline{W}$ with respect to the number $W$ of parameters as follows.

$$\overline{W} = W + \sum_{l=1}^{L} \left[ 2W_{l-1} + 4msH + (m+1)q \right]$$

$$\leq W + 2\sum_{l=1}^{L} W_l$$

$$\leq W + 2LW \leq 3W^2, \tag{96}$$

which implies $nN \leq 3\overline{W} \log_2 \left( 48enN \right) \leq 9W^2 \log_2 \left( 48enN \right)$. Therefore, the Transformer has at least $W = \Omega \left( \sqrt{\frac{nN}{\log(nN)}} \right)$ parameters.

$\square$

## C  MEMORIZATION OF FEED-FORWARD NETWORKS

In this section, we extend the result on the optimal memorization of feed-forward networks proved by Vardi et al. (2022). Specifically, the following lemma states that we can freely add data points without severely affecting the memorization capacity of feed-forward networks, as long as their labels are zero. We would like to note that Vardi et al. (2022) implicitly used this result to show the memorization capacity of feed-forward networks with a bounded depth. Thus, our aim here is to explicitly state the result and provide a rigorous proof.

**Lemma C.1** (Extension of Vardi et al. (2022)). *Let $N, V, d, C \in \mathbb{N}$ with $N \leq V$, and $r \geq 1, 0 < \delta \leq 1$. Let $y^{(1)}, \ldots, y^{(N)} \in [C]$ be a set of $N$ labels and $\boldsymbol{x}^{(1)}, \ldots, \boldsymbol{x}^{(V)} \in \mathbb{R}^d$ be a set of $V$ inputs such that $\|\boldsymbol{x}^{(i)}\| \leq r$ for every $i \in [V]$ and $\|\boldsymbol{x}^{(i)} - \boldsymbol{x}^{(j)}\| \geq \delta$ for every $i, j \in [V]$ with $i \neq j$. Denote $R := 20rV^2 \delta^{-1} \sqrt{\pi d}$. Then, there exists a neural network $F : \mathbb{R}^d \to \mathbb{R}$ with width 12, depth*

$$\lesssim \sqrt{N \log N} + \sqrt{\frac{N}{\log N}} \cdot \max\{\log R, \log C\}, \tag{97}$$

*(for the definition of $\lesssim$, see Section 3.1) and bit complexity*

$$\lesssim \log d + \sqrt{\frac{N}{\log N}} \cdot \max\{\log R, \log C\} \tag{98}$$

*such that $F(\boldsymbol{x}^{(i)}) = y^{(i)}$ for every $i \in [N]$ and $F(\boldsymbol{x}^{(i)}) = 0$ for every $i \in [V] \setminus [N]$.*

*Proof.* The proof goes basically the same as was done in the proof of the original theorem by Vardi et al. (2022): we construct a three sub-networks $F_1, F_2$ and $F_3$ with width at most 12, and then concatenate those networks to create the final network $F = F_3 \circ F_2 \circ F_1$. The only architectural difference lies in the construction of $F_1$, and the rest of the proof is dedicated to verifying that the resulting network $F$ satisfies $F(\boldsymbol{x}^{(i)}) = 0$ for $i \in [V] \setminus [N]$.

STAGE I: PROJECTING ONTO A ONE-DIMENSIONAL SUBSPACE

In this stage, we construct a sub-network $F_1 : \mathbb{R}^d \to \mathbb{R}$, which projects each input onto the line $\mathbb{R}$ while approximately retaining their distance. We use the following lemma from Vardi et al. (2022).

**Lemma C.2** (Vardi et al. (2022)). *Let $\boldsymbol{x}^{(1)}, \dots, \boldsymbol{x}^{(N)} \in \mathbb{R}^d$ with $\|\boldsymbol{x}^{(i)}\| \leq r$ for every $i \in [N]$ and $\|\boldsymbol{x}^{(i)} - \boldsymbol{x}^{(j)}\| \geq \delta$ for every $i, j \in [N]$ with $i \neq j$. Then, there exists a neural network $F : \mathbb{R}^d \to \mathbb{R}$ with width 1, depth 2 and bit complexity $\log(3drN^2\sqrt{\pi}\delta^{-1})$, such that $0 \leq F(\boldsymbol{x}^{(i)}) \leq 10rN^2\delta^{-1}\sqrt{\pi d}$ for every $i \in [N]$ and $|F(\boldsymbol{x}^{(i)}) - F(\boldsymbol{x}^{(j)})| \geq 2$ for every $i, j \in [N]$ with $i \neq j$.*

Instead of applying the above lemma to the set of $N$ inputs $\boldsymbol{x}^{(1)}, \dots, \boldsymbol{x}^{(N)}$, here we apply it to the set of $V$ inputs $\boldsymbol{x}^{(1)}, \dots, \boldsymbol{x}^{(V)}$. Then, we obtain a neural network $\tilde{F}_1 : \mathbb{R}^d \to \mathbb{R}$ with width 1, depth 2 and bit complexity $\log(3drV^2\sqrt{\pi}\delta^{-1})$, such that $0 \leq \tilde{F}_1(\boldsymbol{x}^{(i)}) \leq 10rV^2\delta^{-1}\sqrt{\pi d}$ for every $i \in [V]$ and $|\tilde{F}_1(\boldsymbol{x}^{(i)}) - \tilde{F}_1(\boldsymbol{x}^{(j)})| \geq 2$ for every $i, j \in [V]$ with $i \neq j$.

By a slight modification to the bias term, we may construct a neural network $F_1 : \mathbb{R}^d \to \mathbb{R}$ such that $2 \leq F_1(\boldsymbol{x}^{(i)}) \leq R := 20rV^2\delta^{-1}\sqrt{\pi d}$ without affecting its width, depth and bit-complexity. We adopt $F_1$ as the first sub-network.

STAGE II: FINDING THE RIGHT SUBSET

In this stage, we adopt the same construction strategy for the second sub-network $F_2 : \mathbb{R} \to \mathbb{R}$ as was done in the proof of Vardi et al. (2022). We use Lemma G.3, whose statement is the strengthened version of the one by Vardi et al. (2022).

We denote the outputs $F_1(\boldsymbol{x}^{(1)}), \dots, F_1(\boldsymbol{x}^{(V)})$ of the first sub-network $F_1$ for $\boldsymbol{x}^{(1)}, \dots, \boldsymbol{x}^{(V)}$ by $x_1, \dots, x_V$. In addition, by rearranging labels, we assume without loss of generality that the first $N$ outputs $x_1, \dots, x_N$ are in an increasing order, that is, $x_1 < \cdots < x_N$.

Let $m := \sqrt{N \log N}$, and $w_1, \dots, w_{\sqrt{N \log N}}$ and $u_1, \dots, u_{\sqrt{N \log N}}$ be two sets of $\sqrt{\frac{N}{\log N}} \cdot \log C$-bit sequences and $\sqrt{\frac{N}{\log N}} \cdot \log R$-bit sequences, respectively, such that for every $i \in [N]$, let $j := \left\lceil i \cdot \sqrt{\frac{\log N}{N}} \right\rceil \in [m], k := i \mod \sqrt{\frac{N}{\log N}}$, then $w_1, \dots, w_{\sqrt{N \log N}}$ and $u_1, \dots, u_{\sqrt{N \log N}}$ are defined by identities

$$\text{BIN}_{k \cdot \log C + 1 : (k+1) \cdot \log C}(w_j) = y^{(i)}, \tag{99}$$

$$\text{BIN}_{k \cdot \log R + 1 : (k+1) \cdot \log R}(u_j) = \lfloor x_i \rfloor, \tag{100}$$

where we used the fact that the outputs of the first sub-network $F_1$ are non-negative and upper-bounded by $R := 20rV^2\delta^{-1}\sqrt{\pi d}$

Next, by applying Lemma G.3 to $w_1, \dots, w_{\sqrt{N \log N}}$ and $u_1, \dots, u_{\sqrt{N \log N}}$, respectively, we obtain two networks $F_2^w : \mathbb{R} \to \mathbb{R}$ and $F_2^u : \mathbb{R} \to \mathbb{R}$ with width 4, depth $3\sqrt{N \log N} + 2$ and bit complexity

at most $\sqrt{\frac{N}{\log N}} \cdot \max\{\log C, \log R\} + \lceil \log R \rceil$ such that for every $i \in [N]$,

$$F_2^w(x_i) = w_{j_i} \text{ and } F_2^u(x_i) = u_{j_i} \tag{101}$$

hold with $j_i := \left\lceil i \cdot \sqrt{\frac{\log N}{N}} \right\rceil$. By concatenating these two networks $F_2^w$ and $F_2^u$, we construct a second sub-network $F_2 : \mathbb{R} \to \mathbb{R}^3$ such that for any $i \in [N]$ we have

$$F_2(x_i) = \begin{pmatrix} x_i \\ w_{j_i} \\ u_{j_i} \end{pmatrix}. \tag{102}$$

As for the outputs of $F_2$ for $x_{N+1}, \dots, x_V$, since the construction of the first sub-network $F_1$ assures that $|x_i - x_j| \geq 2$ for every $i, j \in [V]$ with $i \neq j$, Lemma G.3 indicates that for any $i \in [V] \setminus [N]$, we have

$$F_2(x_i) = \begin{pmatrix} x_i \\ w \\ u \end{pmatrix}, \tag{103}$$

where $w$ (resp. $u$) is either 0 or $w_j$ (resp. $u_j$) for some $j \in [m]$.

STAGE III: BIT EXTRACTION FROM THE CRAFTED WEIGHTS

As in the previous stage, we follow the same construction strategy as is done in Vardi et al. (2022). However, here we inspect the behavior of the third sub-network for $x_{N+1}, \dots, x_V$.

We use the function obtained by Lemma G.6 with $\rho = \log C, n = \sqrt{\frac{N}{\log N}}$ and $c = \log R$ as the third sub-network $F_3 : \mathbb{R}^3 \to \mathbb{R}$ with width 12, depth $3\sqrt{\frac{N}{\log N}} \cdot \max\{\log R, \log C\} + 2\sqrt{\frac{N}{\log N}} + 2$ and bit complexity $\sqrt{\frac{N}{\log N}} \max\{\log R, \log C\} + 2$. Then, we construct the final network $F : \mathbb{R}^d \to \mathbb{R}$ by setting $F := F_3 \circ F_2 \circ F_1$.

VERIFICATION OF BEHAVIOR AND MODEL COMPLEXITY

Hereafter, we check that the configured network $F = F_3 \circ F_2 \circ F_1$ correctly outputs the desired values, that is, for any $i \in [N]$ we have

$$F(\boldsymbol{x}^{(i)}) = y^{(i)}, \tag{104}$$

and for any $i \in [V] \setminus [N]$

$$F(\boldsymbol{x}^{(i)}) = 0. \tag{105}$$

Fix $i \in [N]$ with $j_i := \left\lceil i \cdot \sqrt{\frac{\log N}{N}} \right\rceil$. The output of $F_2 \circ F_1$ for $\boldsymbol{x}^{(i)}$ is

$$F_2 \circ F_1(\boldsymbol{x}^{(i)}) = \begin{pmatrix} x_i \\ w_{j_i} \\ u_{j_i} \end{pmatrix}. \tag{106}$$

Since $\lfloor x_i \rfloor = \mathrm{BIN}_{\rho \cdot k+1:\rho \cdot (k+1)}(u_{j_i})$ with $k := i \mod \sqrt{\frac{N}{\log N}}$ by definition, Lemma G.6 implies $F_3 \circ F_2 \circ F_1(\boldsymbol{x}^{(i)}) = \mathrm{BIN}_{\rho \cdot k+1:\rho \cdot (k+1)}(w_{i_j}) = y^{(i)}$ as desired.

On the other hand, for any $i \in [V] \setminus [N]$, the output of $F_2 \circ F_1$ for $\boldsymbol{x}^{(i)}$ is

$$F_2 \circ F_1(\boldsymbol{x}^{(i)}) = \begin{pmatrix} x_i \\ w \\ u \end{pmatrix}, \tag{107}$$

where $w$ (resp. $u$) is either 0 or $w_j$ (resp. $u_j$) for some $j \in [m]$. If $u = 0$, then $x_i$ satisfies

$$|x_i - 1/2 - \mathrm{BIN}_{\rho \cdot j+1:\rho \cdot (j+1)}(u)| = |x_i - 1/2| > 1, \tag{108}$$

because the construction of the first sub-network $F_1$ guarantees that $x_1, \ldots, x_V \geq 2$. Thus, Lemma G.6 implies that $F(\boldsymbol{x}^{(i)}) = F_3 \circ F_2 \circ F_1(\boldsymbol{x}^{(i)}) = 0$ as desired. On the other hand, if $u = u_j$ for some $j \in [m]$, $x_i$ should satisfy $|x_i - 1/2 - \mathrm{BIN}_{\rho \cdot k + 1 : \rho \cdot (k+1)}(u)| > 1$ for any $k \in \{0, \ldots, \sqrt{\frac{N}{\log N}} - 1\}$. This is because for each $k$, $\mathrm{BIN}_{\rho \cdot k + 1 : \rho \cdot (k+1)}(u)$ equals $\lfloor x_l \rfloor$ for some $l \in [N]$ by definition, which together with the separatedness of $x_1, \ldots, x_V$ implies

$$
\begin{aligned}
|x_i - 1/2 - \mathrm{BIN}_{\rho \cdot k + 1 : \rho \cdot (k+1)}(u)| &= |x_i - 1/2 - \lfloor x_l \rfloor| \\
&> |x_i - x_l| - |x_l - 1/2 - \lfloor x_l \rfloor| \\
&\geq 2 - 1/2 > 1.
\end{aligned}
\tag{109}
$$

Therefore, the output of $F = F_3 \circ F_2 \circ F_1$ for $x_i$ in this case is again 0.

The width of $F$ is the maximal width of its sub-networks, which corresponds to the width of $F_3$, i.e., 12. The depth of $F$ is the sum of the depths of $F_1$, $F_2$ and $F_3$, which is estimated as

$$
\begin{aligned}
&2 + 3\sqrt{N \log N} + 2 + 3\sqrt{\frac{N}{\log N}} \cdot \max\{\log R, \log C\} + 2\sqrt{\frac{N}{\log N}} + 2 \\
&\lesssim \sqrt{N \log N} + \sqrt{\frac{N}{\log N}} \cdot \max\{\log R, \log C\}.
\end{aligned}
\tag{110}
$$

The bit complexity of $F$ is the maximal bit complexity of its sub-networks, which is upper-bounded by

$$
\begin{aligned}
&\max\left\{\log(3drV^2\sqrt{\pi}\delta^{-1}), \sqrt{\frac{N}{\log N}} \cdot \max\{\log C, \log R\} + \lceil \log R \rceil, \right. \\
&\left. \qquad \sqrt{\frac{N}{\log N}} \max\{\log R, \log C\} + 2\right\} \\
&\lesssim \log d + \sqrt{\frac{N}{\log N}} \cdot \max\{\log C, \log R\}.
\end{aligned}
\tag{111}
$$

$\square$

## D   Memorization Capacity of Deep Sets

Refer to Appendix A for the definition of multiset and the notation in this paper.

Deep set (Zaheer et al., 2017) is a well-known architecture used for modeling functions that take a set, or more generally a multiset as input. The architecture is stated in a very general form, and it is known (Wagstaff et al., 2022) that any permutation invariant function for multisets over countable domain $\mathcal{X}$ can be decomposed by appropriate functions $\phi$ and $\rho$ as follows:

$$
(\phi, \rho)(m) = \rho\left(\sum_{k=1}^{n} \phi(\boldsymbol{x}_k)\right) \quad \text{with } m = \{\!\{\boldsymbol{x}_1, \ldots, \boldsymbol{x}_n\}\!\} \in \mathbb{N}^{\mathcal{X}}.
\tag{112}
$$

In this paper, we define a deep set by a tuple $(\phi, \rho)$, where $\phi$ and $\rho$ are feed-forward networks. In addition, the **width** of the deep set $(\phi, \rho)$ is defined as the maximum of the widths of $\phi$ and $\rho$, and the **depth** of $(\phi, \rho)$ as the addition of the depths of $\phi$ and $\rho$.

**Theorem D.1** (Memorization of deep sets). *Let $\mathcal{X} := \mathbb{R}^d$ and $(m^{(1)}, y^{(1)}), \ldots, (m^{(N)}, y^{(N)}) \in \mathbb{N}^{\mathcal{X}} \times [C]$ be a sequence of input-label pairs such that $m^{(1)}, \ldots, m^{(N)}$ satisfy the following three conditions:*

    *1. $m^{(1)}, \ldots, m^{(N)}$ are finite multisets whose cardinalities are at most $M$.*

    *2. $m^{(1)}, \ldots, m^{(N)}$ are distinct.*

    3. $m^{(1)}, \ldots, m^{(N)}$ *are element-wise* $(r, \delta)$-*separated for some* $r \geq 1$, $0 < \delta \leq 1$ (*Assumption A.1*).

*Let* $R := 80M^2 N^5 r\delta^{-1}\pi\sqrt{d}$. *Then, there exists a deep set* $(\tilde{\phi}, \tilde{\rho})$ *with width* 12, *depth*

$$\lesssim \sqrt{N \log N} + \sqrt{\frac{N}{\log N}} \cdot \max\{\log R, \log C\}, \tag{113}$$

*(for the definition of $\lesssim$, see Section 3.1) and bit complexity bounded by*

$$\lesssim \log d + \sqrt{\frac{N}{\log N}} \cdot \max\{\log R, \log C\} \tag{114}$$

*which memorizes the dataset, that is,*

$$(\tilde{\phi}, \tilde{\rho})(m^{(i)}) = \tilde{\rho}\left(\sum_{k=1}^{|m^{(i)}|} \tilde{\phi}(\boldsymbol{x}_k^{(i)})\right) = y^{(i)} \tag{115}$$

*holds for every* $i \in [N]$ *with* $m^{(i)} = \{\{\boldsymbol{x}_1^{(i)}, \ldots, \boldsymbol{x}_{|m^{(i)}|}^{(i)}\}\}$.

*Proof of Theorem D.1.* Let $C_\phi := \lceil 4N^3\sqrt{\pi} \rceil$ and $R_\phi := 20r(NM)^2\delta^{-1}\sqrt{\pi d}$. Then, applying Lemma B.3 readily implies that there exists a neural network $\tilde{\phi} : \mathbb{R}^d \to \mathbb{R}$ with width 12, depth

$$\lesssim \sqrt{N \log N} + \sqrt{\frac{N}{\log N}} \cdot \max\{\log R_\phi, \log C_\phi\}, \tag{116}$$

and bit complexity bounded by

$$\lesssim \log d + \sqrt{\frac{N}{\log N}} \cdot \max\{\log R_\phi, \log C_\phi\} \tag{117}$$

such that $\tilde{\phi}(\boldsymbol{x}) \in [[\lceil 4N^3\sqrt{\pi} \rceil]] \cup \{0\}$ holds for any $\boldsymbol{x} \in \bigcup_{i=1}^{N} \operatorname{supp}(m^{(i)})$, and

$$\sum_{k=1}^{|m^{(1)}|} \tilde{\phi}(\boldsymbol{x}_k^{(1)}), \ldots, \sum_{k=1}^{|m^{(N)}|} \tilde{\phi}(\boldsymbol{x}_k^{(N)}) \tag{118}$$

are $(4MN^3\sqrt{\pi}, 1)$-separated.

Since the correspondence $\sum_{k=1}^{|m^{(i)}|} \tilde{\phi}(\boldsymbol{x}_k^{(i)})$ to the label $y^{(i)}$ is injective, we can consider the memorization of $N$ input-label pairs

$$\left(\sum_{k=1}^{|m^{(1)}|} \tilde{\phi}(\boldsymbol{x}_k^{(1)}), y^{(1)}\right), \ldots, \left(\sum_{k=1}^{|m^{(N)}|} \tilde{\phi}(\boldsymbol{x}_k^{(N)}), y^{(N)}\right) \in \mathbb{R} \times [C] \tag{119}$$

with feed-forward networks. Specifically, let $R_\rho$ be

$$R_\rho := 20 \cdot 4MN^3\sqrt{\pi} \cdot N^2 \cdot 1^{-1} \cdot \sqrt{\pi} = 80MN^5\pi. \tag{120}$$

Then, according to Lemma C.1, we have a feed-forward network $\tilde{\rho} : \mathbb{R} \to \mathbb{R}$ with width 12, depth

$$\lesssim \sqrt{N \log N} + \sqrt{\frac{N}{\log N}} \cdot \max\{\log R_\rho, \log C\}, \tag{121}$$

and bit complexity bounded by

$$\lesssim \sqrt{\frac{N}{\log N}} \cdot \max\{\log R_\rho, \log C\} \tag{122}$$

such that for any $i \in [N]$ we have

$$\tilde{\rho} \left( \sum_{k=1}^{|m^{(i)}|} \tilde{\phi}(\boldsymbol{x}_k^{(i)}) \right) = y^{(i)}, \tag{123}$$

as desired.

**Model complexity.** With the configurations defined above, the deep set $(\tilde{\phi}, \tilde{\rho})$ provably memorizes the dataset. Lastly, we check its model complexities, that is, width, depth and bit complexity.

The width of both $\tilde{\phi}$ and $\tilde{\rho}$ is 12, and thus the width of the deep set $(\tilde{\phi}, \tilde{\rho})$ is also 12. As for depth and bit complexity, we define $R$ by

$$R := 80M^2 N^5 r \delta^{-1} \pi \sqrt{d}. \tag{124}$$

Notice that $R_\phi, C_\phi$ and $R_\rho$ are all upper-bounded by $R$, because of the assumption $r \geq 1$ and $0 < \delta \leq 1$. The depth of the deep set $(\tilde{\phi}, \tilde{\rho})$ is the addition of the depth of each feed-forward network, and thus upper-bounded by

$$\lesssim \sqrt{N \log N} + \sqrt{\frac{N}{\log N}} \cdot \max\{\log R, \log C\}. \tag{125}$$

Likewise, the bit complexity of the deep set $(\phi, \rho)$ is the maximum of the bit complexity of each feed-forward network, which is upper-bounded by

$$\lesssim \log d + \sqrt{\frac{N}{\log N}} \cdot \max\{\log R, \log C\}. \tag{126}$$

$\square$

# E   TRANSFORMERS WITH EMBEDDING LAYER

In this section, we examine the memorization capacity of Transformers equipped with an embedding layer. When considering an embedding layer, input sequences consist of a sequence of token ids, rather than a sequence of word vectors. Mathematically, $N$ input sequences we consider in this section are expressed by $N$ vectors

$$\boldsymbol{x}^{(1)}, \ldots, \boldsymbol{x}^{(N)} \in [\omega]^n, \tag{127}$$

where $\omega$ represents the vocabulary size, i.e., the number of distinct token ids that can occur in the input sequence. Then, the embedding layer $\mathcal{F}^{(\mathrm{EM})} : [\omega]^n \to \mathbb{R}^{m \times n}$ is defined by the token-wise operation

$$\mathcal{F}^{(\mathrm{EM})}(\boldsymbol{x})_k := \boldsymbol{W}^{(\mathrm{EM})} \boldsymbol{e}_{x_k} \in \mathbb{R}^m \quad (k \in [n]), \tag{128}$$

with $\boldsymbol{W}^{(\mathrm{EM})} \in \mathbb{R}^{m \times \omega}$ the embedding matrix used as a lookup table, and $\boldsymbol{e}_{x_k} \in \{0, 1\}^\omega$ one-hot vector with 1 in the $x_k$-th position.

Given the input sequences and the embedding layer defined in this way, we now state the theorem on the memorization capacity of Transformers with the embedding layer. As in Theorem B.1, $\mathcal{F}^{(\mathrm{FF})}$ in the next theorem represents a token-wise feed-forward network of arbitrary depth, and $\mathcal{F}^{(\mathrm{UA})}$ is a uniform attention layer (see eq. (39) for its definition). Remarkably, the number of parameters now depends on $\omega$, which is unavoidable due to the use of the embedding layer.

**Theorem E.1.** *Let* $(\boldsymbol{x}^{(1)}, y^{(1)}), \ldots, (\boldsymbol{x}^{(N)}, y^{(N)}) \in [\omega]^n \times [C]$ *be a sequence of input-label pairs that are consistently labeled, in the sense that for any* $i, j \in [N]$, *we have* $y^{(i)} = y^{(j)}$ *if*

$$x_n^{(i)} = x_n^{(j)} \quad and \quad \boldsymbol{x}^{(i)} = \boldsymbol{x}^{(j)} \text{ up to permutations.} \tag{129}$$

*Let* $R := 200\sqrt{3}n^2 r N^5 \delta^{-1} \omega \pi$. *Then, there exists a Transformer with the embedding layer* $\mathcal{N}^{(\mathrm{EM})} : [\omega]^n \to \mathbb{R}^n$ *with the number of parameters*

$$\lesssim \omega + \sqrt{N \log N} + \sqrt{\frac{N}{\log N}} \cdot \max\{\log R, \log C\}, \tag{130}$$

*and the bit complexity*

$$\lesssim \log \omega + \sqrt{\frac{N}{\log N}} \cdot \max\{\log R, \log C\} \tag{131}$$

*that memorizes the dataset, i.e.,*

$$\mathcal{N}^{(\mathrm{EM})}(\boldsymbol{x}^{(i)})_n = \mathcal{E}_{\mathrm{out}} \circ \mathcal{F}^{(\mathrm{FF})} \circ \mathcal{F}^{(\mathrm{UA})} \circ \mathcal{F}^{(\mathrm{EM})}(\boldsymbol{x}^{(i)})_n \tag{132}$$

*holds for every $i \in [N]$.*

*Proof.* The only difference in the proof from Theorem B.1 is that the role previously performed by the feed-forward network $\mathcal{F}_1^{(\mathrm{FF})}$ is now implemented by the embedding layer $\mathcal{F}^{(\mathrm{EM})}$. Specifically, for each input sequence $\boldsymbol{x}^{(i)}$ with $i \in [N]$, we define its multiset expression $m^{(i)} \in \mathbb{N}^{[\omega]}$ by

$$m^{(i)} : [\omega] \to \mathbb{N}, x \mapsto |\{k \mid x_k^{(i)} = x\}|. \tag{133}$$

The cardinality of $m^{(i)}$ for each $i \in [N]$ is at most $n$, and the consistency on the labels are rephrased as follows: for any $i, j \in [N]$, we have $y^{(i)} = y^{(j)}$ if $x_n^{(i)} = x_n^{(j)}$ and $m^{(i)} = m^{(j)}$ hold.

According to Lemma B.1, there exists a subset $A \subset [\omega]$ with its cardinality at most $\min\{\omega, N\}$ such that $m^{(1)}|_A, \ldots, m_A^{(N)}$ are distinct. Then, by applying Lemma B.2 to $m^{(1)}|_A, \ldots, m_A^{(N)}$, we have a function $f : A \to [\lceil 4N^2\sqrt{\pi} \cdot \min\{\omega, N\}\rceil]$ such that

$$\sum_{x \in \mathrm{supp}(m^{(1)}|_A)} m^{(1)}(x)f(x), \ldots, \sum_{x \in \mathrm{supp}(m^{(N)}|_A)} m^{(N)}(x)f(x) \tag{134}$$

are $(4nN^2\sqrt{\pi} \cdot \min\{\omega, N\}, 1)$-separated. We directly implement the function $f$ in the embedding layer. Namely, we define the embedding matrix $\boldsymbol{W}^{(\mathrm{EM})} \in \mathbb{R}^{3 \times \omega}$ in the embedding layer $\mathcal{F}^{(\mathrm{EM})}$ with $m = 3$ by

$$W_{:,x}^{(\mathrm{EM})} := \begin{cases} (f(x), x, 0)^\top & \text{if } x \in A, \\ (0, x, 0)^\top & \text{otherwise,} \end{cases} \tag{135}$$

for each $x \in [\omega]$. The bit complexity of the embedding layer is at most $\log[\lceil 4N^2\sqrt{\pi}\cdot\min\{\omega, N\}\rceil] + \log \omega$, and since the construction of the remaining parts of the Transformer can be carried out similarly to Theorem B.1, we omit those details here. $\square$

## F   MEMORIZATION CAPACITY WITH LIMITED BIT COMPLEXITY

In this section, we consider how the memorization capacity of networks changes when the number of bits available for each parameter of the network is bounded. The following lemma extends Theorem 6.2 from Vardi et al. (2022), with the only difference that it also explicitly supports additional data points with zero labels.

**Lemma F.1** (Extension of Vardi et al. (2022))**.** *Let $N, V, d, C \in \mathbb{N}$ with $N \leq V$, and $r \geq 1, 0 < \delta \leq 1$. Let $y^{(1)}, \ldots, y^{(N)} \in [C]$ be a set of $N$ labels and $\boldsymbol{x}^{(1)}, \ldots, \boldsymbol{x}^{(V)} \in \mathbb{R}^d$ be a set of $V$ inputs such that $\|\boldsymbol{x}^{(i)}\| \leq r$ for every $i \in [V]$ and $\|\boldsymbol{x}^{(i)} - \boldsymbol{x}^{(j)}\| \geq \delta$ for every $i, j \in [V]$ with $i \neq j$. Denote $R := 20rV^2\delta^{-1}\sqrt{\pi d}$ and let $B \in [\sqrt{N}]$. Then, there exists a neural network $F : \mathbb{R}^d \to \mathbb{R}$ with width 13, depth*

$$\lesssim \frac{N\sqrt{\log B}}{B} + \frac{N}{B\sqrt{\log B}} \cdot \max\{\log R, \log C\}, \tag{136}$$

*(for the definition of $\lesssim$, see Section 3.1) and bit complexity*

$$\lesssim \log d + \frac{B}{\sqrt{\log B}} \cdot \max\{\log R, \log C\} \tag{137}$$

*such that $F(\boldsymbol{x}^{(i)}) = y^{(i)}$ for every $i \in [N]$ and $F(\boldsymbol{x}^{(i)}) = 0$ for every $i \in [V] \setminus [N]$.*

*Proof.* The proof idea is the same as the one for Theorem 6.2 from Vardi et al. (2022): we construct a $\frac{N}{B^2} + 1$ sub-networks $F_1, \ldots, F_{N/B^2+1}$ with width at most 13, and then concatenate those networks to create the final network $F$. For the first sub-network $F_1$, we use the same network as in the proof of Lemma C.1, which projects the inputs $\boldsymbol{x}^{(1)}, \ldots, \boldsymbol{x}^{(N)}, \boldsymbol{x}^{(N+1)}, \ldots, \boldsymbol{x}^{(V)}$ into scalars $x_1, \ldots, x_N, x_{N+1}, \ldots, x_V$ while approximately keeping a distance between them. Next, we partition $x_1, \ldots, x_N$ into $\frac{N}{B^2}$ subsets each containing $B^2$ data points. For each subset $x_{(i-1) \cdot B^2+1}, \ldots, x_{i \cdot B^2}$ ($i \in [\frac{N}{B^2}]$), we use Lemma C.1 with zero labels at other data points $x_1, \ldots, x_{(i-1) \cdot B^2}$ and $x_{i \cdot B^2+1}, \ldots, x_V$ to obtain a sub-networks $\tilde{F}_2, \ldots, \tilde{F}_{N/B^2+1}$ with width 12, depth

$$\lesssim B\sqrt{\log B} + \frac{B}{\sqrt{\log B}} \cdot \max\{\log R, \log C\}, \tag{138}$$

and bit complexity

$$\lesssim \frac{B}{\sqrt{\log B}} \cdot \max\{\log R, \log C\}. \tag{139}$$

Finally, we extend the widths of $\tilde{F}_2, \ldots, \tilde{F}_{N/B^2}$ by one to create sub-networks $F_2, \ldots, F_{N/B^2}$ such that

$$F_i \begin{pmatrix} x \\ r \end{pmatrix} = \begin{pmatrix} x \\ r + \tilde{F}_i(x) \end{pmatrix} \quad (i = 2, \ldots, N/B^2). \tag{140}$$

By concatenating all sub-networks and one projection layer $\phi(x, y) := y$ to obtain the final network $F = \phi \circ F_{N/B^2+1} \circ \cdots \circ F_2 \circ F_1$, whose depth is upper-bounded by

$$\lesssim \frac{N}{B^2} \left( B\sqrt{\log B} + \frac{B}{\sqrt{\log B}} \cdot \max\{\log R, \log C\} \right)$$
$$= \frac{N\sqrt{\log B}}{B} + \frac{N}{B\sqrt{\log B}} \cdot \max\{\log R, \log C\}. \tag{141}$$

For each $i = [N]$, there is a unique index $j \in \{2, \ldots, N/B^2\}$ such that $\tilde{F}_j(x_i) = y^{(i)}$ holds and at the same time we have $\tilde{F}_k(x_i) = 0$ for any $k \in \{2, \ldots, N/B^2\}$ with $k \neq j$. Thus, the output of $F$ for $\boldsymbol{x}^{(i)}$ is calculated as

$$F(\boldsymbol{x}^{(i)}) = \phi \begin{pmatrix} x_i \\ \tilde{F}_j(x_i) \end{pmatrix} = y^{(i)}. \tag{142}$$

On the other hand, for any $i \in \{N+1, \ldots, V\}$ and $j \in \{2, \ldots, N/B^2\}$, the output of $\tilde{F}_j(x_i)$ is always zero, which implies $F(\boldsymbol{x}^{(i)}) = 0$ as desired. $\qquad \square$

By replacing feed-forward networks used in the proof of Theorem 4.1 with Lemma F.1, we obtain the upper bound on the memorization capacity of Transformers with limited bit complexity. Notably, the following theorem shows that a Transformer with $\tilde{O}(N^{1-\epsilon})$ parameters can memorize $N$ data points in the next-token prediction setting when each parameter is restricted to $\tilde{O}(N^\epsilon)$ bits for some $\epsilon \in [0, 1/2]$, under the condition that $n, C, r\delta^{-1} = N^{O(1)}$ and $d = \tilde{O}(N^{1-\epsilon})$ as $N \to \infty$.

**Theorem F.1** (Next-token prediction with limited bits). *Let* $(\boldsymbol{X}^{(1)}, y^{(1)}), \ldots, (\boldsymbol{X}^{(N)}, y^{(N)}) \in \mathbb{R}^{d \times n} \times [C]$ *be a sequence of input-label pairs such that*

1. *$(\boldsymbol{X}^{(1)}, y^{(1)}), \ldots, (\boldsymbol{X}^{(N)}, y^{(N)})$ are consistently labeled, in the sense that for any $i, j \in [N]$, we have $y^{(i)} = y^{(j)}$ if*

$$\boldsymbol{X}_{:,n}^{(i)} = \boldsymbol{X}_{:,n}^{(j)} \quad \text{and} \quad \boldsymbol{X}^{(i)} = \boldsymbol{X}^{(j)} \text{ up to permutations.} \tag{143}$$

2. *$\boldsymbol{X}^{(1)}, \ldots, \boldsymbol{X}^{(N)}$ are token-wise $(r, \delta)$-separated for some $r \geq 1$ and $0 < \delta \leq 1$.*

*Let $R := 400\sqrt{3d}n^3rN^5\delta^{-1}\pi$ and $B \in [\sqrt{N}]$. Then, there exists a Transformer $\mathcal{N} : \mathbb{R}^{d\times n} \to \mathbb{R}^n$ with width $15$, depth*

$$\lesssim \frac{N\sqrt{\log B}}{B} + \frac{N}{B\sqrt{\log B}} \cdot \max\{\log R, \log C\}, \tag{144}$$

*(for the definition of $\lesssim$, see Section 3.1) and bit complexity bounded by*

$$\lesssim \log d + \frac{B}{\sqrt{\log B}} \cdot \max\{\log R, \log C\} \tag{145}$$

*that memorizes the dataset, i.e.,*

$$\mathcal{N}\left(\boldsymbol{X}^{(i)}\right)_n = \mathcal{E}_{\text{out}} \circ \overline{\mathcal{F}}_2^{(\text{FF})} \circ \mathcal{F}^{(\text{UA})} \circ \overline{\mathcal{F}}_1^{(\text{FF})} \circ \mathcal{E}_{\text{in}}\left(\boldsymbol{X}^{(i)}\right)_n = y^{(i)} \tag{146}$$

*holds for every $i \in [N]$.*

*Proof.* The proof goes basically the same as is done in the proof of Theorem 4.1, but this time feed-forward networks with limited bit complexity (Lemma F.1) replace two token-wise feed-forward networks in the proof of Theorem 4.1, namely, $\mathcal{F}_1^{(\text{FF})}$ and $\mathcal{F}_2^{(\text{FF})}$.

The first token-wise feed-forward network $\mathcal{F}_1^{(\text{FF})}$ defined by eq. (51) is essentially composed of $\tilde{\phi} : \mathbb{R}^d \to \mathbb{R}$ obtained from Lemma B.3, which in turn is constructed from $\phi : \mathbb{R}^d \to \mathbb{R}$ defined by eq. (32) using Lemma C.1. Therefore, let us consider representing $\phi$ with a feed-forward network with limited bit complexity using Lemma F.1, and the resulting network $\tilde{\phi} : \mathbb{R}^d \to \mathbb{R}$ has width 13, depth

$$\lesssim \frac{N\sqrt{\log B}}{B} + \frac{N}{B\sqrt{\log B}} \cdot \max\{\log R_\phi, \log C_\phi\}, \tag{147}$$

where $C_\phi := \lceil 4N^3\sqrt{\pi} \rceil$ and $R_\phi := 20r(NM)^2\delta^{-1}\sqrt{\pi d}$, and bit complexity

$$\lesssim \log d + \frac{B}{\sqrt{\log B}} \cdot \max\{\log R_\phi, \log C_\phi\}. \tag{148}$$

Then, the first token-wise feed-forward network $\overline{\mathcal{F}}_1^{(\text{FF})} : \mathbb{R}^{d\times n} \to \mathbb{R}^{3\times n}$ with limited bit complexity is defined using $\overline{\phi}$ in the same manner as in eq. (51).

Similarly, the second token-wise feed-forward network $\mathcal{F}_2^{(\text{FF})}$ is defined using Lemma C.1 to associate $\boldsymbol{s}_n^{(1)}, \ldots, \boldsymbol{s}_n^{(N)}$ defined by eq. (53) with labels $y^{(1)}, \ldots, y^{(N)}$. Thus, this time we use Lemma F.1 to construct a feed-forward network $\overline{f}_2^{(\text{FF})} : \mathbb{R}^3 \to \mathbb{R}$ with width 13, depth

$$\lesssim \frac{N\sqrt{\log B}}{B} + \frac{N}{B\sqrt{\log B}} \cdot \max\{\log R_2, \log C\}, \tag{149}$$

with $R_2 := 20 \cdot 20rn^2N^3\delta^{-1}\sqrt{\pi d} \cdot N^2 \cdot n \cdot \sqrt{3\pi} = 400\sqrt{3d}n^3rN^5\delta^{-1}\pi$, and bit complexity bounded by

$$\lesssim \frac{B}{\sqrt{\log B}} \cdot \max\{\log R_2, \log C\} \tag{150}$$

such that $\overline{f}_2^{(\text{FF})}(\boldsymbol{s}_n^{(i)}) = y^{(i)}$ for every $i \in [N]$, which induces the second token-wise feed-forward network $\overline{\mathcal{F}}_2^{(\text{FF})} : \mathbb{R}^{3\times n} \to \mathbb{R}^n$ with limited bit complexity. $\qquad\square$

# G  TECHNICAL LEMMAS

This section summarizes various technical lemmas. In this section, $\text{LEN}(n) \in \mathbb{N}$ for any $n \in \mathbb{N}$ represents the number of bits in its binary representation.

**Lemma G.1** (Park et al. (2021)). *Let $d \in \mathbb{N}$. Then, for any finite subset $\mathcal{X} \subset \mathbb{R}^d$, there exists a unit vector $\boldsymbol{v} \in \mathbb{R}^d$ such that*

$$\frac{1}{|\mathcal{X}|^2} \sqrt{\frac{8}{\pi d}} \|\boldsymbol{x} - \boldsymbol{x}'\|_2 \leq \left| \boldsymbol{v}^\top (\boldsymbol{x} - \boldsymbol{x}') \right| \leq \|\boldsymbol{x} - \boldsymbol{x}'\|_2 \tag{151}$$

*holds for any $\boldsymbol{x}, \boldsymbol{x}' \in \mathcal{X}$.*

**Lemma G.2** (Vardi et al. (2022)). *Let $a, b \in \mathbb{N}$ with $a < b$. Then, there exists a neural network $F$ with depth $2$, width $2$ and bit complexity $\mathrm{LEN}(b)$ such that $F(x) = 1$ for $x \in [a, b]$ and $F(x) = 0$ for $x > b + \frac{1}{2}$ or $x < a - \frac{1}{2}$.*

**Lemma G.3.** *Let $x_1 < \cdots < x_N < R$ with $R > 0$ and $|x_i - x_j| \geq 2$ for every $i, j \in [N]$ with $i \neq j$. Let $m \in \mathbb{N}$ with $m < N$ and let $w_1, \ldots, w_m \in \mathbb{N}$ where $\mathrm{LEN}(w_j) \leq b$ for every $j \in [m]$. Let $k := \lceil \frac{N}{m} \rceil$. Then, there exists a neural network $F : \mathbb{R} \to \mathbb{R}$ with width $4$, depth $3m + 2$ and bit complexity $b + \lceil \log R \rceil$ such that $F$ satisfies*

  1. *for every $i \in [N]$, $F(x_i) = w_{\lceil \frac{i}{k} \rceil}$,*

  2. *for every $x \in \mathbb{R}$ with $|x - x_i| \geq 2$ for all $i \in [N]$, the output $F(x)$ is either $0$ or $w_j$ for some $j \in [m]$.*

*Proof of Lemma G.3.* Most of the proof is the same as in Lemma A.4. from Vardi et al. (2022), and the only difference is that we now examine how the function behaves outside of $x_1, \ldots, x_N$.

For any $j \in [m]$, we use Lemma G.2 with $a = \lfloor x_{j \cdot k - k + 1} \rfloor$ and $b = \lfloor x_{j \cdot k} + 1 \rfloor$ to construct a feed-forward network $\tilde{F}_j : \mathbb{R} \to \mathbb{R}$ such that $\tilde{F}_j(x) = 1$ for any $x \in [\lfloor x_{j \cdot k - k + 1} \rfloor, \lfloor x_{j \cdot k} + 1 \rfloor]$, and $\tilde{F}_j(x) = 0$ for any $x > \lfloor x_{j \cdot k} + 1 \rfloor + \frac{1}{2}$ or $x < \lfloor x_{j \cdot k - k + 1} \rfloor - \frac{1}{2}$. In particular, this means that $\tilde{F}_j(x_i) = 1$ for any $i \in [j \cdot k - k + 1, j \cdot k]$. Here $j \cdot k$ may become bigger than $N$, and in such a case $j \cdot k$ is replaced with $N$. Then, we define a feed-forward network $F_j : \mathbb{R} \to \mathbb{R}$ by

$$F_j \left( \begin{pmatrix} x \\ y \end{pmatrix} \right) := \begin{pmatrix} x \\ y + w_j \cdot \tilde{F}_j(x) \end{pmatrix}, \tag{152}$$

and the whole network $F : \mathbb{R} \to \mathbb{R}$ by $F(x) = \begin{pmatrix} 0 \\ 1 \end{pmatrix}^\top F_m \circ \cdots \circ F_1 \left( \begin{pmatrix} x \\ 0 \end{pmatrix} \right)$. For the verification of the correct behavior of the function $F$ for inputs $x_1, \ldots, x_N$, and the analysis of its model complexity, we refer the reader to the proof by Vardi et al. (2022). Instead, we check the output of $F$ for inputs outside of $x_i$ with $i = 1, \ldots, N$. For any input $x \in \mathbb{R}$ such that $|x - x_i| \geq 2$ for all $i \in [N]$, there are two situations.

  1. $x \in [\lfloor x_{j \cdot k - k + 1} \rfloor, \lfloor x_{j \cdot k} + 1 \rfloor]$ for some $j \in [m]$: in this case, only $\tilde{F}_j(x)$ outputs $1$, and other sub-network $\tilde{F}_{j'}(x)$ with $j' \neq j$ output $0$, which results in $F(x) = w_j$.

  2. for any $j \in [m]$, $x > \lfloor x_{j \cdot k} + 1 \rfloor + \frac{1}{2}$ or $x < \lfloor x_{j \cdot k - k + 1} \rfloor - \frac{1}{2}$ holds: in this case, $\tilde{F}_j(x) = 0$ for $j \in [m]$ and thus $F(x) = 0$.

Putting the above two cases together, we see that the output $F(x)$ for every $x \in \mathbb{R}$ with $|x - x_i| \geq 2$ ($\forall i = 1, \ldots, N$) is $0$ or $w_j$ for some $j \in [m]$. □

**Lemma G.4** (Vardi et al. (2022)). *Let $n \in \mathbb{N}$ and let $i, j \in \mathbb{N}$ with $i < j \leq n$. Denote Telgarsky's triangle function by $\psi(z) := \sigma_R(\sigma_R(2z) - \sigma_R(4z - 2))$. Then, there exists a neural network $F : \mathbb{R}^2 \to \mathbb{R}^3$ with width $5$, depth $3(j - i + 1)$, and bit complexity $n + 2$, such that for any $x \in \mathbb{N}$ with $\mathrm{LEN}(x) \leq n$, if the input of $F$ is $\begin{pmatrix} \psi^{(i-1)} \left( \frac{x}{2^n} + \frac{1}{2^{n+1}} \right) \\ \psi^{(i-1)} \left( \frac{x}{2^n} + \frac{1}{2^{n+2}} \right) \end{pmatrix}$, then it outputs:*

$$\begin{pmatrix} \psi^{(j)} \left( \frac{x}{2^n} + \frac{1}{2^{n+1}} \right) \\ \psi^{(j)} \left( \frac{x}{2^n} + \frac{1}{2^{n+2}} \right) \\ \mathrm{BIN}_{i:j}(x) \end{pmatrix}.$$

**Lemma G.5** (Vardi et al. (2022)). *There exists a network $F : \mathbb{R}^2 \to \mathbb{R}$ with width 2, depth 2 and bit complexity 2 such that $F\left(\begin{pmatrix} x \\ y \end{pmatrix}\right) = 1$ if $x \in [y, y+1]$ and $F\left(\begin{pmatrix} x \\ y \end{pmatrix}\right) = 0$ if $x > y + \frac{3}{2}$ or $x < y - \frac{1}{2}$.*

The following lemma is an extension of the lemma by Vardi et al. (2022), in that the outputs for unexpected inputs are also considered.

**Lemma G.6.** *Let $\rho, n, c \in \mathbb{N}$. Let $u \in \mathbb{N}$ with $\mathrm{LEN}(u) = \rho \cdot n$ and let $w \in \mathbb{N}$ with $\mathrm{LEN}(w) = c \cdot n$. Assume that for any $\ell, k \in \{0, 1, \ldots, n-1\}$ with $\ell \neq k$ we have that $|\mathrm{BIN}_{\rho \cdot \ell + 1 : \rho \cdot (\ell+1)}(u) - \mathrm{BIN}_{\rho \cdot k + 1 : \rho \cdot (k+1)}(u)| \geq 2$. Then, there exists a network $F : \mathbb{R}^3 \to \mathbb{R}$ with width 12, depth $3n \cdot \max\{\rho, c\} + 2n + 2$ and bit complexity $n \cdot \max\{\rho, c\} + 2$, such that for every $x > 0$, if there exist $j \in \{0, 1, \ldots, n-1\}$ where $\lfloor x \rfloor = \mathrm{BIN}_{\rho \cdot j + 1 : \rho \cdot (j+1)}(u)$, then:*

$$F\left(\begin{pmatrix} x \\ w \\ u \end{pmatrix}\right) = \mathrm{BIN}_{\rho \cdot j + 1 : \rho \cdot (j+1)}(w). \tag{153}$$

*In addition, if $x$ satisfies $|x - 1/2 - \mathrm{BIN}_{\rho \cdot j + 1 : \rho \cdot (j+1)}(u)| > 1$ for any $j \in \{0, \ldots, n-1\}$, then*

$$F\left(\begin{pmatrix} x \\ w \\ u \end{pmatrix}\right) = 0. \tag{154}$$

*Proof.* We follow exactly the same construction of a neural network by Vardi et al. (2022). As such, for a detailed analysis of the depth and bit complexity of each network defined here, we refer the reader to the original paper and omit it here.

For each $i = 0, \ldots, n-1$, we construct a network $F_i$ as follows. First, we use Lemma G.4 for $u$ and $w$, respectively to obtain two networks $F_i^w$ and $F_i^u$ with width 5, depth at most $3 \cdot \max\{\rho, c\}$ and bit complexity at most $n \max\{\rho, c\} + 2$ such that

$$F_i^u \begin{pmatrix} \psi^{(i \cdot \rho)}\left(\frac{u}{2^{n \cdot \rho}} + \frac{1}{2^{n \cdot \rho + 1}}\right) \\ \psi^{(i \cdot \rho)}\left(\frac{u}{2^{n \cdot \rho}} + \frac{1}{2^{n \cdot \rho + 2}}\right) \end{pmatrix} = \begin{pmatrix} \psi^{((i+1) \cdot \rho)}\left(\frac{u}{2^{n \cdot \rho}} + \frac{1}{2^{n \cdot \rho + 1}}\right) \\ \psi^{((i+1) \cdot \rho)}\left(\frac{u}{2^{n \cdot \rho}} + \frac{1}{2^{n \cdot \rho + 2}}\right) \\ \mathrm{BIN}_{i \cdot \rho + 1 : (i+1) \cdot \rho}(u) \end{pmatrix}, \tag{155}$$

$$F_i^w \begin{pmatrix} \psi^{(i \cdot c)}\left(\frac{w}{2^{n \cdot c}} + \frac{1}{2^{n \cdot c + 1}}\right) \\ \psi^{(i \cdot c)}\left(\frac{w}{2^{n \cdot c}} + \frac{1}{2^{n \cdot c + 2}}\right) \end{pmatrix} = \begin{pmatrix} \psi^{((i+1) \cdot c)}\left(\frac{w}{2^{n \cdot c}} + \frac{1}{2^{n \cdot c + 1}}\right) \\ \psi^{((i+1) \cdot c)}\left(\frac{w}{2^{n \cdot c}} + \frac{1}{2^{n \cdot c + 2}}\right) \\ \mathrm{BIN}_{i \cdot c + 1 : (i+1) \cdot c}(w) \end{pmatrix}. \tag{156}$$

Next, we use Lemma G.5 with inputs $x$ and $y = \mathrm{BIN}_{i \cdot \rho + 1 : (i+1) \cdot \rho}(u)$ to obtain the neural network $F_i^{\tilde{y}} : \mathbb{R} \to \mathbb{R}$ with width 2, depth 2 and bit complexity at most $\rho$ such that

$$F_i^{\tilde{y}}\left(\begin{pmatrix} x \\ \mathrm{BIN}_{i \cdot \rho + 1 : (i+1) \cdot \rho}(u) \end{pmatrix}\right) = \begin{cases} 1 & \text{if } \mathrm{BIN}_{i \cdot \rho + 1 : (i+1) \cdot \rho}(u) \leq x \leq \mathrm{BIN}_{i \cdot \rho + 1 : (i+1) \cdot \rho}(u) + 1, \\ 0 & \text{if } |x - 1/2 - \mathrm{BIN}_{i \cdot \rho + 1 : (i+1) \cdot \rho}(u)| > 1. \end{cases} \tag{157}$$

In addition, we construct a 1-layer feed-forward network $F_i^y$ by

$$F_i^y \begin{pmatrix} x \\ y \end{pmatrix} := \sigma_R(x \cdot 2^{c+1} - 2^{c+1} + y). \tag{158}$$

Putting the networks defined above and trivial modifications together, we define a neural network $F_i$ such that $F_i$ satisfies

$$
F_i : \begin{pmatrix} x \\ \psi^{(i\cdot\rho)}\left(\frac{u}{2^{n\cdot\rho}} + \frac{1}{2^{n\cdot\rho+1}}\right) \\ \psi^{(i\cdot\rho)}\left(\frac{u}{2^{n\cdot\rho}} + \frac{1}{2^{n\cdot\rho+2}}\right) \\ \psi^{(i\cdot c)}\left(\frac{w}{2^{n\cdot c}} + \frac{1}{2^{n\cdot c+1}}\right) \\ \psi^{(i\cdot c)}\left(\frac{w}{2^{n\cdot c}} + \frac{1}{2^{n\cdot c+2}}\right) \\ y \end{pmatrix}
$$

$$
\mapsto \begin{pmatrix} x \\ \psi^{((i+1)\cdot\rho)}\left(\frac{u}{2^{n\cdot\rho}} + \frac{1}{2^{n\cdot\rho+1}}\right) \\ \psi^{((i+1)\cdot\rho)}\left(\frac{u}{2^{n\cdot\rho}} + \frac{1}{2^{n\cdot\rho+2}}\right) \\ \psi^{((i+1)\cdot c)}\left(\frac{w}{2^{n\cdot c}} + \frac{1}{2^{n\cdot c+1}}\right) \\ \psi^{((i+1)\cdot c)}\left(\frac{w}{2^{n\cdot c}} + \frac{1}{2^{n\cdot c+2}}\right) \\ y + \sigma_R\left(F_i^{\tilde{y}}\left(\begin{pmatrix} x \\ \mathrm{BIN}_{i\cdot\rho+1:(i+1)\cdot\rho}(u) \end{pmatrix}\right)\right) \cdot 2^{c+1} - 2^{c+1} + \mathrm{BIN}_{i\cdot c+1:(i+1)\cdot c}(w) \end{pmatrix} .
$$

$$(159)$$

Finally, we concatenate $F_i$ for each $i = 0, \ldots, n-1$ to construct a network $F : \mathbb{R}^3 \to \mathbb{R}$ by

$$
F := G \circ F_{n-1} \circ \cdots \circ F_0 \circ H \tag{160}
$$

where $G$ and $H$ are additional 1-layer feed-forward networks such that

$$
H : \mathbb{R}^3 \to \mathbb{R}^5, \quad \begin{pmatrix} x \\ w \\ u \end{pmatrix} \mapsto \begin{pmatrix} x \\ \frac{u}{2^{n\cdot\rho}} + \frac{1}{2^{n\cdot\rho+1}} \\ \frac{u}{2^{n\cdot\rho}} + \frac{1}{2^{n\cdot\rho+2}} \\ \frac{w}{2^{n\cdot c}} + \frac{1}{2^{n\cdot c+1}} \\ \frac{w}{2^{n\cdot c}} + \frac{1}{2^{n\cdot c+2}} \\ 0 \end{pmatrix}, \tag{161}
$$

and $G : \mathbb{R}^5 \to \mathbb{R}$ outputs the fifth coordinate of the input. Note that with these configurations, it can be proved by induction that inputs of $F_i$ for each $i = 0, \ldots, n-1$ are always of the form eq. (159).

Hereafter, we verify that the network $F$ actually satisfies the desired behavior. Notice that the output of $F$ is expressed as

$$
F\left(\begin{pmatrix} x \\ w \\ u \end{pmatrix}\right) = \sum_{i=0}^{n-1} \sigma_R\left(F_i^{\tilde{y}}\left(\begin{pmatrix} x \\ \mathrm{BIN}_{i\cdot\rho+1:(i+1)\cdot\rho}(u) \end{pmatrix}\right)\right) \cdot 2^{c+1} - 2^{c+1} + \mathrm{BIN}_{i\cdot c+1:(i+1)\cdot c}(w).
$$

$$(162)$$

If there exist $j \in \{0, 1, \ldots, n-1\}$ with $\lfloor x \rfloor = \mathrm{BIN}_{\rho\cdot j+1:\rho\cdot(j+1)}(u)$, the right-hand side becomes

$$
F\left(\begin{pmatrix} x \\ w \\ u \end{pmatrix}\right) = \sigma_R\left(1 \cdot 2^{c+1} - 2^{c+1} + \mathrm{BIN}_{j\cdot c+1:(j+1)\cdot c}(w)\right)
$$

$$
+ \sum_{\substack{i=0 \\ i\neq j}}^{n-1} \sigma_R\left(0 \cdot 2^{c+1} - 2^{c+1} + \mathrm{BIN}_{i\cdot c+1:(i+1)\cdot c}(w)\right)
$$

$$
= \mathrm{BIN}_{j\cdot c+1:(j+1)\cdot c}(w), \tag{163}
$$

because $\mathrm{BIN}_{i\cdot c+1:(i+1)\cdot c}(w) \leq 2^{c+1}$ holds for any $i = 0, \ldots, n-1$.

On the other hand, if $x$ satisfies $|x - 1/2 - \mathrm{BIN}_{\rho\cdot j+1:\rho\cdot(j+1)}(u)| > 1$ for any $j \in \{0, \ldots, n-1\}$, the output of $F$ becomes

$$
F\left(\begin{pmatrix} x \\ w \\ u \end{pmatrix}\right) = \sum_{i=0}^{n-1} \sigma_R\left(0 \cdot 2^{c+1} - 2^{c+1} + \mathrm{BIN}_{i\cdot c+1:(i+1)\cdot c}(w)\right)
$$

$$
= 0, \tag{164}
$$

as desired. $\qquad\square$

# H EXPERIMENTS

In this section, we empirically investigate whether the memorization capacity of real-world Transformers aligns with the behavior predicted by our theoretical analysis when varying the size of the dataset and the length of input sequences.

## H.1 SETUP

We trained Transformers in the next-token prediction setting on two real-world datasets and one randomly generated dataset of various sizes and evaluated the minimum network size required to memorize each dataset, plotting the results to examine the correlation between dataset size and network size. To validate our theoretical analysis, the architecture of the Transformer used in our experiments followed the same structure as the model described in Theorem B.1. To be more precise, we consider the following architecture:

$$\mathcal{N}\left(\boldsymbol{X}^{(i)}\right)_n = \mathcal{E}_{\text{out}} \circ \mathcal{F}_2^{(\text{FF})} \circ \mathcal{F}^{(\text{UA})} \circ \mathcal{F}_1^{(\text{FF})} \circ \mathcal{E}_{\text{in}}\left(\boldsymbol{X}^{(i)}\right)_n, \tag{165}$$

where $\mathcal{F}_1^{(\text{FF})}$ and $\mathcal{F}_2^{(\text{FF})}$ are token-wise feed-forward layers (eq. (3)) stacked for #blocks blocks with the hidden dimension $q = 4m$ and embedding dimension $m = 2$ [3]. Since the number of parameters in the model is approximately proportional to #blocks, we use #blocks as a proxy for memorization capacity in our experiments by varying it to evaluate the minimum network size required for memorization. The model was trained using the AdamW optimizer (Loshchilov & Hutter, 2019) with full-batch updates. To focus on the representational capacity of models and minimize the influence of optimization, we tuned hyperparameters such as a learning rate and warmup interval using Optuna (Akiba et al., 2019).

## H.2 RESULTS

**Validation of memorization with Transformers using single uniform-attention**: We first validate that a single layer of uniform attention actually suffices for memorization. Specifically, we trained a simplified Transformer defined in eq. (165), consisting of one uniform attention layer and two token-wise feed-forward networks, on two real-world datasets: MultiNLI dataset (Williams et al., 2018) from GLUE benchmark (Wang et al., 2018) and IMDb dataset (Maas et al., 2011). For both datasets, the length of input sequences was truncated to 8, and outputs at the 0-th token were compared with labels using cross-entropy loss. While this setup does not correspond to next-token prediction in the traditional sense, it aligns with the next-token prediction setting considered in this paper.

The results are summarized in figure 1 for MultiNLI dataset and figure 2 for IMDb dataset. Overall, our experiments confirmed that as the number of blocks increases, the training loss can be reduced to nearly zero, and the accuracy tends to approach one. This outcome aligns with the predictions of Theorem B.1, demonstrating that even a single layer of uniform attention, when paired with an appropriate number of feed-forward networks, is sufficient for memorization.

**Varying the dataset size while the sequence length is fixed**: We next examined how the memorization capacity of Transformers changes when varying the dataset size while keeping the sequence length fixed. Specifically, we trained Transformers on datasets sampled from the MultiNLI dataset, where the sequence length was fixed at $n = 8$ and the dataset size $N$ ranged from 600 to 1700 in increments of 100. For each dataset, we determined the minimum number of #blocks required for the network to memorize the data. Here, a network was considered to have successfully memorized the dataset when the training error fell below a threshold of $\epsilon = 0.01$.

Figure 3 summarizes the evaluation of the memorization capacity of Transformers on MultiNLI datasets of varying sizes. From this figure, we can observe the following two points.

1. **Square-root scaling for small datasets**: For smaller dataset sizes, particularly in the range of $N = 600$ to $N = 1400$, the memorization capacity of the Transformer scales approximately as $\sqrt{N}$. This behavior aligns well with the theoretical prediction of Theorem 4.1

---

[3]The embedding dimension was set to 2 so that it becomes difficult for models to memorize even small datasets.

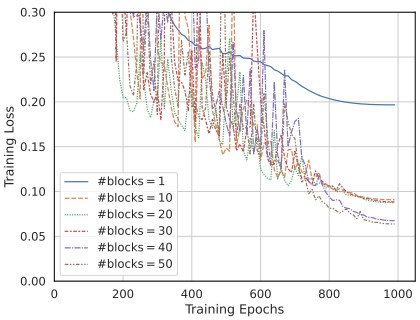 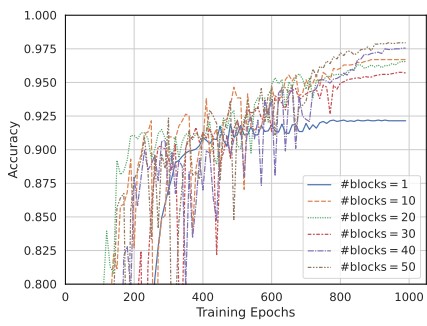

**(a)** Training loss on MultiNLI dataset                    **(b)** Accuracy on MultiNLI dataset

**Figure 1:** Training losses and accuracies of Transformers with #blocks $= 1, 10, 20, 30, 40, 50$ on a dataset of size $N = 2000$ with input sequence length $n = 8$ sampled from MultiNLI dataset. Each model was trained using full-batch gradient descent for 1000 epochs, and the best-performing model was selected after running two trials of hyperparameter tuning with Optuna.

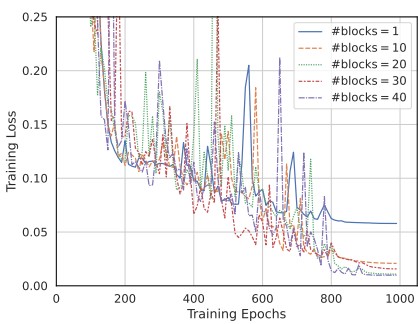 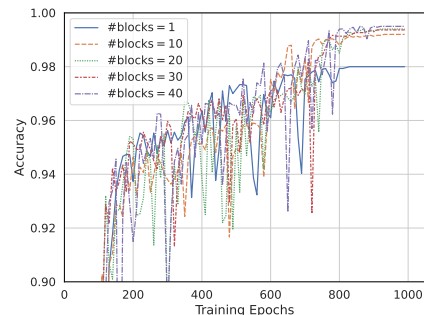

**(a)** Training loss on IMDb dataset                    **(b)** Accuracy on IMDb dataset

**Figure 2:** Training losses and accuracies of Transformers with #blocks $= 1, 10, 20, 30, 40$ on a dataset of size $N = 3000$ with input sequence length $n = 8$ sampled from IMDb dataset. Each model was trained using full-batch gradient descent for 1000 epochs, and the best-performing model was selected after running two trials of hyperparameter tuning with Optuna.

and Theorem 4.2, which suggests that the memorization capacity of Transformers in the next-token prediction setting scales as $\Theta\sqrt{N}$.

2. **Rapid increase for larger datasets**: Beyond $N = 1400$, the memorization capacity exhibits a sharp increase, deviating from the earlier $\sqrt{N}$ scaling. This phenomenon has also been observed in the experiments conducted by Kim et al. (2023). A plausible explanation is that the bit-length of each parameter in the network is fixed during the experiments. As the dataset size grows, the precision of the parameters becomes insufficient for optimal memorization. Under this regime, the analysis of Transformers with limited bit complexity, as discussed in Appendix F, becomes applicable, predicting that the memorization capacity scales linearly with the dataset size $N$.

**Varying the sequence length while the dataset size is fixed**: We also investigated how the memorization capacity changes when the size of a randomly generated dataset is fixed at $N = 500$ and the input sequence length $n$ is varied across $10, 100, 1000$ and $10000$. In this experiment, each word token is a 6-dimensional vector, with each element sampled independently from the uniform distribution over the interval $[0, 1)$. Similarly, each label is either $+1$ or $-1$, sampled independently from the Rademacher distribution. Using the mean squared error as the loss function, a network was considered to have successfully memorized the dataset when the training error fell below a threshold of $\epsilon = 0.01$. Surprisingly, the results showed that, for all sequence lengths, a Transformer with #blocks $= 4$ was the smallest model capable of achieving memorization. An insight from this ex-

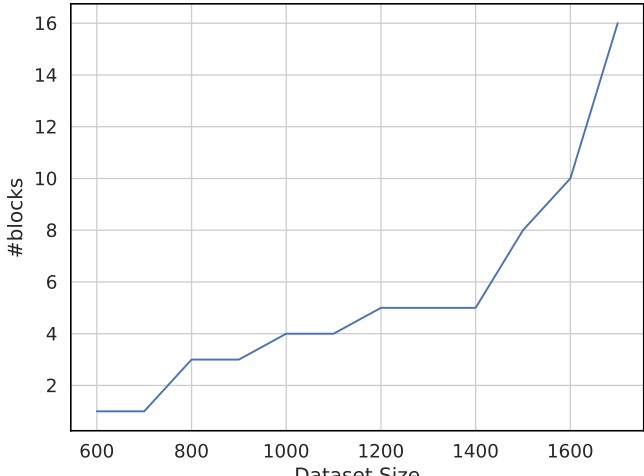

**Figure 3:** Memorization capacity, that is, the minimum size of Transformers required for memorizing MultiNLI dataset with size $N = 600, \ldots, 1700$ in increments of 100. In this figure, the depth #blocks of the two token-wise feed-forward networks $\mathcal{F}_1^{(\mathrm{FF})}$ and $\mathcal{F}_2^{(\mathrm{FF})}$ in eq. (165) is used as the variable on the vertical axis to control the size of the network. Each model was trained using full-batch gradient descent for 1000 epochs, and the best-performing model was selected after running ten trials of hyperparameter tuning with Optuna.

perimental result is that, while the upper bound of the memorization capacity given by Theorem 4.1 has a gap of $O(\log n)$ compared to the lower bound in Theorem 4.2, real-world Transformers appear to align more closely with the lower bound of Theorem 4.2, that is, the memorization capacity might be nearly independent of the input sequence length $n$.

