# OpenReview forum: "On the Optimal Memorization Capacity of Transformers"
_ICLR.cc/2025/Conference — ICLR 2025 Poster_

### Official Review · Reviewer_hFhi · 2024-11-01

**Soundness:** 3
**Presentation:** 2
**Contribution:** 3
**Rating:** 6
**Confidence:** 3

**Summary:**

The paper examines the memorization capacity of Transformers. The authors claim that Transformers can memorize labels with $O(\sqrt{nN})$ parameters in next-token prediction for sequences. They argue that parameter sharing reduces input length sensitivity. In sequence-to-sequence tasks, $ O(\sqrt{nN})$ parameters are shown to be necessary and sufficient for hardmax transformers. While self-attention identifies input sequences effectively, the feed-forward network seems to be a limiting factor.

**Strengths:**

1. The paper addresses the important problem of quantifying the memorization capacity of transformers.
2. The authors present theorems that cover a wide range of tasks for which transformers are used, making the analysis comprehensive.

**Weaknesses:**

1. The sketches of the proofs in the paper were somewhat difficult to follow without referring to the full proofs. Improving the presentation could help a wider audience better understand and appreciate the work.
2. While the paper is theoretical, and experiments may not be essential, including experiments to support the claims would strengthen the work.

**Questions:**

1. I don't completely follow the reasoning behind the statement: "the primary bottleneck is not the contextual mapping of tokens, but rather the feed-forward layers’ capacity to map this token-level contextual information to labels." Clarification on this point would be helpful.

2. If the feed-forward layer is identified as the bottleneck, is there an approach to mitigate this through architectural changes or modifications?

3. Are the data assumptions too restrictive? It feels surprising that the results hold if the attention layer primarily averages, especially when compared to findings in studies such as (a) and (b), despite the differences in setting.

4. Would it be possible to include experiments, even toy examples, to support some of the paper’s claims? Such experiments could significantly enhance the paper's impact.

I am open to increasing the score for the paper based on the authors' responses and feedback from other reviewers.

References:

a) Rajaraman, Nived, et al. "Transformers on Markov data: Constant depth suffices." arXiv preprint arXiv:2407.17686 (2024).

b) Makkuva, Ashok Vardhan, et al. "Local to Global: Learning Dynamics and Effect of Initialization for Transformers." arXiv preprint arXiv:2406.03072 (2024).

---

> ### Author Response · Authors · 2024-11-24
> **Response to Reviewer hFhi**
>
> Thank you for taking the time to review our paper and sharing your insightful comments. We will address your questions below.
>
> > W1. The sketches of the proofs in the paper were somewhat difficult to follow without referring to the full proofs. Improving the presentation could help a wider audience better understand and appreciate the work.
>
> We have added more detailed explanations about the high-dimensional representations $\tilde{X}^{(1)},\dots,\tilde{X}^{(N)}$ used to construct the feed-forward network $\phi$ in the proof sketch of Theorem 4.1, particularly around eq. (9).
> We hope this provides additional clarity and aids in understanding.
> If there are still unclear parts, please let us know, and we would be happy to provide further explanations.
>
>
> > W2. While the paper is theoretical, and experiments may not be essential, including experiments to support the claims would strengthen the work.
>
> In response to your suggestion, we have included three types of experiments in the revised version of the paper to verify our theoretical analysis (details are provided in Appendix H).
> First, we conducted an experiment to verify that the model described in Theorem 4.1 can successfully perform memorization using two real-world datasets.
> Specifically, we trained a Transformer consisting of two feed-forward networks and a single layer of uniform attention in between on MNLI and IMDb datasets.
> The experimental results are summarized in the table below, where $\\#\mathrm{blocks}$ indicates the depth of each of the two token-wise feed-forward networks in the model.
>
> For MultiNLI dataset:
> |               |        |         |         |         |         |         |
> | ------------- | ------ | ------- | ------- | ------- | ------- | ------- |
> | $\\# \mathrm{blocks}$       | 1      | 10      | 20      | 30      | 40      | 50      |
> | Training Loss | 0.1967 | 0.09080 | 0.08744 | 0.08839 | 0.06746 | 0.06378 |
> | Accuracy      | 0.9215 | 0.9670  | 0.9655  | 0.9570  | 0.9755  | 0.9795  |
>
> For IMDb dataset:
> |               |         |         |         |         |          |
> | ------------- | ------- | ------- | ------- | ------- | -------- |
> | $\\# \mathrm{blocks}$       | 1       | 10      | 20      | 30      | 40       |
> | Training Loss | 0.05798 | 0.02088 | 0.01089 | 0.01573 | 0.009921 |
> | Accuracy      | 0.9800  | 0.9920  | 0.9940  | 0.9937  | 0.9950   |
>
> Overall, the results confirm that, given adequately deep feed-forward networks, the training error can be reduced to near zero, which validates that uniform attention actually possesses sufficient representational capacity for memorization.
>
> In addition, we evaluated the behavior of the memorization capacity, that is, the minimum number of $\\#\mathrm{blocks}$ required for memorizing real-world dataset (MultiNLI) as the dataset size varies.
>
> The experimental results are summarized in the table below.
>
> | Dataset Size | 600 | 700 | 800 | 900 | 1000 | 1100 | 1200 | 1300 | 1400 | 1500 | 1600 | 1700 |
> | ------------ | --- | --- | --- | --- | ---- | ---- | ---- | ---- | ---- | ---- | ---- | ---- |
> | $\\#\mathrm{blocks}$ | 1   | 1   | 3   | 3   | 4    | 4    | 5    | 5    | 5    | 8    | 10   | 16   |
>
> The results indicate that for smaller datasets, the memorization capacity scales as $\sqrt{N}$, consistent with the predictions of Theorems 4.1 and 4.2.
> For larger datasets, however, the memorization capacity increases rapidly.
>
> While the influence of optimization in real-world experiments may result in sub-optimal models, a plausible explanation for this phenomenon is that the fixed bit-length of model parameters becomes a bottleneck as the dataset size grows.
> We have added a discussion in Appendix F on the memorization capacity of Transformers in scenarios where the bit-length of parameters is insufficient.
> Specifically, the result from Appendix F suggests that the memorization capacity scales linearly with $N$ when the bit-length of parameters are limited (up to logarithmic factors).
>
> We also trained the model on randomly generated datasets with varying input sequence length and fixed dataset size.
> The dataset size was fixed at $N=500$, and we considered four input sequence lengths: $n=10,100,1000,10000$.
> For all cases, we verified that a Transformer combining a single uniform attention layer with $4$-block feed-forward networks achieved a training error below the threshold $ϵ=0.01$.
> This result not only demonstrates that memorization is possible even with a single uniform attention layer, but also confirms the prediction made in Theorem 4.1 that the memorization capacity in the next-token prediction setting is nearly independent of the input sequence length $n$.
>
> The experiments we have conducted significantly deepened our understanding of the memorization capacity of Transformers.
> It has also allowed us to experimentally confirm the occurrence of the phase transition in memorization capacity.
> Thank you for your valuable feedback!

---

> ### Author Response · Authors · 2024-11-24
> **Response to Reviewer hFhi**
>
> > Q1. I don't completely follow the reasoning behind the statement: "the primary bottleneck is not the contextual mapping of tokens, but rather the feed-forward layers’ capacity to map this token-level contextual information to labels." Clarification on this point would be helpful.
>
> Thank you for pointing this out.
> To provide some background, prior work [1] suggested that self-attention mechanism performs contextual mapping, while feed-forward layers are responsible for the actual memorization, that is, mapping token-level contextual information to labels.
> The statement "the primary bottleneck is..." was written with this context in mind and reflects how our results align with this background.
>
> Here is a more detailed explanation: as shown in Theorem 4.1, the upper bound for the next-token prediction setting already allows for distinguishing between $N$ input sequences using $\tilde{O}(\sqrt{N})$ parameters.
> Given that the ability to distinguish input sequences, in addition to token-level information, implies that the context of each token can also be identified, this implies that contextual mapping (in the sense of Definition 4.1) is already achieved within this parameter bound.
>  The additional $\sqrt{n}$ factor in the seq-to-seq prediction setting reflects the extra effort required to map this contextual information to labels.
> In this sense, we mentioned that the feed-forward network is the bottleneck, as it is primarily responsible for this context-to-label mapping.
>
>
> > Q2. If the feed-forward layer is identified as the bottleneck, is there an approach to mitigate this through architectural changes or modifications?
>
> Addressing the bottleneck in the feed-forward network would essentially require designing a model that performs memorization more efficiently than a feed-forward network.
> At present, this appears to be a challenging task.
> One potential research direction could involve exploring architectures that are better suited to real-world datasets, as memorization capacity is a worst-case analysis.
> For instance, just as mixture of experts can serve as an alternative to pure feed-forward layers, there may be architectures that are more effective for specific data distributions.
> In this context, advancing the study of memorization capacity of feed-forward networks in more developed settings could be a valuable avenue for future research.
>
>
> > Q3. Are the data assumptions too restrictive? It feels surprising that the results hold if the attention layer primarily averages, especially when compared to findings in studies such as (a) and (b), despite the differences in setting.
>
> We would like to note that in the study of memorization capacity, $(r,\delta)$-separatedness is considered a standard assumption [1, 2, 3, 4, 5, 6, 7].
> Since memorization capacity is the notion for the framework of finite dataset, and any finite discrete dataset (for example, any text dataset) is inherently $(r,\delta)$-separated to some extent, this assumption is relatively loose.
>  As a next step, it would indeed be valuable to consider more sophisticated data assumptions in future research.
>
> Regarding the cited work, we believe the differences arise primarily from the problem settings, not the data assumption. The paper (a) investigates how Transformers can represent conditional $k$-th order Markov processes.
> Here, "conditional" refers to computing an empirical $k$-th order Markov distribution in the in-contextual way, given a specific input sequence, and as such this is a distinct problem setting from our work.
> In such settings, uniform attention may indeed be insufficient, and fully leveraging the keys and queries in self-attention could be essential for capturing these contextual relationships.
>
> The paper (b), on the other hand, analyzes Transformers trained on data generated by a first order Markov process.
> Their results effectively disregard self-attention mechanism (this conclusion is natural, considering the fact that a first-order Markov process depends only on the current token).
>
>
> **References**
>
> [1] Kajitsuka and Sato, Are Transformers with One Layer Self-Attention Using Low-Rank Weight Matrices Universal Approximators? ICLR 2024.
>
> [2] Hardt and Ma. Identity Matters in Deep Learning. ICLR 2017.
>
> [3] Vershynin, Memory Capacity of Neural Networks with Threshold and ReLU Activations. arXiv 2020.
>
> [4] Park et al., Provable Memorization via Deep Neural Networks using Sub-linear Parameters. COLT 2021.
>
> [5] Vardi et al., On the Optimal Memorization Capacity of ReLU Neural Networks. ICLR 2021.
>
> [6] Kim et al., Provable Memorization Capacity of Transformers. ICLR 2023.
>
> [7] Siegel, Sharp Lower Bounds on Interpolation by Deep ReLU Neural Networks at Irregularly Spaced Data. arXiv 2024.

---

> > ### Comment · Reviewer_hFhi · 2024-11-26
> >
> > Thank you for your response. I have gone over the responses as well as the reviews of other reviewers and basis this have decided to increase my score.

---

### Official Review · Reviewer_6ZoB · 2024-11-03

**Soundness:** 3
**Presentation:** 3
**Contribution:** 3
**Rating:** 6
**Confidence:** 2

**Summary:**

This paper studied the memorization capacity of transformers in two tasks, namely next-token prediction and sequence-to-sequence prediction. The paper showed that, by construction, a transformer with $\tilde{O}(\sqrt{N}log n)$ layers can memorize $N$ data points of length $n$. The paper further showed that a transformer with $\tilde{O}(\sqrt{nN})$ layers can memorize $N$ data points in the sequence-to-sequence prediction problem. A lower bound is provided to demonstrate the near-optimality of the upper bound.

**Strengths:**

1. The results seem promising and strong, because both upper and lower bounds are provided.
2. The studied problem and method provide interesting insight into transformers.

**Weaknesses:**

1. The title wording could be improved for accuracy, I would suggest 'near-optimal...' instead of 'optimal...' since there is a logarithm term mismatch between lower bound and upper bound.
2. The paper can have more discussion on the separatedness assumption. For example, how realistic is the assumption?

**Questions:**

Following the weaknesses part, I have the following questions.

1.  What is the significance of the logarithmic gap in the paper. For instance, do authors believe this gap can be closed with further analysis, or is it likely to be fundamental?

2.1 Is there any examples of real-world datasets that satisfy or approximately satisfy this assumption?
2.2 Can authors discuss how the results might change if this assumption is relaxed?

---

> ### Author Response · Authors · 2024-11-24
> **Response to Reviewer 6ZoB**
>
> Thank you for taking the time to review our paper and sharing your insightful comments. We will address your questions below.
>
>
> > W1. The title wording could be improved for accuracy, I would suggest 'near-optimal...' instead of 'optimal...' since there is a logarithm term mismatch between lower bound and upper bound.
>
> Thank you for your suggestion.
> Following the title of prior work [1], which established upper and lower bounds on the memorization capacity of feed-forward networks with a logarithmic factor gap, we have revised the title to "On the" Optimal Memorization Capacity of Transformers.
>
>
> > W2. The paper can have more discussion on the separatedness assumption. For example, how realistic is the assumption?
>
> In the study of memorization capacity, the $(r,\delta)$-separatedness is a standard assumption [1, 2, 3, 4, 5, 6, 7], and any finite discrete dataset (for example, any text dataset) is inherently $(r,\delta)$-separated to some degree.
> More specifically, what really matters is the values of $r$ and $\delta$ in the $(r,\delta)$-separatedness.
> Prior work [7] has shown that  when $r\delta^{-1} = \Omega(\exp(N))$, $O(\sqrt{N})$ parameters are no longer sufficient, and $\Omega(N)$ parameters are required for memorization.
> Given this, the assumption $r\delta^{-1} = N^{O(1)}$ in Theorem 4.1 is sufficiently loose to accommodate cover scenarios outside the range identified in prior work [7].
> Since we demonstrated that optimal memorization is achievable under standard $(r,\delta)$-separated datasets in this work, it would indeed be valuable to explore more advanced data assumptions in future research.
>
>
> > Q1. What is the significance of the logarithmic gap in the paper. For instance, do authors believe this gap can be closed with further analysis, or is it likely to be fundamental?
>
>  In our current lower bound analysis, we did not explicitly incorporate the $(r,\delta)$-separatedness of the data into the proof.
> However, we believe that by integrating this assumption into the analysis, it might be possible to eliminate the logarithmic gap.
> In fact, it is already known [7] that when $r\delta^{-1} = \Omega(\exp(N))$, $O(\sqrt{N})$ parameters are not sufficient, and the lower bound on the memorization capacity becomes $\Omega(N)$.
> This suggests that the separation of the data is inherently linked to the lower bound of the memorization capacity.
>
>
> > Q2. Is there any examples of real-world datasets that satisfy or approximately satisfy this assumption? 2.2 Can authors discuss how the results might change if this assumption is relaxed?
>
> Memorization capacity is the notion for the framework of finite datasets, and any finite, discrete dataset (for example, any text dataset) is inherently token-wise separated to some degree.
> We hope this answers your question.
>
> **References**
>
> [1] Vardi et al., On the Optimal Memorization Capacity of ReLU Neural Networks. ICLR 2021.
>
> [2] Hardt and Ma. Identity Matters in Deep Learning. ICLR 2017.
>
> [3] Vershynin, Memory Capacity of Neural Networks with Threshold and ReLU Activations. arXiv 2020.
>
> [4] Park et al., Provable Memorization via Deep Neural Networks using Sub-linear Parameters. COLT 2021.
>
> [5] Kim et al., Provable Memorization Capacity of Transformers. ICLR 2023.
>
> [6] Kajitsuka and Sato, Are Transformers with One Layer Self-Attention Using Low-Rank Weight Matrices Universal Approximators? ICLR 2024.
>
> [7] Siegel, Sharp Lower Bounds on Interpolation by Deep ReLU Neural Networks at Irregularly Spaced Data. arXiv 2024.

---

> > ### Comment · Reviewer_6ZoB · 2024-11-24
> >
> > Thanks for the reply. I have no further question and decide to retain my score.

---

### Official Review · Reviewer_ZZQ4 · 2024-11-03

**Soundness:** 3
**Presentation:** 3
**Contribution:** 2
**Rating:** 8
**Confidence:** 3

**Summary:**

This paper studies the number of parameters and the bit complexity required by a transformer to memorize a dataset of size $N$. In the context of next token prediction, the paper proves that for any dataset with well separated tokens, there exists a transformer with $\tilde O(\sqrt{N})$ parameters+bit-complexity which interpolates it, along with a matching lower bound that $\Omega(\sqrt{N})$ parameters are necessary. In the sequence to sequence setting, the paper proves a similar upper bound with $\tilde O(\sqrt{nN})$ parameters+bit complexity where $n$ is the sequence length, and they prove a matching lower bound for hardmax transformers.

**Strengths:**

- The proof sketch of Theorem 4.1 is clear and well motivated
- In the next-token setting, the paper proves matching upper and lower bounds, and in the sequence to sequence setting they match for hardmax transformers
- The authors provide clear comparisons with prior work throughout the paper

**Weaknesses:**

- **bit complexity:** As the authors point out, the transformer needs at least $\Omega(N)$ bits, so to succeed with $\sqrt{N}$ parameters it needs a bit complexity of $\sqrt{N}$. However, this is very unrealistic as these models are trained in finite precisions, most often 16 bit precision. It is therefore unclear if these results say much about transformers ability to memorize in practice. This also affects the motivation, for example I would expected double descent (lines 102-107) to occur at $\Theta(N)$ parameters, not $\Theta(\sqrt{N})$ parameters.

- **experiments:** Related to the above point, the paper would benefit from some simple experiments that demonstrate whether such constructions are learnable or whether the empirical limit is $\Theta(N)$ or $\Theta(\sqrt{N})$ parameters. It could also be interesting to test this with various levels of precisions (FP16/32/64) to see whether higher precision really does let the transformer memorize with fewer parameters.

- minor typo on line 462: viewd -> viewed

**Questions:**

- If I understand correctly, neither of the setting in the paper correspond directly to the standard next-token prediction training paradigm in which the transformer needs to simultaneously predict the $i$th token given $X_{:,< i}$ for all $i$. Corollary 4.1 does not directly apply because of the varying sequence lengths, but it suggests a construction with $\tilde O(N)$ parameters+bit complexity. However, it's not clear if this is optimal since Theorem 4.3 is lower bound against the more general sequence-to-sequence task?

- Could you explain footnote 2 (line 1025)? I assume it's not possible to remove the $\log(r/\delta)$ dependence so is the goal to remove the $\log d$ or $\log C$ dependencies? If so, what is the idea?

- Is it possible to achieve the full tradeoff between bit-complexity and number of parameters? E.g. $N^{\alpha}$ parameters and $N^{1-\alpha}$ bit complexity for any $\alpha \in [1/2,1]$?

---

> ### Author Response · Authors · 2024-11-24
> **Response to Reviewer ZZQ4**
>
> Thank you for taking the time to review our paper and sharing your insightful comments. We will address your questions below.
>
> > W1. **bit complexity**: As the authors point out, the transformer needs at least $\Omega(N)$ bits, so to succeed with $\sqrt{N}$ parameters it needs a bit complexity of $\sqrt{N}$.
> However, this is very unrealistic as these models are trained in finite precisions, most often 16 bit precision.
> It is therefore unclear if these results say much about transformers ability to memorize in practice.
> This also affects the motivation, for example I would expected double descent (lines 102-107) to occur at $\Theta(N)$ parameters, not $\Theta(\sqrt{N})$ parameters.
>
> We have included an analysis of the memorization capacity of Transformers with limited bit complexity in Appendix F.
> As you mentioned in your question 3, Theorem F.1 demonstrates that a Transformer constrained to a bit complexity of $\tilde{O}(N^{1 - \alpha})$ for some $\alpha \in [1/2,1]$ can memorize $N$ data points in the next-token prediction setting using $\tilde{O}(N^{\alpha})$ parameters.
>
>
> > W2. **experiments**: Related to the above point, the paper would benefit from some simple experiments that demonstrate whether such constructions are learnable or whether the empirical limit is $\Theta(N)$ or $\Theta(\sqrt{N})$ parameters.
> It could also be interesting to test this with various levels of precisions (FP16/32/64) to see whether higher precision really does let the transformer memorize with fewer parameters.
>
> In response to your feedback, we evaluated the behavior of the memorization capacity, that is, the minimum size of Transformers required for memorizing real-world dataset (MultiNLI) as the dataset size varies (details are provided in Appendix H).
> The experimental results are summarized in the table below, where $\\#\mathrm{blocks}$ indicates the depth of each of the two token-wise feed-forward networks in the model and was used as a proxy for the model size.
>
> | Dataset Size | 600 | 700 | 800 | 900 | 1000 | 1100 | 1200 | 1300 | 1400 | 1500 | 1600 | 1700 |
> | ------------ | --- | --- | --- | --- | ---- | ---- | ---- | ---- | ---- | ---- | ---- | ---- |
> | $\\#\mathrm{blocks}$ | 1   | 1   | 3   | 3   | 4    | 4    | 5    | 5    | 5    | 8    | 10   | 16   |
>
> Overall, the results indicate that for smaller datasets, the memorization capacity scales as $\sqrt{N}$, consistent with the predictions of Theorems 4.1 and 4.2.
> For larger datasets, however, the memorization capacity increases rapidly.
>
> While the influence of optimization in real-world experiments may result in sub-optimal models, a plausible explanation for this phenomenon, as you suggested, is that the fixed bit-length of model parameters becomes a bottleneck as the dataset size grows.
> In such a regime, the results from Appendix F on Transformers with limited bit complexity suggest that the memorization capacity scales linearly with $N$.
>
> Your feedback has greatly enhanced our understanding of the memorization capacity in real-world models.
> It has also allowed us to experimentally confirm the occurrence of this phase transition in memorization capacity, thereby broadening the scope of this work.
> We sincerely appreciate your insightful comment.
>
>
> > W3. minor typo on line 462: viewd -> viewed
>
> Thank you for pointing this out.
> We have corrected the typo in the revised paper.

---

> > ### Author Response · Authors · 2024-11-24
> > **Response to Reviewer ZZQ4**
> >
> > > Q1. If I understand correctly, neither of the setting in the paper correspond directly to the standard next-token prediction training paradigm in which the transformer needs to simultaneously predict the $i$-th token given $X_{:,<i}$ for all $i$.
> > Corollary 4.1 does not directly apply because of the varying sequence lengths, but it suggests a construction with $\tilde{O}(N)$ parameters+bit complexity.
> > However, it's not clear if this is optimal since Theorem 4.3 is lower bound against the more general sequence-to-sequence task?
> >
> >  You are correct that neither the next-token prediction setting nor the seq-to-seq prediction setting considered in this work directly supports datasets with varying input sequence lengths.
> > This is because our current formulation does not assume positional encoding, which allows Transformers to inherently distinguish between inputs of different lengths.
> > For example, a Transformer without positional encoding cannot distinguish between sequences such as [a,a,a] and [a,a,a,a,a,a].
> > However, as is commonly done when deploying Transformers in practice, padding inputs to the same sequence length would allow our theoretical analysis to be directly applicable in these cases.
> >
> > Regarding the second part of your question, we would like to clarify: were you perhaps referring to Theorem 4.1 instead of Corollary 4.1 in your comment?
> > In the structure of this paper, Theorems 4.1 and 4.2 correspond to the upper and lower bounds on the memorization capacity in the next-token prediction task, while Corollary 4.1 and Theorem 4.3 address the upper and lower bounds for the seq-to-seq prediction task.
> > Therefore, Corollary 4.1 and Theorem 4.3 are based on the same problem setting, and Theorem 4.3 does not provide a lower bound against a more general seq-to-seq task.
> >
> >
> > > Q2. Could you explain footnote 2 (line 1025)? I assume it's not possible to remove the $\log(r/\delta)$ dependence so is the goal to remove the $\log d$ or $\log C$ dependencies?
> > If so, what is the idea?
> >
> > We apologize for the confusing explanation in footnote 2.
> > The intended meaning is that "Theorem 4.1 currently has an upper bound of the form $\sqrt{N \log N} \cdot \log n$, but it can be improved to $\sqrt{N \log (nN)}$."
> > We have revised the wording in footnote 2 to make this clarification more apparent in the updated paper.
> >
> >
> > > W3. Is it possible to achieve the full tradeoff between bit-complexity and number of parameters? E.g. $N^\alpha$ parameters and $N^{1-\alpha}$ bit complexity for any $\alpha \in [1/2,1]$?
> >
> > We have included an analysis of the memorization capacity of Transformers with limited bit complexity in Appendix F, in which Theorem F.1 states exactly what you expects: for a Transformer with $\tilde{O}(N^{1-\alpha})$ bit complexity to memorize $N$ data points for some $\alpha \in [1/2,1]$, $\tilde{O}(N^{\alpha})$ parameters are sufficient.
> > Thank you for your insightful question!

---

> > > ### Comment · Reviewer_ZZQ4 · 2024-11-27
> > >
> > > I thank the authors for addressing my comments. In particular, Figure 3 in Appendix H is interesting and does support the main message of this paper (although a full empirical justification would also require testing whether the jump from $\sqrt{N}$ scaling to $N$ scaling is affected by the level of precision, e.g. fp16/32/64). In addition, Appendix F is nice and shows that given that $dB \ge N$ is necessary, implies that it is possible to achieve this tradeoff for any $B \le \sqrt{N}$. These contributions address my main concerns with the paper and I have raised my score accordingly.
> > >
> > > Regarding the autoregressive setting, I indeed meant to refer to Theorem 4.1. I was referencing the fact that this setting can be mimicked by truncating the sequence at different lengths, and casting it as $nN$ independent classification problems to which Theorem 4.1 could be applied. This does not exactly work since the resulting sequences would not all have the same length, but it should still be true. I was curious if in this setting the optimal memorization capacity would scale with $\sqrt{N}$ or if you have to pay $\sqrt{nN}$, but this can be left to future work.
> > >
> > > Regarding footnote 2, placing a claim like that in a footnote merits further discussion. If the extension is trivial, the authors should update the upper bound in Theorem 4.1, and if it requires additional effort, the footnote should one or two
> > > sentences sketching these calculations at a high level to justify the claim. However, this is a relatively minor point.

---

> > > > ### Author Response · Authors · 2024-11-28
> > > > **Response to Reviewer ZZQ4**
> > > >
> > > > We are delighted to hear that your concerns have been addressed.
> > > > Your feedback has helped us gain new insights into Transformers with limited bit complexity, and we are deeply grateful for your thoughtful comments.
> > > > > Regarding the autoregressive setting, ...
> > > >
> > > > Covering datasets with varying input lengths is indeed an important avenue for future work.
> > > > Generally speaking, the number of parameters required for memorization depend on the degrees of freedom in the dataset’s labels.
> > > > While we cannot assert this definitively for now, we anticipate the following: For a dataset comprising $N$ input sequences of varying lengths,
> > > > - the required order of parameters in the next-token prediction setting would likely remain of order $\sqrt{N}$.
> > > > - In contrast, for the seq-to-seq prediction setting, we expect the required order to depend on the square root of the total sum of the sequence lengths across all $N$ input sequences, i.e., the total number of tokens in the dataset.
> > > >
> > > > > Regarding footnote 2, ...
> > > >
> > > > Thank you for pointing this out.
> > > > We have revised Footnote 2 to include a high-level outline of the proof. The sketch is as follows:
> > > > In Lemma C.1, which constructs a feed-forward network for optimal memorization, Stage II involves partitioning the $N$ data points into $\sqrt{N \log N}$ subsets, each containing $\sqrt{\frac{N}{\log N}}$ elements.
> > > > By modifying this step to partition the data points into $\sqrt{N \log (nN)}$ subsets, each containing $\sqrt{\frac{N}{\log (nN)}}$ elements, the bound in Theorem B.1 is updated to $O(\sqrt{N \log (nN)})$.
> > > >
> > > > Thank you again for your insightful feedback, which has helped us refine our work.

---

### Official Review · Reviewer_VZKs · 2024-11-04

**Soundness:** 3
**Presentation:** 3
**Contribution:** 2
**Rating:** 6
**Confidence:** 4

**Summary:**

The paper focuses on the memorization capacity of Transformers in next-token prediction and sequence-to-sequence settings, and shows the scaling in the number of input sequences and their lengths.

**Strengths:**

- This paper is well-motivated and clearly presented, making it easy to follow.
- This paper is technically solid; the construction of a single-layer Transformer that can identify identical sequence ids from token-wise \((r,\delta)\)-separated sequences with near-optimal parameter order is novel. Additionally, the authors provide a parameter lower bound for the seq2seq case when the self-attention layer uses hard attention.
- Furthermore, the paper considers bit complexity, which is often overlooked in related work.

**Weaknesses:**

- Token-wise \((r,\delta)\)-separated sequences may not be an expressive enough model to fully capture the memorization capacity of Transformers. It appears that the authors construct a Transformer that captures contextual information by 'counting' the occurrences of tokens in a subset. However, Transformers typically capture contextual information from the mutual relationships among tokens via the attention mechanism.
- In the proof of Theorem 4.1, the attention layer is set to uniform attention, functioning as a column-wise average operator. This over-simplification of the attention mechanism limits the ability of the work to reveal how attention contributes to memorization, as the attention mechanism should gather context-aware representations of tokens based on their relationships with others in the sequence. Therefore, how do the authors understand the role that the attention mechanism plays in the memorization capacity of the Transformer within their framework?
 - The proof for the parameter lower bound in seq2seq cases relies on hardmax activation, deviating from the real-world scenario where the attention layer adopts softmax activation. Is it possible to extend the proof to the case with softmax activation?

**Questions:**

See weaknesses above.

---

> ### Author Response · Authors · 2024-11-24
> **Response to Reviewer VZKs**
>
> Thank you for taking the time to review our paper and sharing your insightful comments. We will address your questions below.
>
> > W1. Token-wise $(r,\delta)$-separated sequences may not be an expressive enough model to fully capture the memorization capacity of Transformers.
> It appears that the authors construct a Transformer that captures contextual information by 'counting' the occurrences of tokens in a subset.
> However, Transformers typically capture contextual information from the mutual relationships among tokens via the attention mechanism.
>
> Regarding the use of $(r,\epsilon)$-separated sequences, we note that this is a standard assumption in the study of memorization capacity [1, 2, 3, 4, 5, 6, 7].
> We understand the reviewer’s point about Transformers capturing contextual information through the mutual relationships among tokens via the self-attention mechanism.
> One of the main messages of our work is that to fully understand how keys and queries are utilized in real-world self-attention, it would be better to analyze Transformers from the perspectives of optimization or generalization, rather than solely focusing on memorization capacity.
>
>
> > W2. In the proof of Theorem 4.1, the attention layer is set to uniform attention, functioning as a column-wise average operator.
> This over-simplification of the attention mechanism limits the ability of the work to reveal how attention contributes to memorization, as the attention mechanism should gather context-aware representations of tokens based on their relationships with others in the sequence.
> Therefore, how do the authors understand the role that the attention mechanism plays in the memorization capacity of the Transformer within their framework?
>
> We would like to clarify that we did not oversimplify the problem setting by replacing self-attention with uniform attention.
> Rather, our analysis of the memorization capacity of a Transformer with general version of self-attention revealed that uniform attention already achieves the sufficient representational capacity.
> Moreover, prior research [8, 9] has indicated that even with uniform attention, Transformers can achieve sufficient accuracy for certain classification tasks.
> In this sense, our work provides theoretical support for these empirically observed phenomena, reinforcing the understanding of uniform attention's effectiveness in specific contexts, including memorization.
>
> In response to your feedback, we conducted experiments to demonstrate that the model described in Theorem 4.1 can indeed achieve memorization on two real-world datasets (details are provided in Appendix H).
> Specifically, we trained a Transformer consisting of a single uniform attention layer and two token-wise feed-forward networks on MultiNLI dataset and IMDb dataset.
> The experimental results are summarized in the table below, where $\\# \mathrm{blocks}$ indicates the depth of each of the two token-wise feed-forward networks in the model.
>
> For MultiNLI dataset:
> |               |        |         |         |         |         |         |
> | ------------- | ------ | ------- | ------- | ------- | ------- | ------- |
> | $\\# \mathrm{blocks}$       | 1      | 10      | 20      | 30      | 40      | 50      |
> | Training Loss | 0.1967 | 0.09080 | 0.08744 | 0.08839 | 0.06746 | 0.06378 |
> | Accuracy      | 0.9215 | 0.9670  | 0.9655  | 0.9570  | 0.9755  | 0.9795  |
>
> For IMDb dataset:
> |               |         |         |         |         |          |
> | ------------- | ------- | ------- | ------- | ------- | -------- |
> | $\\# \mathrm{blocks}$       | 1       | 10      | 20      | 30      | 40       |
> | Training Loss | 0.05798 | 0.02088 | 0.01089 | 0.01573 | 0.009921 |
> | Accuracy      | 0.9800  | 0.9920  | 0.9940  | 0.9937  | 0.9950   |
>
> Overall, the results confirm that, given adequately deep feed-forward networks, the training error can be reduced to near zero, which validates that uniform attention actually possesses sufficient representational capacity for memorization.
>
>
> **References**
>
> [1] Hardt and Ma. Identity Matters in Deep Learning. ICLR 2017.
>
> [2] Vershynin, Memory Capacity of Neural Networks with Threshold and ReLU Activations. arXiv 2020.
>
> [3] Park et al., Provable Memorization via Deep Neural Networks using Sub-linear Parameters. COLT 2021.
>
> [4] Vardi et al., On the Optimal Memorization Capacity of ReLU Neural Networks. ICLR 2021.
>
> [5] Kim et al., Provable Memorization Capacity of Transformers. ICLR 2023.
>
> [6] Kajitsuka and Sato, Are Transformers with One Layer Self-Attention Using Low-Rank Weight Matrices Universal Approximators? ICLR 2024.
>
> [7] Siegel, Sharp Lower Bounds on Interpolation by Deep ReLU Neural Networks at Irregularly Spaced Data. arXiv 2024.
>
> [8] Yu et al., MetaFormer Is Actually What You Need for Vision. CVPR 2022.
>
> [9] Hyeon-Woo et al., Scratching Visual Transformer's Back with Uniform Attention. ICCV 2023.

---

> > ### Author Response · Authors · 2024-11-24
> > **Response to Reviewer VZKs**
> >
> > > W3. The proof for the parameter lower bound in seq2seq cases relies on hardmax activation, deviating from the real-world scenario where the attention layer adopts softmax activation. Is it possible to extend the proof to the case with softmax activation?
> >
> > The proofs of Theorems 4.2 and 4.3, which establish the lower bound for memorization capacity, rely on the fact that the model’s output can be expressed as a polynomial of its parameters.
> > As such, the current proofs cannot be directly extended to the case with softmax activation.
> > It is worth noting that even for feed-forward networks with sigmoid activations, there remains a significant gap between known upper and lower bounds for memorization capacity.
> > Given this, analyzing softmax activation in this context remains a challenging problem at present.
> > In general, softmax is notoriously difficult to analyze, and theoretical studies often adopt the hardmax function as a substitute for tractability [10, 11].
> >
> > **References**
> >
> > [10] Merrill and Sabharwal, The Expressive Power of Transformers with Chain of Thought. ICLR 2024.
> >
> > [11] P´erez et al., Attention is Turing complete. JMLR 2022.

---

> > > ### Comment · Reviewer_VZKs · 2024-11-27
> > > **Thank you for the rebuttal**
> > >
> > > Thank the authors for addressing my comments. I don't have any other questions and decide to keep my score.

---

### Author Response · Authors · 2024-11-27
**Updates Based on Reviewer Feedback**

Thank you so much for taking the time to review our paper and for your helpful comments. Your feedback has really helped us make the paper clearer and better.
Below, we summarize the revisions and additions made during the rebuttal period based on your responses:
1. **Title Revision**: To enhance precision, we have updated the paper's title to "On the" Optimal Memorization Capacity of Transformers, following the title of prior work [1], which established upper and lower bounds on the memorization capacity of feed-forward networks with a logarithmic factor gap
2. **Proof Sketch of Main Theorem**: We have included more detailed explanations regarding the high-dimensional representations $\tilde{X}^{(1)},\dots,\tilde{X}^{(N)}$ used to construct the feed-forward network in the proof sketch of Theorem 4.1, particularly around eq. (9).
3. **Memorization Capacity with Limited Bit Complexity**: We have added a discussion in Appendix F about cases where the bit complexity of parameters is insufficient for optimal memorization. Theorem F.1 demonstrates that a Transformer constrained to a bit complexity of $\tilde{O}(N^{1 - \alpha})$ for some $\alpha \in [1/2,1]$ can memorize $N$ data points in the next-token prediction setting using $\tilde{O}(N^{\alpha})$ parameters.
4. **Experiments**: To support our theoretical analysis, we conducted experiments on two real-world datasets and one randomly generated datase (details are provided in Appendix H):
    - **Verification of Uniform Attention's Expressive Power**: We trained the model described in Theorem 4.1, that is, a Transformer consisting of single uniform attention layer sandwiched between two token-wise feed-forward networks, on the MNLI and IMDb datasets. The results confirm that given adequately deep feed-forward networks, the training error can be reduced to near zero, which validates that uniform attention actually possesses sufficient representational capacity for memorization.
    - **Analysis of Memorization Capacity**: Using the same model, we trained it on MNLI datasets of varying sizes and plotted the minimum model size required for memorization. The results show that for smaller datasets, the memorization capacity scales as $\sqrt{N}$, consistent with the predictions of Theorems 4.1 and 4.2.
For larger datasets, however, the memorization capacity increases rapidly. A plausible explanation for this phenomenon is that the fixed bit-length of model parameters becomes a bottleneck as the dataset size grows.
In such a regime, the results from Appendix F on Transformers with limited bit complexity suggest that the memorization capacity scales linearly with $N$.
    - **Effect of Input Sequence Length on Memorization Capacity**: We trained the model on randomly generated datasets with fixed data sizes but varying input sequence lengths. The results indicate that the minimum model size required for memorization remains unchanged regardless of input sequence length, consistent with the predictions of Theorem 4.1 that the memorization capacity in the next-token prediction setting is nearly independent of the input sequence length $n$.

Your thoughtful feedback not only helped us address the concerns you raised but also deepened our understanding of the memorization capacity of real-world Transformer models. We are deeply grateful for your valuable insights and for giving us the opportunity to refine and strengthen our work.

Thank you once again for your constructive comments.

**Reference**
[1] Vardi et al., On the Optimal Memorization Capacity of ReLU Neural Networks. ICLR 2021.

---

### Meta-Review · Area_Chair_mNF3 · 2024-12-22

**Metareview:**

This paper studies the number of parameters and the bit complexity required by a transformer to memorize a dataset of size. In the context of next token prediction, the paper proves that for any dataset with well separated tokens, there exists a transformer optimal
 parameters+bit-complexity (both upper and lower bounds.) In the sequence to sequence setting, the paper proves a similar upper bound with, and they prove a matching lower bound for hardmax transformers. All reviewers believe this paper is technically strong, and the AC agrees and recommends acceptance.

**Additional Comments On Reviewer Discussion:**

The reviewers have comments on the writing and experiments. The comments were addressed by the authors.

---

### Decision · Program_Chairs · 2025-01-22

Accept (Poster)